# Designing fuel cell catalyst support for superior catalytic activity and low mass-transport resistance

Muhammad Naoshad Islam[1], Abdul Bashith Mansoor Basha [1],
Vinayaraj Ozhukil Kollath[1], Amir Peyman Soleymani [2], Jasna Jankovic [2] &
Kunal Karan [1]✉

The development of low-Platinum content polymer electrolyte fuel cells (PEFCs) has been hindered by inexplicable reduction of oxygen reduction reaction (ORR) activity and unexpected $O_2$ mass transport resistance when catalysts have been interfaced with ionomer in a cathode catalyst layer. In this study, we introduce a bottom-up designed spherical carbon support with intrinsic Nitrogen-doping that permits uniform dispersion of Pt catalyst, which reproducibly exhibits high ORR mass activity of $638 \pm 68$ mA $mg_{Pt}^{-1}$ at 0.9 V and 100% relative humidity (RH) in a membrane electrode assembly. The uniformly distributed Nitrogen-functional surface groups on the carbon support surface promote high ionomer coverage directly evidenced by high-resolution electron microscopy and nearly humidity-independent double layer capacitance. The hydrophilic nature of the carbon surface appears to ensure high activity and performance for operation over a broad range of RH. The paradigm challenging large carbon support (~135 nm) combined with favourable ionomer film structure, hypothesized recently to arise from the interactions of an ionic moiety of the ionomer and Nitrogen-functional group of the catalyst support, results in an unprecedented low local oxygen transport resistance (5.0 s cm⁻¹) for ultra-low Pt loading ($34 \pm 2$ $\mu g_{Pt}$ cm⁻²) catalyst layer.

Hydrogen fuel cells are experiencing a resurgence as zero-emission power source for vehicles, especially mid-size and heavy-duty vehicles and light railway transits. For large-scale commercialization, a key developmental target for these polymer electrolyte fuel cells (PEFCs) is to reduce the expensive Platinum (Pt) catalyst content in a 100 kW stack from 30 g to <10 g. To attain this target, we need: (a) a catalyst with high mass activity for oxygen reduction reaction (ORR)[1–3] to reduce kinetic losses, (b) a catalyst layer (CL) microstructure designed for facile oxygen transport to mitigate the unexpectedly high local transport losses observed in low-Pt loading CLs[4,5] and (c) a high coverage of ionomer to ensure proton accessibility of the catalyst sites to maximize Pt utilization. Independent studies by General Motors,

Toyota, and Nissan researchers showed that reduction of total Pt loading in the cathode (<0.1 $mg_{Pt}$ cm⁻²) to attain lower Pt content in a stack resulted in a surprisingly much lower cell performance and an increased $O_2$ transport resistance beyond what would be expected due lower amount of catalyst[4–9]. This drove the development of several new catalysts, including alloys and shape-controlled nanoparticles with impressive oxygen reduction reaction (ORR) activity (20× activity of pure Pt catalyst) in liquid electrolyte[10–12]. Inexplicably, these catalysts have failed to exhibit the same high ORR activity in the membrane electrode assembly (MEA) of a fuel cell[13]. Concerns regarding the retention of the physical structure (nanoframe catalysts) or the chemical features (alloy composition, core-shell architecture) upon their

[1]Department of Chemical and Petroleum Engineering, University of Calgary, Calgary, AB, Canada. [2]Center for Clean Energy Engineering, Institute of Materials Science, and Materials Science and Engineering Department, University of Connecticut, Storrs, CT, USA. ✉e-mail: kkaran@ucalgary.ca

integration in a CL have also been raised[14]. A promising alternative that can alleviate these concerns is the use of high activity, pure Pt catalyst tailored for its particle size within the 1.8-3 nm range to maximize surface sites with generalized coordination number (GCN) between 7.5 and 8.3[15]. Separately, recent studies by Strasser and Gasteiger groups postulate that Nitrogen-functionalized carbon (N-C) support for Pt improves both ionomer coverage and local oxygen transport resistance[16,17], and another very recent study by the Toyota group reports that upon coating Pt/C catalysts with ~1 nm dopamine layer, the mass activity was enhanced but the local oxygen transport resistance increased[18]. Yet, the use of 20−30 nm carbon black as catalyst support has remained unchanged. Such small catalyst support particles limit the CL pores to dimensions comparable to the mean free path length of oxygen ($\lambda_{O2,mfpl}$), thereby constraining the gas-phase oxygen transport in a CL to occur by the restrictive Knudsen diffusion rather than by molecular diffusion[19]. Simulation work by Secanell group has shown that pore sizes scale up with the size of the support size[20], as would be expected from a packing of spheres.

Herein, we propose and implement a catalyst/support design strategy−size-controlled Pt nanoparticles that are well-dispersed on large carbon support with uniformly distributed surface Nitrogen group−to create a high-activity, low $O_2$ transport resistance catalyst layer. The design strategy shown in Fig. 1 comprises four targets: (i) control Pt nanoparticle size to 1.8−3 nm range to yield optimal GCN, (ii) attain uniform dispersion of Pt nanoparticle on the catalyst support with high Pt-to-Pt distance ($l_{Pt\text{-}Pt}$) to mitigate the 'territory effect' originally suggested by Watanabe et al.[21], (iii) paradigm challenging use of large spherical catalyst support (-130 nm) to create large CL pores to enable facile CL $O_2$ transport by molecular diffusion rather than by restrictive Knudsen diffusion, and (iv) a uniform distribution of N-functional groups on a carbon-based catalyst support surface for favorable support-carbon interactions as suggested by Ott and colleagues[16,17].

## Catalyst support development

Uniformly sized polymerized dopamine (PDA) particles spherical in shape and 215 ± 11 (SD) nm in diameter (Fig. 2a) are synthesized by a facile, one-step aerobic polymerization method[22] optimized for size and poly-dispersity control (details in method section). TEM image (Fig. 2b) confirmed that the shape and structure of the PDA nanoparticles were retained upon carbonization. However, the carbonized PDA (cPDA) nanoparticles were smaller compared to the PDA nanoparticles, i.e., a 37% reduction from 215 nm to 135 nm. A previous study utilizing high-resolution TEM and Raman spectroscopy had revealed that cPDA nanoparticles possess graphite-like nanostructure consisting of several tens of stacking carbon layers[23]. Upon carbonization, the Brunauer−Emmett−Teller (BET) surface area of PDA particles increased from 17 m² g⁻¹ to 365 m² g⁻¹, as indicated by N₂-adsorption data (Supplementary Table S1). The micro-/meso-pores contributed to -82% of the total BET surface area of cPDA (Supplementary Table S1).

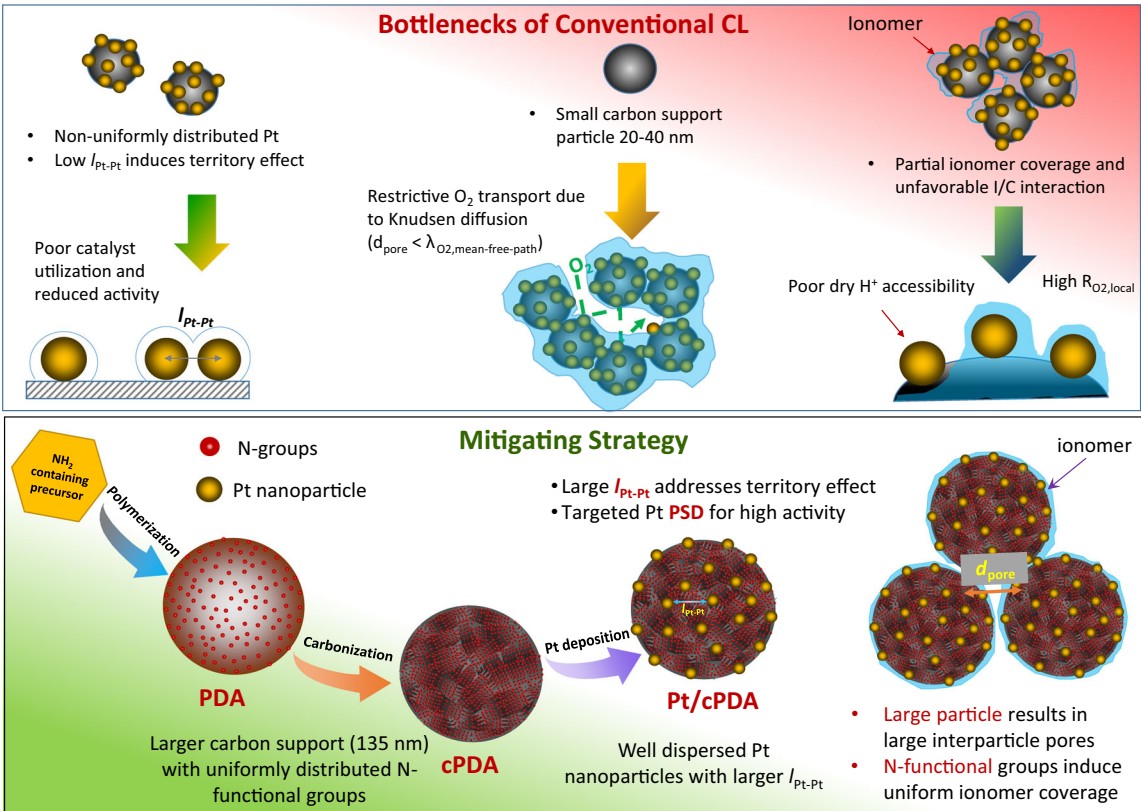

**Fig. 1 | Schematic diagram depicting the bottlenecks of conventional CL and new catalyst design strategy.** Top Panel: In conventional Pt/C catalysts, Pt nanoparticles are distributed non-uniformly due to the non-uniform surface functional group, which results in low platinum interparticle distance ($l_{Pt\text{-}Pt}$) inducing territory effect that leads to reduced mass specific activity and poor catalyst utilization at high current densities. The usage of small carbon particles (-20-40 nm) results in catalyst layers with small pore diameter ($d_{pore}$), which leads to restrictive gas-phase oxygen transport dominated by Knudsen diffusion. Lack of surface functional groups or non-uniform distribution of surface functional groups result in non-uniform ionomer coverge of the Pt/C catalyst, which further limit the proton accessibility of Pt catalyst under low RH (dry) conditions. Bottom Panel: To mitigate these issues, a NH₂-containing small molecule is polymerized to create large spherical particles with well-distributed N-functional group on the particle surface. The particle is carbonized prior to Pt deposition. The uniformly distributed N-functional groups on carbonized particles enables deposition of uniformly distributed Pt and size-controlled nanoparticles with large Pt-Pt interparticle distance ($l_{Pt\text{-}Pt}$)which addresses territory effect. Large carbon support results in large pores improving gaseous oxygen transport while the uniformly distributed N-functional group hypothesized to have favorable interaction with the ionomer induces high ionomer coverage, which improves catalyst utilization.

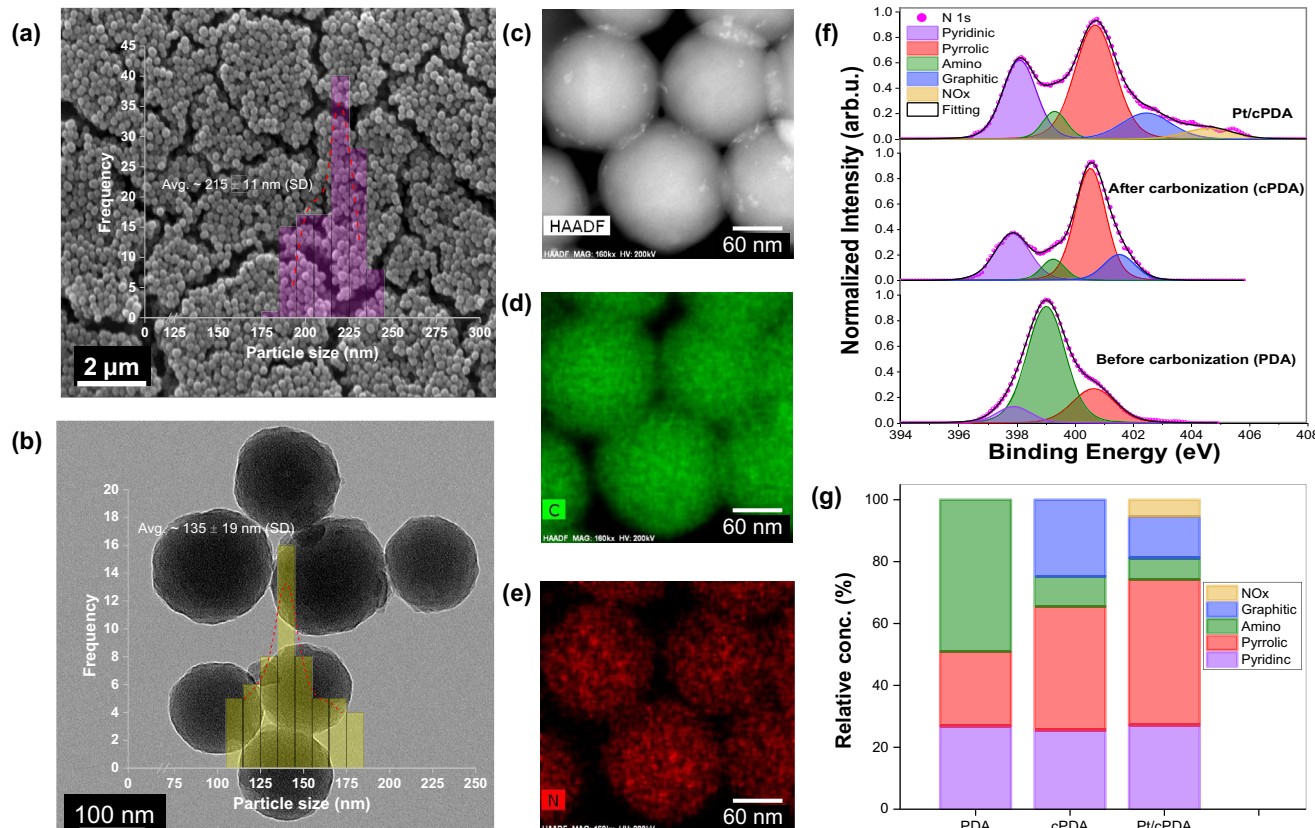

**Fig. 2 | Physical and chemical characteristics of catalyst support. a** SEM micrograph with particle size distribution (PSD) (inset) of PDA nanospheres, **b** TEM micrographs with PSD (inset) of cPDA nanospheres, **c**–**e** STEM energy-dispersive spectroscopy (EDS) mapping of cPDA nanospheres, **f** high-resolution deconvoluted XPS spectra of N1s for PDA, cPDA and Pt/cPDA, **g** relative concentration of nitrogen species in PDA, cPDA and Pt/cPDA samples (estimated by deconvoluting the XPS spectra of N1s). The standard deviation presented in figures **a** and **b** is obtained by analyzing 109 particles for PDA and 52 particles for cPDA.

The bottom-up approach of creating catalyst support particles using N-containing precursor ensures uniformly distributed Nitrogen groups on the cPDA surface. The TEM/EDS images (Fig. 2c–e) confirm the uniform distribution of Nitrogen on the cPDA spheres. This homogenous distribution of N-species plays an instrumental role in the uniform spatial distribution of Pt particles during the catalyst deposition process, leading to high inter-particle distance ($l_{Pt-Pt}$). Identification of N-functional groups by X-ray photoelectron spectra (XPS) also revealed the chemical changes upon carbonization of PDA and subsequent interaction of Pt with cPDA. XPS survey spectra (Supplementary Fig. S2a) reveals the presence of N 1s (~400 eV), C 1s (~284 eV), and O 1s (~532 eV) peaks for all the samples, and Pt 4f (~72 eV) as well as Pt 4d (~314 eV) peaks for Pt/cPDA sample. High-resolution XPS spectra permitted deconvolution of C1s, O1s and Pt 4f peaks (Supplementary Fig. S2b-d) as well as of N 1s peak, which led to the identification of Nitrogen functional groups (Fig. 2f)—pyridinic (~398 eV), amino (~399 eV), pyrrolic (~400.5 eV) and graphitic (~401.5 eV)[24,25]. Quantitative analyses of the peaks (Fig. 2g) reveal that PDA carbonization resulted in a dramatic decrease in amino groups, a slight increase in the pyrrolic group and an evolution of the graphitic-N group, whereas the pyridinic remained relatively unchanged. Subsequent Pt deposition caused a slight positive shift in pyridinic peak and a significant shift in graphitic-N peak, which arises from Pt-N interactions resulting in a decrease in electron density of the neighboring atom manifesting as an increase in binding energy and peak shift.

## Pt/cPDA catalyst characterization

The synthesized Pt/cPDA catalyst were characterized for key physical properties to ascertain that first of the four design targets, i.e.,

high Pt dispersion and size control in 2–3 nm range, was achieved. In addition, the location of Pt catalyst (surface versus embedding within micropores of cPDA) and the Pt content on cPDA were determined.

The uniform distribution of Nitrogen-rich moieties (3.5 at% – from XPS data) and pyridinic type Nitrogen on the cPDA surface are expected to induce uniform nucleation of Pt, resulting in high spatial dispersion of Pt nanoparticles[26,27] and controlled Pt size. TEM micrograph (Fig. 3a) of Pt nanoparticles deposited cPDA nanospheres provides direct evidence of very well-dispersed Pt nanoparticles. The average inter-particle distance (AID) calculated via equation S1 (see Supplementary information) developed by Meier et al.[28] yields a value of ~26 nm for Pt/cPDA, which is larger than the ~16 nm AID for commercial Pt/C (TKK 10 wt% Pt on Vulcan carbon) catalyst. For poorly dispersed catalysts with short Pt-to-Pt nearest neighbor distance ($l_{Pt-Pt}$) or AID[21,29], the effective ORR activity is adversely affected by the so-called territory effect, first proposed by Watanabe and colleagues[21]. The physical location of Pt particles determines if they interface with ionomer or not, depending on whether they are on the carbon support surface or embedded within micropores. Pt particles, if present in micropores are not prone to poisoning by the sulfonic group of the ionomer, thereby are thought to exhibit higher activity, but such Pt catalyst particles may have proton accessibility problems[30,31]. The 3D TEM of Pt/cPDA catalysts confirmed that the Pt particles were located on the surface of the cPDA support and not within the micropores (see video as supplementary information). STEM images of a ~100 nm thin microtomed CL slice (Fig. 3d and Fig. S1) show cross-sections of the cPDA particles with Pt placed only on the surface of the particles and their absence inside the cPDA support.

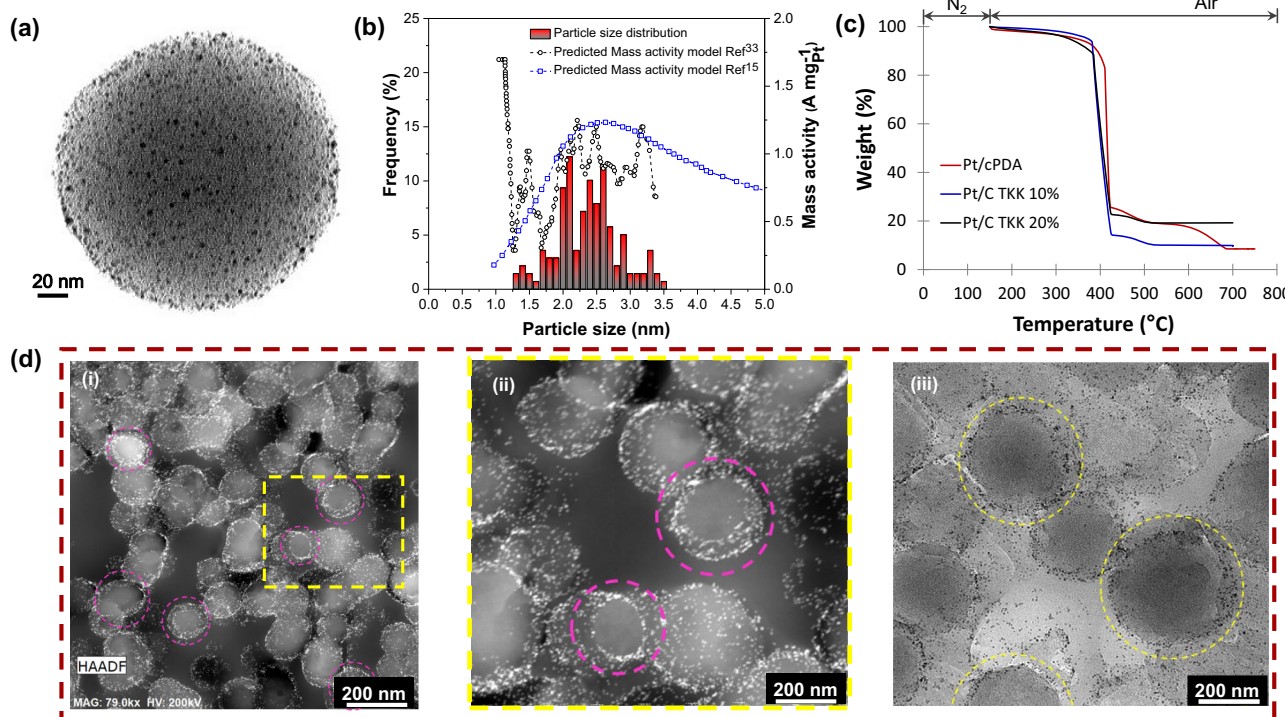

**Fig. 3 | Physical characteristics of the Pt/cPDA catalyst. a** TEM micrograph of Pt deposited cPDA nanospheres (Pt/cPDA), **b** PSD histogram of deposited Pt nanoparticles and computationally estimated mass activity of Pt at 0.9 V ($i_{m,0.9v}$ vs RHE) digitized from refs. 15, 33. **c** TGA thermogram of Pt/cPDA and TKK Pt/C (10.2% and 19.8%) catalysts for the determination of Pt content (details in supplementary information), and **d** STEM images of microtomed Pt/cPDA CL in−(i) and (ii) dark field and (iii) bright field mode. The marked regions are the cross-section of Pt/cPDA catalysts highlighting that the Pt particles are located at the exterior of the cPDA support.

Another design target for our Pt/cPDA catalyst is to control the Pt particle size to a range of 2–3 nm, which was determined from a theoretical study to possess the highest ORR activity[32]. The TEM image analyses of the Pt/cPDA catalyst (Fig. 3b) revealed that a majority of the particles are in the 1.8–2.5 nm range, which is expected to exhibit the high mass activity of 0.9–1.2 A $mg_{Pt}^{-1}$ as per the GCN correlation[15,33]. The Pt surface area computed from this size distribution was determined to be 117 $m^2$ $g_{Pt}^{-1}$, which was close to the value of 107 ± 4 $m^2$ $g_{Pt}^{-1}$ (via $H_{ad}$) obtained from ECSA measurement of the ionomer-free catalyst in 0.1 M $HClO_4$ electrolyte. The Pt content of Pt/cPDA catalyst (wt% Pt in Pt+cPDA), required for ECSA calculation, was determined from thermogravimetric analyses (Fig. 3c, Table S2). The Pt content of two commercial Pt/C TKK with supplier stated Pt loadings of 10.2 and 19.8 wt% were determined from TGA measurements to be 9.5 wt% and 19.5 wt%, respectively. The Pt content of Pt/cPDA catalyst from TGA measurement was determined to be 8.5 wt%. The Pt content determined from image analyses yielded a comparable value of 7.4–8.5 wt% (Supplementary Information, Table S3).

### Characterization of Pt/cPDA CLs
Successful implementation of our catalyst design strategy should manifest in a Pt/cPDA catalyst layer that exhibits: (i) high ionomer coverage arising from favorable interaction between ionomer and N-containing group, (ii) high ORR activity in MEA (and not merely in the liquid electrolyte) and (iii) improved local oxygen transport resistance.

STEM/EDX mapping reveals a Fluorine signal over the Pt/cPDA particles indicating high ionomer coverage (Fig. 4a). The double layer capacitance ($C_{DL}$) determined from the cyclic voltammograms in MEA over 30–100% RH range (Fig. 4b) indicates that $C_{DL}$ remain almost unchanged (~1 F $m^{-2}_{cPDA}$) further indicating well hydrated Pt/cPDA-ionomer interfacial structure. This value is also within 10% of the $C_{DL}$

determined by CV in $HClO_4$, where support accessibility by electrolyte is maximum. The weak dependency of ECSA on RH also supports the high ionomer coverage. The ECSA in MEA was 113 $m^2$ $g_{Pt}^{-1}$ via CO stripping (101 ± 5 $m^2$ $g_{Pt}^{-1}$ via $H_{ad}$) for Pt/cPDA and 70 $m^2$ $g_{Pt}^{-1}$ (via CO stripping) for commercial Pt/C catalyst (TKK 10 wt% Pt/C), respectively. The ECSA of Pt/cPDA calculated from TEM-based average particle size is around 117 $m^2$ $g_{Pt}^{-1}$, almost similar compared to the $ECSA_{MEA}$ and $ECSA_{Liquid\ electrolyte}$. The dry proton accessibility ($\frac{ECSA_{30\%RH}}{ECSA_{100\%RH}}$) was calculated to be ~80% (using CO stripping; Fig. S13) for Pt/cPDA CL compared to ~70% for TKK Pt/C CL. A dry proton accessibility value of 0.8 compared to that of 0.7 for TKK Pt/C CL might indicate higher ionomer coverage or more homogenous ionomer distribution in the Pt/cPDA CL. Similar dry proton accessibility (80%) has also been reported by Padgett et al.[34] for 10 wt% Pt/V sample. However, the CL composition was different−Padgett et al.[34] employed a higher ionomer to carbon ratio (0.95 vs 0.8 in this study), and the ionomer had a shorter side chain ionomer (EW-950 vs 1100). The hydrophilic nature of cPDA, likely due to N-functional groups[35], as noted by spreading of water on a layer of particle, may help to keep the ionomer/Pt and ionomer/cPDA interfaces well hydrated. Thus, the slightly high ionomer coverage coupled with the hydrophilic nature of support is expected to result in a well-hydrated Pt/ionomer interface even in dry conditions facilitating the proton accessibility to most of the Pt particles.

A critical design target for the developed Pt/cPDA catalyst is the attainment of high ORR activity in MEA under fuel cell conditions. ORR activity for ionomer-free Pt/cPDA (in 0.1 M $HClO_4$) determined from rotating disk electrode measurements yielded specific and mass activity of 0.95 ± 0.07 mA $cm_{Pt}^{-2}$ and 944 ± 10 mA $mg_{Pt}^{-1}$ (Supplementary Fig. S7, Table S4), respectively at 0.9 V in $O_2$ saturated electrolyte. Upon incorporation in MEA, Pt/cPDA exhibited area-specific and mass activity of 0.632 ± 0.06 mA $cm_{Pt}^{-2}$ and 638 ± 68 mA $mg_{Pt}^{-1}$ (Fig. 4c, Table S6) at 0.9 V in a fuel cell operated

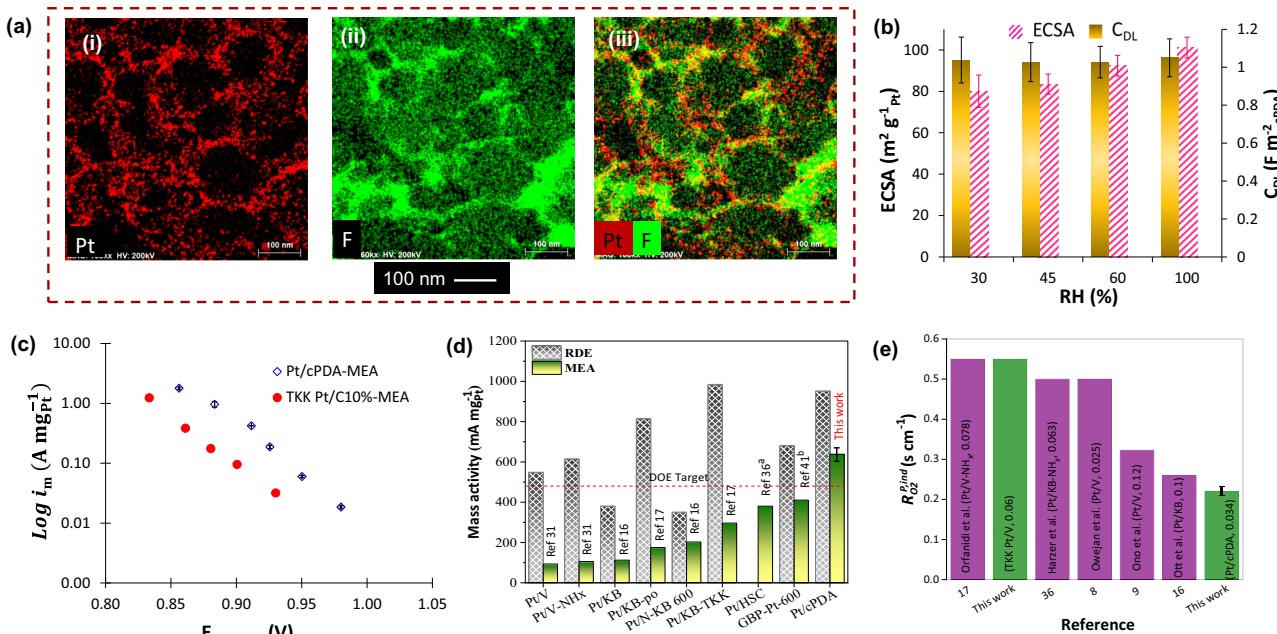

**Fig. 4 | Physical and electrochemical characteristics of Pt/cPDA based catalyst layers. a** STEM-EDX mapping of Pt/cPDA CL showing the ionomer and Pt distribution, **b** Effect of RH on ECSA (H_ad) and $C_{DL}$ of Pt/cPDA CL, **c** Mass activity Tafel plot comparison between Pt/cPDA and TKK Pt/C catalyst in MEA condition[*] (**d**) $i_{m,0.9V}$ comparison in both RDE and MEA condition between Pt/cPDA[*] and different Pt catalysts reported in literature[16,17,31,36,48] (all the reported RDE activity values in (**d**) were performed in the ionomer-free system except Pt/HSC and GBP-Pt-600), **e** comparison of pressure independent $O_2$ transport resistance between Pt/cPDA and other Pt-based catalysts reported in the literature[8,9,16,17,36] (values inside the bracket indicate Pt loadings in mg_Pt cm^-2), [*]Pt/cPDA MEA condition: 70 °C, 100% RH (80% RH for $O_2$ transport resistance), 140 kPa_abs, $H_2/O_2$ for activity measurements. E was corrected for iR-losses, and current density was corrected by adding crossover current to the measured current. Literature data recorded at 80 °C, 100% RH, 150 kPa_abs, $H_2/O_2$, [a]$i_{m,0.9V}$ was assumed to be 150 kPa_abs; operating pressure was not mentioned; [b] $i_{m,0.9V}$ measured at 300 kPa_abs, recalculated at 150 kPa_abs following the method explained in ref. 49. The error bar in **b**–**d** are deviation from the average of data from three individual testings, except for the mass transport resistance (**e**), which was repeated twice.

at 70 °C, 100% RH and 140 kPa_abs pressure in $O_2$, which exceeds the DOE target (0.44 A mg_Pt^-1). A comparison with other Pt-based catalysts reported further confirms the impressive attribute of Pt/cPDA catalysts (Fig. 4d). Pt/cPDA illustrated almost 1.7 times higher activity compared to the next best-reported activity of 372 mA mg_Pt^-1 for Pt/KB_TKK catalyst by Harzer and coworkers[36] in an MEA. The nanoframed Pt₃Ni reported by Stamenkovic and co-workers[37] with a staggering mass activity of 5.7 A mg_Pt^-1 at 0.9 V in a liquid electrolyte but with the much-reduced activity of 760 mA mg_Pt^-1 in MEA is the only catalyst with comparable activity to the Pt/cPDA catalyst in an MEA. The much smaller suppression of ORR activity of Pt/cPDA catalyst in MEA due to ionomer poisoning was also observed in the liquid electrolyte (0.1 M HClO₄), as evident from the comparison of activity between ionomer-free and ionomer-containing films (Fig. S7; Table S4). Whereas 42% suppression of activity was observed for commercial Pt/C catalyst (10 wt% TKK) at an I:C ratio of 0.2, consistent with the literature (Fig S8), Pt/cPDA catalyst at a similar I:C ratio showed 24% suppression. The exact role of cPDA support in minimizing the activity suppression is not fully understood and may be due either to the ionomer/cPDA interactions affecting the Pt/ionomer interfacial structure or to the nucleation and growth of Pt crystallites with facets less prone to ionomer poisoning. Any contribution of the N-containing group to ORR activity was quantified by testing Pt-free cPDA in MEA condition and observed a very low, 13 μA cm^-2 (~10 μA mg_cPDA^-1) current at 0.9 V at 140 kPa, 70 °C and 100% RH (Supplementary Fig. S15a). This confirmed that the superior ORR activity of Pt/cPDA catalyst is not due to unaccounted contribution from N-functional support. The observed ORR activity of cPDA support may be ascribed to the edge plane sites of pyridinic nitrogen, which facilitates the adsorption of oxygen[38,39].

The last design target for Pt/cPDA catalyst is the realization of low local oxygen transport resistance (R_O2) hypothesized to result from the combined effect of enhanced gas phase mass transport in the larger pores of CL and better $O_2$ transport in the ionomer film with better permeability hypothesized to arise from the open structure of ionomer film due to N-functional group/ionomer interactions[16]. Limiting current measurements[40] for a range of $O_2$ concentration (2–6 mol%) and several total pressure (140–290 kPa) was undertaken at 80% RH to quantify $O_2$ transport characteristics—total ($R_{O2}^{total}$), pressure-dependent ($R_{O2}^{P,dep}$), and -independent ($R_{O2}^{P,ind}$) oxygen transport resistances (Fig. S21). The breakdown of $R_{O2}^{total}$ and calculation of $R_{O2}^{P,dep}$ and ($R_{O2}^{P,ind}$) is shown in Equations S4, S5 and S6 in Supplementary Information. Many different factors, including porous transport layer (carbon paper), flow-field configuration and catalyst layer microstructure, influence the cell oxygen transport resistance (and overall cell performance), as briefly discussed in Supplementary Information. Thus, a comparison of total R_O2 is not meaningful unless similar component materials and cell design were accounted for. It has been emphasized in the literature that the main concern with low-Pt content cathodes (especially when the roughness factor falls below 50) is the unusually high pressure-independent transport resistance ($R_{O2}^{P,ind}$)[4–9] which is the cumulative effect of gas-phase Knudsen diffusion in the CL and the microporous layer as well as of $O_2$ transfer/transport in ionomer film coating the catalyst/support. The $R_{O2}^{P,ind}$ for Pt/cPDA CL is the lowest compared with other low Pt-loaded electrodes reported in the literature[7–9,16,17,36]. The Pt/cPDA CL also exhibited significantly lower pressure-dependent ($R_{O2}^{P,dep}$) transport resistance (0.25 ± 0.04 s cm^-1)

compared to the in-house CL made with commercial Pt/C TKK 10 wt% catalyst (-0.55 s cm$^{-1}$) (Fig. 4e). The pressure-independent $O_2$ transport resistance of Pt/cPDA CL is even lower than those reported for Pt supported on N-functionalized KB CLs (Fig. 4d) reported in recent works[16,17] despite having lower Pt loading (0.034 mg$_{Pt}$ cm$^{-2}$ compared to that 0.1 mg$_{Pt}$ cm$^{-2}$). Remarkably, the $R_{O2}^{P,ind}$ for Pt/cPDA CL at different roughness factor (Fig. 4e, extracted from Supplementary Fig. S21d) is significantly lower than those pure-Pt based CL with similar low RF or low-Pt loading. In fact, it appears that the $R_{O2}^{ind}$ trendline observed for Pt/cPDA CLs compared to trendline observed for CLs made of Pt supported on carbon black particles with ~30 nm average diameter are remarkably shifted to left and downwards, i.e., toward a lower $R_{O2}$ (Fig. S21f). The Pt/cPDA CL exhibit a $R_{O2}^{P,ind}$ of 0.25 ± 0.04 s cm$^{-1}$ at RF of 35 ± 2 (via H$_{ad}$) compared to the $R_{O2}^{P,ind}$ of 0.23 s cm$^{-1}$ for Pt catalyst on N-functionalized Ketjen black (KB-N) support at much higher RF of 60 reported in a recent work[16]. Following the approach in Ott et al.[16], the contribution of Knudsen diffusion and transport/transfer through ionomer to $R_{O2}^{P,ind}$ was estimated (Fig. S21e). As hypothesized, the large cPDA particles result in dramatic decrease in the Knudsen resistance, which is estimated to be around 0.04 s cm$^{-1}$ compared to the values of 0.14 s cm$^{-1}$ and 0.08 s cm$^{-1}$ reported for Pt/KB and Pt/KB-N CLs[16]. Despite having a higher estimated effective average ionomer thickness (~6 nm vs ~3.4 nm for Pt/V) due to the higher contribution from micropores in the Pt/cPDA catalyst, lower ionomer-related resistance of 5.0 s cm$^{-1}$ for Pt/cPDA is observed compared to the closest value of 8.63 s cm$^{-1}$ reported recently for Pt/KB-N[16]. On the other hand, we must consider the Pt-Pt interparticle distance or catalyst dispersion in mitigating the territory effect resulting in lower $R_{O2}^{P,ind}$, which was observed by Owejan et al.[8] upon decreasing

Pt particle density at a particular Pt loading. For Pt/cPDA CLs, the lower oxygen transport resistance can be attributed at least to the dual effects of our design strategy: first, a facile gas-phase transport in large CL pores and second via the mitigation of the territory effect.

## Performance and durability characterization

The performance and durability characteristics of Pt/cPDA CL is reported for the sake of completeness of catalyst layer characterization. As with $R_{O2}^{total}$, a direct comparison of the performance of fuel cells is fraught with challenges of delineating the differences arising from different cell components, including membrane (type and thickness) and porous transport layer, as well as cell hardware (most notably flow-field plate design). Accordingly, for comparison, results from two cells varying only in the in-house made cathode CLs−Pt/cPDA and commercial Pt/C (TKK 10 wt% Pt on Vulcan carbon)−are reported. Polarization measurements were conducted under fuel cell conditions at varying RH (30−100%) (70 °C, 140 kPa$_{abs}$ in both Air and pure $O_2$). The cells with Pt/cPDA CLs exhibited higher ohmic resistance (high-frequency resistance or HFR) compared to cells with Pt/C CLs. Although the origin of high HFR for Pt/cPDA is not known, it is speculated to arise from contact resistance between CL and MPL. The HFR-free performance data are included in SI (Fig. S16). Overall, the Pt/cPDA catalyst layer illustrated better-uncorrected performance (560 mA cm$^{-2}_{geo}$ compared to that of 330 mA cm$^{-2}_{geo}$ of TKK Pt/C CL) as well as better HFR-free performance (700 mA cm$^{-2}_{geo}$ compared to that of 420 mA cm$^{-2}_{geo}$ of TKK Pt/C CL) on an electrode geometric area basis at 0.6 V in H$_2$/Air (Fig. 5a, Supplementary Fig. S16). TKK Pt/C CL showed a sharp drop in performance in the transport-dominated region (>1000 mA cm$^{-2}_{geo}$) due to higher transport resistance as opposed to the Pt/cPDA CL. Although the Pt loading for the reference CL was higher, the roughness factor of both CLs was similar (~40 cm$^2_{Pt}$ cm$^{-2}_{geo}$)

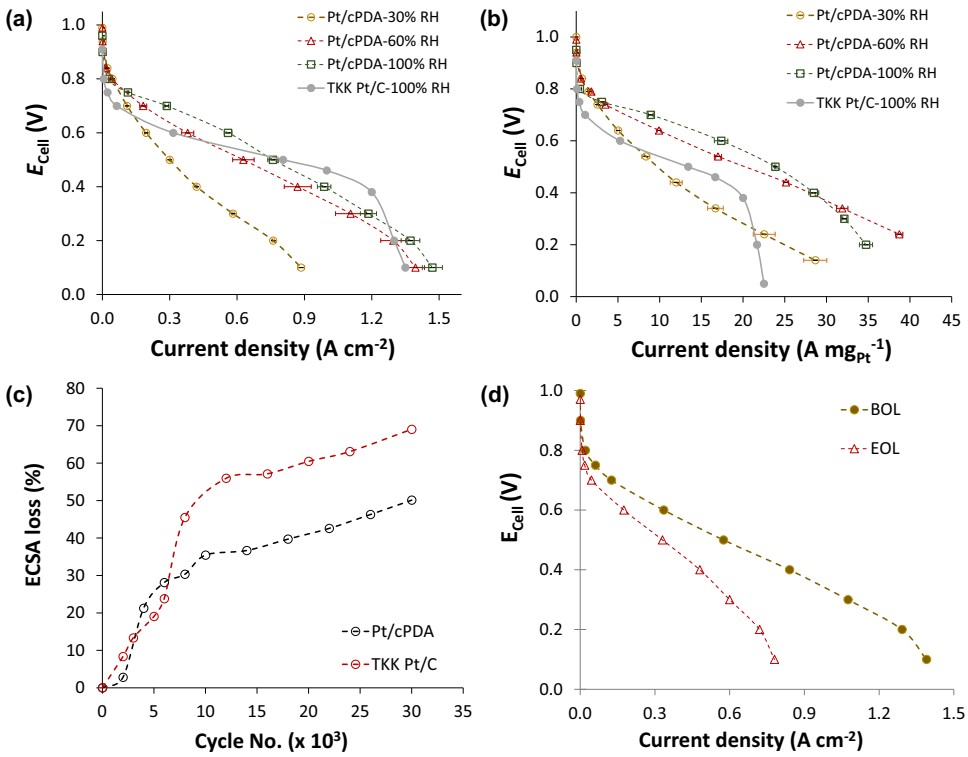

**Fig. 5 | Fuel cell performance and catalyst durability comparison for MEAs made with Pt/cPDA and commercial Pt/C catalysts. a** Uncorrected cell performance and **b** Uncorrected Pt mass loading normalized performance comparison between Pt/cPDA and commercial Pt/C (TKK 10 wt% Pt) catalyst in H$_2$/Air#, **c** ECSA loss during square wave AST degradation cycles (AST protocol: DOE square wave, 0.6–0.95 V, potential changed at ~700 mV s$^{-1}$), at 70 °C, 100% RH and atmospheric pressure in H$_2$/N$_2$ (0.2/0.2 NLPM), **d** Uncorrected cell performance at BOL and EOL (after 30,000 AST cycles) at 70 °C, 100% RH and 140 kPa pressure in H$_2$/Air, ECSA was measured using CO stripping method. The error bar presented is the average of three individual testing results. #MEA condition: 70 °C, 100% RH; 140 kPa$_{abs}$, H$_2$/Air (0.3/0.5 NLPM) for performance.

(thickness, Pt/cPDA CL – 9 ± 1 μm, Pt/C TKK CL – 10 ± 1 μm). On the other hand, when normalized with Pt loading Pt/cPDA CL demonstrates noteworthy improvement in performance (almost threefold increase in mass activity at 0.6 V in Air) compared to TKK Pt/C CL at the same condition (see Supplementary Fig. S18) and convincingly surpasses (~20 A mg$_{Pt}^{-1}$, ~12 kW g$_{Pt}^{-1}$) the DOE Pt mass-specific power requirement (8 kW g$_{Pt}^{-1}$)[31] at 140 kPa in Air (see Fig. S20b).

The durability of the Pt/cPDA catalyst and the reference TKK Pt/C catalyst was also assessed over 30,000 potential cycles following the DOE-suggested square wave (SW) accelerated stress test (AST) protocol (0.6–0.95 V, at 70 °C, 100% RH and atmospheric pressure in H$_2$/N$_2$ (0.2/0.2 NLPM), details in Method, Supplementary information) with the only exception that the temperature was 70 °C instead of 80 °C. After 30,000 cycles, the ECSA loss was around 50% for the Pt/cPDA catalyst, whereas the reference TKK Pt/C catalyst suffered around 70% ECSA drop (Fig. 5a), which is in good agreement with literature[41]. There was a significant drop in the performance (Fig. 5d, Fig. S22a) and the mass activity (Fig. S22b) of the Pt/cPDA catalyst layer subjected to 30,000 cycles of AST. Analogous to the ECSA drop, the current density at 0.6 V (H$_2$/Air) also dropped by almost a factor of 2. A sharp 30% ECSA loss was observed after 6000 cycles which was followed by a slower decrease resulting in 50% loss by 30,000 cycles (Fig. 5c). Nevertheless, the Pt/cPDA catalyst exhibited lower degradation compared to the reference TKK Pt/C catalyst. It is a well-known fact that in addition to improved catalyst anchoring, dispersion and ionomer distribution, N-moieties also enhance catalyst durability[42–44].

## Discussion

The successful implementation of the catalyst/support design strategy demonstrated all proposed advantages of deploying large-sized catalyst support that has uniformly distributed N-functional group on the surface: (a) attainment of uniform and well-dispersed deposition of Pt catalyst NPs with controlled size and inter-particle distance, both of which result in effective high ORR activity; (b) achievement of high ionomer coverage of Pt/support via favorable interaction between ionomer and N-functional groups posted by Ott and colleagues[16] and thereby high catalyst utilization; (c) decrease in CL oxygen transport resistance affected by larger CL pores and the territory effect. The outcome is the achievement of two key developmental goals for PEFC technology—(i) high ORR activity of pure-Pt-based catalyst under real fuel cell conditions, well exceeding the US DOE target, and (ii) demonstrable reduction in O$_2$ transport resistance for low-Pt content catalyst layer. In addition, the hydrophilic nature of cPDA may be contributing to an enhancement in overall ORR activity. It is speculated that hydrophilic cPDA may result in higher interfacial water, which can then result in higher ORR activity consistent with higher activity for Pt/C catalyst upon dilution of perchloric acid[45], lower energy for oxygen transfer into liquid phase[46], and reduced poisoning by sulfonic acid upon hydration[47]. These findings highlight the beneficial effects of deploying seemingly counterintuitive large-sized catalyst support, which, combined with a coating of uniformly distributed catalyst-nucleating, N-functional groups, could herald the development of the next-generation catalyst layer through a rational strategy for catalyst support design. For example, in a very recent study from Toyota Central R&D group[18], coating the conventional platinized carbon catalyst (30 nm carbon support) with ~1 nm dopamine was found to be beneficial for ORR activity, but the oxygen transport resistance was compromised with their approach. On the other hand, the approaches to minimize the high HFR for Pt/cPDA CLs must also be addressed if such materials are to be incorporated into MEA. Our results indicate that a more rational approach would be to start with a larger carbon or other electron-conducting support particles, coat it with thin layer of dopamine, carbonize the coating, and then deposit the Pt nanoparticles to achieve Pt size control and dispersion. Enhancement in oxygen transport resistance would then be achieved through dual

benefits of large carbon support particles and favorable interaction of ionomer with N-containing functional groups of the dopamine coating. Further investigation would be needed to determine optimal support size and to delineate the effect of the N-functional group in controlling ionomer-support interaction.

## Methods

### Synthesis of PDA nanospheres

The monomer (Dopamine hydrochloride, 1 g) was dispersed in DI water in a 20 mL glass vial. A dispersion medium was prepared separately by mixing water-ethanol with 2.1 vol% of (aq) NH$_4$OH (aq, 28–30 vol%) in a 500 mL glass beaker. Subsequently, the dopamine dispersion was added slowly to the dispersion medium; here, dopamine concentration was kept constant at 3.3 mg mL$^{-1}$ while maintaining the final water:ethanol ratio at 67:33 vol%. Then, the mixture was stirred for 4 h at 1600 rpm in the presence of Air at room temperature (23 °C). Once polymerized, the sample was centrifuged at 27216 × g (15,000 rpm) (Beckman coulter Avanti J-26S XP, USA) and washed with an ample amount of DI water and ethanol to remove unreacted monomer and other residues. The sample was re-dispersed in DI water and ethanol by sonication and then centrifuged again. The procedure was repeated three times till the dispersion media became pH neutral. The sample was then vacuum dried at 60 °C. The overall yield (starting from monomer to final recovered dry PDA spheres) was calculated to be around 65% (gravimetrically). Carbonization of the PDA nanoparticles was carried out at 700 °C under an N$_2$ atmosphere which yielded smaller-sized but spherical carbonized PDA (cPDA) nanoparticles. The carbonization protocol featured three stages: (i) heating from room temperature to 700 °C with a heating rate of 5 °C/min; (ii) a dwell time of 1 h at 700 °C for complete carbonization; and (iii) slow, uncontrolled cooling to room temperature.

### Platinum deposition on cPDA nanospheres

500 mg of cPDA were added to 140 mL of Ethylene Glycol (EG) in a 400 mL round bottom flask and sonicated for around 30 min, followed by magnetic stirring at 1600 rpm to obtain a well-dispersed slurry. In order to achieve a 10 wt% Pt loading on cPDA, 105 mg of H$_2$PtCl$_6$ was dissolved in 35 mL of water. The cPDA dispersion was preheated at 140 °C in an oil bath, and H$_2$PtCl$_6$ (aq) solution was added dropwise to the dispersion. Subsequently, the pH of the mixture was adjusted to around 10–11 by adding a pre-made 5 M NaOH (aq) solution. The reaction mixture was stirred at 1600 rpm for ~4 h under reflux. The pH was monitored using a pH meter (Model EL20, Mettler Toledo, USA) during this period and maintained at the desired level (between 10 and 11) by adding the NaOH (aq) solution if required. The level of the solution in the round bottom flask was maintained by the addition of a 20% (v/v) water/EG mixture. After the completion of the reaction, the sample was filtered and washed with DI water, ethanol, and acetone, followed by vacuum drying at 60 °C. Next, the dried sample was reduced under a 10% H$_2$ in N$_2$ atmosphere inside a furnace to ensure complete reduction of deposited Pt. Prior to H$_2$ treatment, the furnace was purged with N$_2$. Thereafter, the gas was changed to 10% H$_2$ in N$_2$, and the temperature was raised to 300 °C at 5 °C min$^{-1}$, held for 3 h, and afterwards cooled to room temperature under an N$_2$ atmosphere.

### PDA, cPDA, and Pt/cPDA characterization

The size and the morphology of PDA and cPDA nanospheres were examined by using Hitachi H-600 (Japan) Scanning Electron Microscopy (SEM), Tecnai F20 200 kV (USA) Transmission Electron Microscope (TEM) and Talos F200X Scanning transmission electron microscope (USA) equipped with a Super-X four silicon drift detectors of energy dispersive spectrometry (Super-X SSD EDS, EDAX). The TEM thin sections (~100 nm) were cut by Leica UCT ultramicrotome setup (Germany) equipped with an Ultra 45°DiATOME knife (USA) from a block prepared by embedding a small piece of the catalyst-coated

membrane (CCM) in a 1:1 mixture of trimethylolpropane triglycidyl ether resin (Sigma-Aldrich, USA) and 4,4'-Methylenebis (2-methylcyclohexylamine, Sigma-Aldrich, USA) hardener, polymerized overnight at 60 °C. The sections were situated onto multiple 200 mesh Cu/Pd grids. Laboratory-based X-ray photoelectron spectroscopy (XPS) measurements were carried out at room temperature in an ultrahigh vacuum (UHV, $3 \times 10^{-8}$ Torr) setup using a monochromatized Al Kα (1486.6 eV) excitation and a hemispherical analyzer (Kratos Axis Ultra DLD, UK).

### Electrochemical characterization in liquid electrolyte
Electrochemical characterization of the Pt/cPDA catalyst and commercial TKK 10% Pt/C catalyst was performed via a rotating disk electrode (RDE) test. Extensive care was taken to clean the glassware and work with high purity electrolyte as described in the following subsection.

### Rotating disk electrode test
**Glassware and component cleaning.** The electrochemical cell glassware (Pine Instrument, USA) and the components were soaked in concentrated sulfuric acid (Millipore Sigma, USA) overnight, followed by a 5–6 times repeat sequence of boiling with DI water followed by DI water replacement after each boil. This procedure is followed after each experiment to eliminate the trace amount of impurities. Before the electrochemical testing, the electrochemical cell and the components are rinsed 2–3 times with freshly prepared 0.1 M HClO₄ (diluted from 70% Veritas Doubly Distilled GFS Chemicals, USA). The glassy carbon (5 mm diameter, Pine Instrument) tips are polished before each experiment using 0.05 μm alumina slurry (Pine Instrument, USA), followed by rinsing and subsequent sonication for 3–5 min using DI water. The glassy carbon (GC) tip was dried using nitrogen gas before the catalyst coating.

**Ink formulation and coating.** Two different fabrication techniques were applied for the two different catalyst films—(i) ionomer-free and (ii) ionomer-based catalysts. For ionomer-free catalyst film preparation, the stationary air-dry technique was followed, whereas the rotational air-dry approach was adopted for the ionomer-based catalyst film.

The ionomer-free ink was prepared by adding 9.6 mg of catalysts in a 20 vol% IPA in a water mixture. The resulting ink was sonicated in an ice water bath sonicator for ~20 min to obtain a good dispersion slurry (10 μl aliquot, ~10 $\mu g_{Pt}$ cm⁻²). For ionomer-based catalyst, 9.6 mg of catalyst was dispersed in the required amount of 20 vol% IPA in a water mixture to which 39.5 μl of 5 wt% Nafion was further added to achieve an overall I/C mass ratio of 0.2 such that 10 μl aliquot would yield ~10 $\mu g_{Pt}$ cm⁻² in the coating. The ink was sonicated with an ice water bath sonicator for ~20 min. After sonication, the ink was deposited on the GC tip mounted on the inverted rotator shaft (Pine Instrument, USA) for the ionomer-based catalyst. It was rotated at 500 rpm for 15–20 min until the film dried. An aliquot of ink was deposited onto the GC disk and dried under an air atmosphere for the Nafion-free catalyst. The sample with partial or non-uniform coating assessed by visual inspection was rejected.

**Electrochemical characterization in liquid electrolyte.** All the electrochemical measurements were carried out using a conventional three-electrode setup consisting of a catalyst film coated on GC as a working electrode, Pt gauze as a counter electrode and a reversible hydrogen electrode (RHE) as a reference electrode.

The electrochemical characterization followed a sequence of experiments comprising catalyst conditioning, ECSA determination, and ORR activity measurement:

1. The conditioning of the catalyst is performed by cycling the electrode potential from 0.025 V–1.2 V at 500 mV s⁻¹ for 50–100 cycles under 1600 rpm in N₂-saturated 0.1 M HClO₄ until repeatable cyclic voltammograms were obtained. The voltage has been restricted to 1.2 V to minimize carbon corrosion. The conditioning of the catalyst is essential to obtain high ORR activity.

2. To evaluate the electrochemical active surface area (ECSA), the $H_{UPD}$ charge was obtained from hydrogen adsorption observed between -0.05 V and -0.4 V in the third cycle of the CV measured in 0.025 V–1 V at 20 mV s⁻¹ with no electrode rotation and the ECSA was estimated using 210 μC cm⁻²$_{Pt}$.

3. In addition to the $H_{UPD}$, the ECSA of the catalyst was also measured using the CO-stripping protocol; the 50 mol% CO in N₂ is bubbled into the electrolyte for 30 min. After bubbling, the working electrode was held at 0.08 V for 30 min; then, the electrolyte was purged with N₂ for 30 min. Two consecutive cyclic voltammograms were recorded from 0.05 to 1.1 V at a scan rate of 20 mV s⁻¹ and held at the final potential for 2 min to electrochemically strip all available CO molecules adsorbed on the Pt surface. The CO stripping charge was obtained from the difference in the charge between the two cyclic voltammograms. The ECSA was determined from the CO stripping charge by using the specific charge of 420 μC cm⁻²$_{Pt}$.

4. ORR I-V curve was measured during an anodic sweep from −0.01 to 1 V using a scan rate of 20 mV s⁻¹ at different rotation speeds (400–1600 rpm) to evaluate the activity and the kinetics of the catalyst in O₂ saturated 0.1 M HClO₄. The obtained LSV is corrected to baseline voltammetry in N₂ saturated condition, iR correction based on the uncompensated ohmic electrolyte measured via high-frequency AC impedance in N₂ saturated 0.1 M HClO₄, and correction for low local atmospheric pressure (88–90 kPa in Calgary, Canada). The data were corrected to 100 kPa based on the reaction order concerning O₂ of 0.85. The electrochemical data were collected using SP-200 potentiostat (Biologic, France). This study referred to all the electrochemical potentials represented as the RHE reference electrode. The equation used to calculate the ECSA was described briefly in Supplementary Information.

### Fabrication of membrane electrode assembly (MEA)
In this study, the decal transfer method was used in order to prepare an MEA. First, the desired amount of IPA to achieve a solid to liquid ratio (S/L) of 0.18 (20% water/IPA) and 20 wt% Nafion ionomer dispersion (EW 1100, Ion Power Inc.) was added to achieve an overall I/C mass ratio of 0.8 with catalyst (Pt/cPDA) in a 20 mL glass vial and dispersed ultrasonically using an ultrasonic bath for 20 minutes; both procedures were performed under the ice to avoid Pt degradation. For the commercial TKK 10% Pt/C catalyst, a similar ink preparation recipe was used. Here, an ionomer to carbon ratio (I/C) of 0.8, a solid to liquid ratio (S/L) of 0.18 and a solvent mixture of 20% water/IPA were used to prepare the catalyst ink for the commercial TKK 10% Pt/C catalyst. Then, to further homogenize, the inks (both Pt/cPDA and TKK Pt/C) were magnetically stirred in a 20 mL glass vial containing 5 g of 5 mm diameter ZrO₂ beads for 24 h at room temperature (23 °C) prior to its use. The catalyst layers were then coated onto a 75 μm thick PTFE substrate (McMaster-Carr, 8569K75) using an automatic film coater (MSK AFA-II, MTI Corporation, USA) at a speed of 10 mm s⁻¹. During the coating, a wet film thickness of 100 μm was set. Subsequently, the catalyst-coated decals were Air dried for 1 h followed by drying under vacuum for 12 h at 80 °C to evaporate residual solvents. CCMs with an active area of 1 cm² (sub-framed and controlled by Kapton window) were prepared by hot-pressing catalyst coated decal against 25 μm thick Nafion 211 (NRE-211, Fuel cell store, USA) membrane at 150 °C and 2 MPa pressure for 3 minutes with an applied force of 0.12 kN cm⁻². Pt loading in the prepared CCMs was calculated gravimetrically by weighting the decals before and after the catalyst layer transfer. The thickness of Pt/cPDA and TKK Pt/C CLs was determined by a

micrometer (Marathon digital micrometer, fisher scientific, USA) $9 \pm 1\,\mu m$ and $10 \pm 1\,\mu m$, respectively.

After hot-pressing, a cell was assembled by sandwiching hot-pressed CCM between two gas diffusion layers with a microporous layer (~230–240 $\mu m$, Toray, TGP-H-060; Fuel Cell Store, USA) guided by 175 $\mu m$ thick PTFE gaskets to ensure around 25% compression of GDL. GDL dimensions were 1.5 cm × 1.5 cm on both cathode and anode sides. The area of the GDL was designed to be larger than the active area of the CCM to ensure independent control of compression of the catalyst layer and the GDL and also to avoid edge degradation. A torque of 30 in-lb was applied during the cell assembly in three steps (10-20-30 in-lb). Fuel cell hardware consisting of 50 cm² flow field with a serpentine channel (16 cm² channel area) was used (purchased from fuel cell technologies, USA).

## Fuel cell testing

Fuel cell testing with in situ high-frequency resistance (HFR, for iR correction) measurement at each voltage were performed using a Biologic SP-200 potentiostat and a commercial 100 W, G20 Greenlight Innovation test station (Greenlight Innovation corp., Canada) in a high differential cell. The following sequence of events were followed and are described in Fig S10.

**Diagnostics.** First, a series of primary diagnostic tests ($H_{ad}$ CV, LSV and EIS) was performed to determine the electrochemical surface area (ECSA), $H_2$ crossover and series resistance or high-frequency resistance (HFR) of the assembled cell at 70 °C, 100% RH, 140 $kPa_{abs}$ in $H_2/N_2$ (0.1/0.2 NLPM). The reactants, i.e. compressed Air (99.999%), carbon monoxide (99.5%), hydrogen, oxygen (99.999%), and nitrogen (99.999%), were obtained from Air Liquide, Canada. The anode (hydrogen electrode) served as the reference (RHE) and counter electrode (CE), while the cathode was the working electrode (WE). The impedance spectra were obtained at 0.4 V by sweeping frequencies in the range of 1 MHz to 1 Hz with an amplitude of 10 mV. The anode and working electrode (cathode) were fed with different humidified hydrogen and nitrogen at the rate of 0.1 NLPM and 0.2 NLPM, respectively.

**Activation Protocol.** All cells were conditioned prior to testing to activate the MEA, hydrate the ionic network and remove possible contamination. The entire conditioning protocol is summarized in Table S5.

**$H_2$ pumping.** First, an $H_2$ pumping procedure was performed while the cell (both anode and cathode), anode humidifier and cathode humidifier temperature was set at 30 °C, 45 °C and 45 °C, respectively. $H_2$ pumping was performed for 30 min by applying a current of 50–200 mA cm⁻² while the resulting voltage was mildly negative. Then, the cell was purged with high flow $N_2$.

**Flooding.** The cell was supplied with over-humidified gas $H_2/N_2$ (0.1/0.2 NLPM) for around 8–12 h by setting the cell and humidifier temperature (both anode and cathode) at 70 °C and 80 °C, respectively to hydrate the membrane and ionomer network. This step resulted in a 10–20 mΩ·cm² reduction in the series resistance. Subsequently, the cell was purged again with a high flow of $N_2$ for 20 min to remove excess water, and the humidifier temperature was decreased to 70 °C.

**0.6 V hold.** Once the temperature is equilibrated, humidified $H_2$ and Air were introduced to the anode and cathode, respectively, and the back pressure was set at 200 $kPa_g$ for both sides. When an open circuit voltage (OCV) of ~1.0 V or steady state was reached, a constant voltage hold of 0.6 V was applied to the cell and current was drawn from the cell for ~12 h until the current stabilizes (±5 mA cm⁻²). A typical current profile is provided in Fig. S11.

**Potential cycling.** The cell was purged again with a high flow of $N_2$ gas to remove the excess water. Finally, the cathode potential was cycled between OCV to 0.6 V by holding for 5 min at each potential at 70 °C, 100% RH and 50 $kPa_g$ pressure in $H_2/Air$ (0.3/0.5 NLPM); this was repeated 5 times. Finally, the cell was purged with a high flow rate of $N_2$ for 20 min after the conditioning to remove any excess moisture trapped in the cell.

## Polarization curve

All polarization curves were determined in potentiostatic mode. The tests were recorded at desired operating conditions with $H_2$ and Air or $O_2$ (0.3/0.5 NLPM) at 140 $kPa_{abs}$ pressure. Both cathode and anode reactant gases were maintained at the same pressure during testing. Polarization plots were obtained from open circuit voltage (OCV) to around 0.1 V (until limiting current) potentiostatically by holding the voltage constant at each voltage for 3 min (steps of ~0.1 V after 0.9 V), and the resulting equilibrated current values were averaged over the final 30 s. In the case of pure $O_2$, polarization plots were obtained potentiostatically by recoding the I-V plot for kinetic parameter determination (as explained in SI, from OCV to around 0.8 V), followed by recording current from 0.8 V to around 0.1 V.

## Accelerated Stress Test (AST) protocol

The square-wave catalyst AST protocol was used in this work to study the durability of Pt catalyst, with the only exception that the temperature was 70 °C instead of 80 °C. It consisted of potential cycling between 0.6 V and 0.95 V by holding the voltage at each potential for 3 s for 30,000 cycles. It was conducted in $H_2$(anode)/$N_2$(cathode) at 0.2/0.2 NLPM at 70 °C, 100% RH and atmospheric pressure.

The detailed experimental method of $N_2$ sorption isotherm, Platinum and Ionomer content determination by TGA, Platinum content determination by TEM image analysis, equation used to estimate the average inter-particle platinum distance, ECSA determination by both $H_{UPD}$ and CO-stripping, determination of kinetic parameter and limiting current study to determine the $O_2$ transport resistance are explained in SI.

## Data availability

Relevant data supporting the key findings of this study are available within the article and the Supplementary Information file. All raw data generated during the current study are available from the corresponding authors upon request.

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

## Acknowledgements

K.K. acknowledges funding from the Natural Sciences and Engineering Research Council of Canada (NSERC) Discovery Grant (RGPIN/04856-2017), NSERC Alliance - Alberta Innovates Advance Program (ALLRP/561086-2020 and 202102615). K.K. acknowledges the infrastructure grant from Canadian Foundation for Innovation (CFI) Project 31936 and Alberta Innovates Research Capacity Program Project RCP-14-27-SEG. V.O.K. and M.N.I. acknowledge the University of Calgary Eyes High Postdoctoral and Doctoral Fellowships, respectively. J.J. and A.P.S. acknowledge: the University of Connecticut start-up funds for funding this work; The UConn Thermo Fisher Scientific Center for Advanced Microscopy and Materials Analysis (CAMMA) for equipment use; Marcia Reid from Canadian Center for Electron Microscopy at McMaster University for TEM sample preparation.

## Author contributions

K.K., M.N.I., and V.O.K. conceptualized the catalyst support design. V.O.K. and M.N.I. synthesized and characterized the PDA and cPDA nanoparticles. M.N.I. carried out platinum deposition, MEA fabrication, and fuel cell testing studies. M.N.I. and V.O.K. conducted TEM of Pt/cPDA nanoparticles, and M.N.I. and K.K. analyzed the results. M.N.I. and K.K. analyzed TGA results. A.B.M.B. synthesized an additional batch of Pt/cPDA nanoparticles, performed all the RDE tests, and carried out some MEA testing. M.N.I. and K.K. analyzed the fuel cell data; A.B.M.B. and K.K. analyzed the RDE data. A.P.S. and J.J. performed high-resolution STEM-EDS of catalyst layer and analyzed the STEM-EDS results. All authors contributed to the writing of the manuscript.

## Competing interests

The authors declare no competing interests.
