## [Peer review file · Nature Communications]

REVIEWER COMMENTS

Reviewer #1 (Remarks to the Author):

The work submitted by Muhammad Naoshad Islam et al, is of high importance to the fuel cell community as it highlights the importance of the carbon support, and more specifically the effect on the meso/macro-porous structure, in order to be able to improve the mass activity and the mass transport resistance for low Pt loaded MEAs in a fuel cell. This work tries to show for the first time the importance of the mesoporous/macroporous structure of the catalyst layer, as opposed to previously published work so far that have only focused on the carbon microstructure. The novelty of this work certainly merits a publication in this journal, but not to its present form. Overall the paper is well written, but it needs to address some serious gaps in the research and repeat experiments using a proper testing protocol. In the current state even though the idea and the novelty of the research is appreciated, based on the way experiments were conducted or the lack of in certain parts of the paper, it needs serious alterations and repetitions of experiments in order to be ready for publication and prove the claims the authors are making.

So I recommend Major Revision and only if the below major comments are properly addressed to go ahead for publication in this journal.

Major comments that need to be addressed before considering for publication:

- Unfair comparison with commercial catalyst (TKK) and MEA comprising of the TKK catalyst:

The authors are using a commercial catalyst from Tanaka K.K. (TKK) and they are several problems with the catalyst they use as reference:

The authors do not specify the type of catalyst besides its Pt loading. The carbon support –as also shown in the present under revision work –plays a critical role in mass activity and mass transport. So the authors should mention exact type of catalyst and its carbon support. Based on the CVs in the supporting information one can deduct –with a certain level of error of course- that the TKK catalyst is supported on a lower surface area carbon compared to the cPDA one- either a Vulcan but most probably a graphitized Ketjen based on the DLC value of the CV. My latter assumption is also supported based on the provided mass activity of the reference catalyst at 80 A/gPt – where a high surface area carbon would have been in the range of 250-300 A/gPt in a fuel cell. Nevertheless in both cases (Vulcan or graphitized Vulcan or graphitized Ketjen) - the Pt particles are located on the external carbon support and hence the authors would only see the effect of meso/macro porous structure of the catalyst in the catalyst layer, as no effect could originate from mass transport resistances from the Pt particles located in the micropores of the carbon as is usually seen in commercial high surface area carbons (Nagappan Ramaswamy et al 2020 J. Electrochem. Soc. 167 064515). So the reviewer understands and agrees with the use of such type of carbon supported catalyst in general for fair comparison as the aim of this work is to investigate the effect of the carbon size and hence the effect of the catalyst layer structure on the mass transport resistance.

However the difference of the Pt loading over the carbon support is a major concern that the authors overlooked: the Pt loading of ~46 wt% on the carbon support of TKK is extremely high as opposed to the claimed 8,5 wt% on the cPDA. The difference in the Pt loading affect dramatically the inter-particle distance and hence the local mass transport resistance as has been thoroughly shown in numerous publication over the past decade. So comparing the cPDA catalyst superiority over inter-Particle distances in the manuscript is completely unfair comparison as this alone would affect local mass transport resistance and hence performance (especially at high current densities) , in addition to any changes in activity . All recent studies –including the ones heavily sighted in the paper- when comparing different carbon supports for fair comparison all have similar Pt loading over the carbon support and same Pt loading in the MEA. This is largely done to ensure that no additional kinetic or ohmic or transport phenomena could –to the best of their ability- compromise the outcome of the study. So the authors should have done the same in order to prove their claims.

In addition, the choice of such a high loaded Pt catalyst -46wt%Pt/C from TKK – created an additional problem during the MEA manufacturing. Due to the high Pt loading of the catalyst, an electrode containing ~0,03 mgPt/cm² for direct comparison with the cPDA would have been impossible due to make due to the extremely thin electrode thickness. Hence the authors –if my assumption is correct- had to manufacture electrodes with almost a factor of 10 higher Pt loading in order to achieve a minimum of ~3µm of electrode thickness(assuming a stacking density of 28µm/mgC/cm²) to ensure proper electrode transfer and homogeneous electrode thickness. However that by itself creates an additional issue- that there is no direct performance comparison –especially in air – for demonstrate being reasonable doubt the advantage and superiority of cPDA catalyst.

The reviewer strongly recommended a repetition of the experiments using a proper reference catalyst in the same range of Pt loading over the carbon support (with a maximum Pt loading of 20wt%Pt/C), where the Pt particles are all located on the external surface only (like a Vulcan or a graphitized Ketjen supported catalyst) and with an MEA that has a similar Pt loading with the cPDA.

- This work does not take into consideration the Pt location or the pore opening of the carbon supports as it has been proven to affect significantly mass transport resistance and mass activity. A discussion and comparison on this effect with their work is needed in order not to mislead readers regarding the effect of the carbon support on fuel cell performance. In addition the authors failed to mention or demonstrate the Pt location with respect to their carbon support. This is very critical as in the manuscript the authors are comparing Pt accessibilities.
- The Pt accessibility or dry proton accessibility was not measured properly. The authors used CVs in H₂/N₂ vs RH and used the Hupd region for the determination, were it has been proven since 2011 that it has a high level of uncertainty as Hupd features are highly affected by the RH. Hence only CO stripping is used to accurately draw conclusions regarding Pt accessibility vs RH[examples of publications that proves the validity and superiority of CO stripping: Journal of The Electrochemical Society, 165 (3) F173-F180 (2018), Journal of The Electrochemical Society, 158 (5) B467-B475 (2011)].

The above is highly probable to be the origin of the discrepancy of the Pt utilization (or dry proton accessibility) of the commercial 46wt%Pt/C. Even though it is not mentioned-based on the provided evidence CV double layer- if my hypothesis is correct and they used a Vulcan or a graphitized Ketjen then the Pt accessibility should be much higher than the one they are claiming. [Journal of The Electrochemical Society, 2020 167 06451 and Journal of The Electrochemical Society, 158 (5) B467-B475

(2011)]. Another possibility for the low (please refer to the next comment) is the improper catalyst layer manufacturing recipe that could result in poor ionomer distribution over the reference catalyst. Which is also evident from the polarization curves but will be discussed further down.

Nevertheless, repetition of the experiments with CO stripping is needed to draw a proper conclusion regarding ionomer distribution and Pt accessibility.

- The authors used the same catalyst layer recipe for both catalysts. It is widely known that Pt loading over the carbon, the type of carbon and the carbon surface groups on the carbon supports as well as the type of ionomer used plays a critical role in the ionomer distribution and require unique catalyst ink recipes. The authors used for the two very different catalyst types but use the same ink recipe. Even though the authors demonstrated beyond reasonable doubt that the cPDA catalysts exhibit excellent ionomer distribution (TEM) using this particular ink recipe, one has to wonder regarding the reference catalyst. And if one takes into consideration the discrepancy regarding the Pt accessibility (see above comment) it is rightfully so that one would assume that the catalyst layer was not properly manufactured and hence a direct fair comparison cannot be made. As it is known that the ink recipe can affect the ECSA, performance, mass activity and mass transport (Journal of The Electrochemical Society, 165 (14) F1254-F1263 (2018)).

Further more there are evidences that support that the ionomer distribution in catalyst layer using the TKK catalyst was not very good if one takes a close look at the 100%RH polarization curve. The polarization curve exhibits already mass transport resistance even in Pure O₂. The latter is a clear indication that the catalyst layer was not properly made-especially considering the Pt loading that it is not that small. The polarization curve in air exhibits such massive mass transport resistance that it is simply impossible due to the Pt loading of this MEA (0,26mgPt/cm²), unless the ionomer distribution was really inhomogeneous causing blockage of pores and flooding of the catalyst layer. Under the authors experimental conditions and Pt loading they authors should have a minimum current density at 0,6V of 1,8 A/cm² or 6,9 A/mgPt after HFR correction if the cathodic catalyst layer was properly manufactured. If the authors had taken the polarization curves at different RH for the reference catalyst as well this would have been evident. So any comparison is unfortunately in my opinion not valid. Unless the authors used a high surface area Ketjen black where most Pt particles are located inside the carbon support-hence this could explain the higher mass transport losses at high current densities. But the latter is contradicting the DLC data and the mass activity that they have provided.

The authors have to provide evidences that the ionomer distribution in the catalyst layer of the reference was the best possible –such that the Pt accessibility of the reference (based on the type of catalyst used) is in agreement with published work based on their Pt location to ensure fair comparison of the presented data and also polarization curves with no significant mass transport resistance at a Pt loading of 0,26mgPt/cm² especially under such high flows for a 1cm² MEA that the experiment were taking place (by taking consideration the flows mentioned in figure S4).

- The authors mentioned that the experiments were repeated twice for each type of experiments. However error bars only appear in tables and not on the performance curves. Error bars should be added in the x-axis (as it is normalized by Pt loading of the electrode) and y axis direction of the polarization curves.

- The experimental section does not meet the necessary transparency requirements. The experimental section lacks sufficient information for testing conditions and protocols in order to ensure that the results can be reproduced by others and well understood. In addition it is widely known that the experimental conditions affects the outcome of the tests therefore are needed for complete evaluation of the validity of the presented data. To name a few:

- o Materials section: no mention the supplier and exact precursor used

- o Fuel Cell testing:

- ☐ The conditions of CV for the evaluation of ECSA are not mentioned at all (Cell temperature, RH, backpressure etc)

- ☐ What are the conditions of the polarizations curves in terms of flows? There is no record in the experimental section. Also where the polarization curve obtained from OCV to low voltages or vis versa? As it plays a critical role in the performance as specified in the DOE and EU harmonized fuel cell testing protocol.

- ☐ Proper activation protocol was not used as per DOE or EU Harmonized protocols and also details on how the cell was activated is very vague. It is widely known that Activation protocol can affect the performance of the MEA and hence lead to misleading results. A proper justification is needed why the authors did not use standard activation protocols for their study. Especially since they use Low Pt loaded electrodes the proper activation and recovery of the Pt surface is of paramount importance.

- o The fuel cell testing hardware is not mentioned and it is known to affect the performance, assembly of the cell ,compression of the GDLs and in particular mass transport resistance evaluation during the limiting current (Baker et al, Journal of Electrochemical Society)etc.

- o What kind of fuel cell test bench was used (commercial or inhouse made) Please specify.

- o Synthesis of PDA section: the authors do not mention the device they used as this can influence reproducibility for the readers.

Recommendation: As the authors have sighted extensively in their work another Nature publication by Ott et al, we would recommend to use the same transparency in experimental section.

- Pt determination of the catalyst is not properly determined. Even though the authors claimed that the acquired values from ICP-MS where significantly lower –especially when comparing the commercial catalyst and therefore did not trust the outcome, the manner of which they evaluated the Pt loading has extreme error and assumptions. Electrochemical active area -even in RDE- is never the same as the ECSA derived by TEM particle size. It is always smaller.

It is widely known that the ECSA in RDE depends on the film quality, the type of ionomer used, how much ionomer has been used in the ink etc. It is also known that when using Nafion or any other type of ionomer in RDE the ECSA and Activity of the catalyst significantly decrease as a result of SO_3 poisoning originating from the ionomer. So since the authors used Nafion in their RDE (RDE section of experimental) how can they claim that this is the highest possible obtained ECSA? The authors should have then used alternative way to determine Pt loading, like using a TGA- which has been proven to be very reliable in Pt determination by other groups. The authors need to confirm the Pt loading with an alternative ex-situ characterization technique, as the one currently used has a lot of assumptions. The wrong Pt loading on the carbon can alter the reported mass activity, roughness factor, ECSA in the MEA etc. In addition since all the MEA data and performance enhancement using this new catalyst is shown via polarization curves that are being normalized for the electrode Pt loading it is essential to accurately determine the Pt loading.

- Corrections of HFR of all the performance curves is wrong at the low RH. As mentioned in the experimental section(Supporting information) for the HFR determination only 1 value was measured at 0,4V. Hence this would mean that the entire polarization curve was corrected using this one value. Even though the MEA is 1cm^2 and thus to obtain the polarization curves in stoichiometric mode is very challenging to say the least -- it is obvious that the flows used were much higher than the one would use under standard stoichiometric mode ($\text{H}_2=1,5$ and $\text{Air}=2,0$ and $\text{O}_2=9,5$). Therefore the HFR would change significantly with respect to the current density of the cell due to the water production -- however this depends on the total flow rate and the partial pressure of the produced water. So to conclude at 100% RH the HFR is not expected to change at all due to the excess of water over the different current densities so the error of correcting set polarization curve is minor. But that is not the case for low RH obtained polarization curves. The HFR should have been taken at each voltage during the polarization curve acquisition and not separately. Therefore the HFR correction of all $\text{RH} < 80\% \text{RH}$ are not reliable- unless the authors can prove that the HFR did not change during the polarization curve acquisition under differential flows. Where the latter would not be possible especially for the $30\% \text{RH}$ and $45\% \text{RH}$: It is clear from Figure S4 at $30-45\% \text{RH}$ at high current density that there has been an over-correction of the polarization curve --as the polarization curve is parallel to the x-axis. This is clearly due to the fact that at higher current densities due to higher water production the HFR goes down at those low RH conditions and hence when corrected for the value at 0,4V is over-corrects the curve. Please comment.

Minor comment in Main manuscript:

- It is generally known that when referring for a publication as a reference in a figure or in a text the first author name is used and not the last author's name in order to avoid confusion as to per which reference is used for which data. Please correct throughout the manuscript.

Example  Figure 4 please correct.

Example  Line 276 : change author from Orfanidi to Ott et al. and from Gasteiger to Harzer et al.

- The axis of the polarization curves thought out the manuscript are wrong: when one corrects a polarization curve of the H₂ crossover current, the measured or applied current density is then corrected by adding the H₂-crossover current to the respective current. So the x-axis should be (measured current + H₂ crossover current)/mgPt. And the H₂ crossover wording should be removed from the y-axis.

- There are a lot of typos throughout the manuscript that needs to be corrected.

- Reference list format does not use a consistent nomenclature. Please make sure all references have the same format when listed.

- Line 212-215 : the fact that the lowering of RH results in increase of mass activity is a direct indication of the local flooding within the catalyst layer. This is highly probable to be due to the hydrophilic nature of the carbon support .So the authors should also take that into consideration during their conclusion.

- Section of Degradation of Pt

- o Figure 5c contains data from Reference 16 for comparison. The data acquired from reference 16 were obtained at different cell temperature (80°C with 100%RH under 100kPaabs, out pressure of both anode and cathode)- which is known to influence degradation kinetics. So one has to wonder if this comparison is valid to show their catalyst superiority. Please comment.

- o Line 276-277 : The slight increase of the activity reported by Reference 16 and 46 only occurs during the first 100 cycles, thereafter activity is decreasing. So the authors should be careful when using this statement as it might be misleading as they only present BOL, 3000 and 5000 cycles. Please rephrase to avoid misconception or add in your graph the 100 cycles activity for comparison purposes.

- Mass transport resistance comparison: One has to be careful when comparing total mass transport resistances between different groups- as there are significant differences in the MEAs and testing hardware that affect the recorded resistances (GDL, flow field geometry, compression of GDL, ionomer type, I/C and effective ionomer film thickness, Pt location, type of carbon, flows of the reacting gasses etc). The author does not compare or even discuss these differences between each publication with his own experimental set up. In particular the EW of the ionomer is known to significantly affect the local mass transport resistance –as also recently published by the same group (Poojary et al. *Molecules* 2020, 25, 3387). The different works that the authors have sighted all use different ionomer from different suppliers with different EW and different I/C and hence have different effective ionomer film thickness over the Pt particles which are known to hinder mass transport resistance. Since the authors did not specify in the experimental what is the EW used in their study it is hard to draw a concrete conclusion. A thorough discussion over this matter is needed in order not to draw the wrong conclusions. Due to the limitations of the journal this discussion could also be added in the supporting information. However it should be at least mentioned and referenced in the main text.

- Line 88 -91: The statement of this sentence is without evidences. There are no evidences that the N groups of the aforementioned publications do not distribute homogeneously N groups over the carbon support. On the contrary-especially for Ref 16, due to its high Pt accessibility almost reaching 100% actually shows the opposite that Ionomer is homogeneously distribute and hence one could assume that the N groups are also. Please clarify.

- The additional film uploaded avi from the Authors did not work.

- Figure 4f the graphs is problematic:

o The figure caption is wrong. Based on the absolute values of the kW/gPt the obtained values were in air and not in pure O₂ as stated. Please correct

Reviewer #2 (Remarks to the Author):

Review comment on "Pt nanoparticles on novel N-containing carbon 1 support ..." by Islam, et. al

This paper describes a comprehensive investigation on a low loading platinum catalyst for ORR and fuel cell application. The authors provided extensive study including catalyst structural characterization, RDE/fuel cell tests and mass transfer modeling. The effort and information gathered are highly commendable and information could be useful for the researchers in the field of fuel cell research. This reviewer, however, is not convinced that the manuscript is publishable at Nature Comm in its current form due to the following major deficiencies:

1. Although the Authors reported the PEMFC data, which is very important to this reviewer's opinion in any fuel cell manuscript, the data were not properly represented. First of all, only iR corrected fuel cell performances were reported for both H₂/O₂ and H₂/air cells. This is not acceptable. Particularly, it does not make much sense to report iR corrected data for H₂/air cell. Uncorrected data must be reported since they represent how the cell will perform in real application.
2. Using Tanaka's 46.6 wt% Pt on vulcan carbon as a benchmark is a poor choice. It contains about 5x more Pt than that prepared by Authors, rendering it an inappropriate comparative sample for structural characterization. Case in point, Authors argued that their average inter-particle distance (AID) is significantly longer (22 nm) compared to that of Tanaka's (7nm) so that theirs would have the benefit to mitigate the 'territory effect'. Wouldn't high AID is expected when the loading is much lower. In fact, we do not even know what is the specific surface area that Authors used in their comparative sample. It was only loosely stated between 17 to 365 m²/g, a huge variation. Tanaka market Pt/C catalyst from 5% to 50%. Authors should selected a more propriety one, such as 10% for structure and activity comparisons.
3. A similar problem arose when Authors used Tanaka's 46.6 wt% Pt/C as comparative sample for fuel cell and RDE tests. It is more favorable for Authors' sample in term of mass activity when the benchmark catalyst has so much higher Pt loading. Again, Authors are suggested to use Tanaka catalyst with similar loading to retest RDE and fuel cell.
4. There are some issues regarding to fuel cell tests. First, the mass activity was derived at H₂/O₂ cell at 70 C and 140 kPa. The data were used to compare of mass activity to US DOE target, which should be tested at 80 C and 150 kPa. Why not just to test at 80 C then? The accelerated stress test was also

stopped at 5k cycles while DOE test protocol calls for 30k cycles. These missing data reduced the significance of the manuscript.

5. Ref 16 is reported by Strasser's group instead of Gasteiger group.

I recommend that these issues are addressed by Authors before resubmitting the manuscript to Nature Comm.

Reviewer #3 (Remarks to the Author):

The authors report very high ORR activities on a "novel" N-containing carbon support.

I cannot recommend publication as the activity data seem not reliable for a number of reasons.

- The measurements are not done according to state of the art. To compare the data to published ones - also the Tkk catalyst - the RDE measurements should be performed in perchloric not sulfuric acid. The latter is known to interfere with the ORR.

- the shown RDE curves look very strange, both for the tkk and the home-made catalyst. The diffusion limited currents do not overlap and no rotation dependent plots are shown also it is written that they were recorded. Also one should show a Tafel plot as the Tafel slope is very different from the tkk catalyst. Also the CV seems to have an issue with the iR drop.

- The Pt loading via digestion seems not to work according to the authors and they resort to indirect measures. Nevertheless very high accuracy is suggested, e.g. 8.42 wt.%

- the data in Figure 3 suggest ORR activity above the OCP for Pt, which is very unlikely.

Last but not least the N-modification of supports is not new and the term novel should be avoided in the title

The authors' responses are presented in green text while the reviewer's comments are in black text.

First of all, we would like to thank the reviewers for putting their valuable time and effort in reviewing the manuscript. We especially appreciate the reviewers for their many critically constructive comments. In reviewing the comments (especially related to baseline catalyst), it also became apparent to us that the key claims of our work was not articulated well.

Our key claims are: (1) highest ORR activity of pure Pt-based catalyst in a MEA under fuel cell conditions and not in a liquid electrolyte. This is to demonstrate that high ORR activity is observed in a functional device and not just under idealized conditions in a liquid electrolyte; (2) low local-to-Pt oxygen transport resistance (pressure-independent R_{O_2}) for the Pt/cPDA catalysts in CLs with ultra-low Pt loading.

It needs to be emphasized that neither of the two claims would be meaningful **if the comparison is restricted merely to in-house CL made with commercial catalyst** (10 wt% Pt/VC – TKK – updated in revised manuscript). Rather, the comparison has to be made with data in the literature, which we have tried to bring out more clearly in the revised manuscript.

Accordingly, the manuscript has been significantly reorganized with following sub-sections:

- **Catalyst support development and characterization:** This section focuses on *large size* of catalyst support particles before and after carbonization (215 nm and 135 nm); demonstration of uniformity of Nitrogen distribution from EDX/STEM; chemical nature of the N-groups from XPS.
- **Pt/CPDA catalyst characterization:** In this section, we have focused on – (i) attainment of well-spaced or *well-dispersed Pt particles* within a *desirable size range of 2-3 nm*; (ii) discussion on *whether the Pt particles are on the surface or within the micropores* – via high resolution TEM and CO stripping experiments insisted by one of the reviewers; (iii) *critically important data on Pt content* (wt% Pt on cPDA) of the catalyst (new data based using TGA). A direct measurement of Pt content was missing in the original submission.
- **Characterization of CLs made with Pt/cPDA catalysts:** In this section, we present the data in support of our two major claims: (1) Comparison of ORR mass activity of Pt/cPDA in MEA against other data in literature also in MEA; (2) Comparison of local oxygen transport resistance (R_{O_2} , p-independent) against other data in the literature. We also discuss the ionomer distribution because this issue was also brought up by the reviewers.

- **Performance and durability characterization:** The results in this section is not linked to any of our claims but we recognize the value of such measurements and, thank the reviewers for insisting we perform experiments with a relevant Pt/C catalyst; i.e. low Pt content catalyst and low Pt loading catalyst layer. We compare the performance and durability characteristics of two in-house made CLs – one with our new Pt/cPDA and another with commercial 10 wt% Pt on Pt/VC.

Reviewer #1 (Remarks to the Author):

The work submitted by Muhammad Naoshad Islam et al, is of high importance to the fuel cell community as it highlights the importance of the carbon support, and more specifically the effect on the meso/macro-porous structure, in order to be able to improve the mass activity and the mass transport resistance for low Pt loaded MEAs in a fuel cell. This work tries to shows for the first time the importance of the mesoporous/macroporous structure of the catalyst layer, as opposed to previously published work so far that have only focused on the carbon microstructure. The novelty of this work certainly merits a publication in this journal, but not to its present form. Overall the paper is well written, but it needs to address some serious gabs in the research and repeat experiments using a proper testing protocol. In the current state even though the idea and the novelty of the research is appreciated, based on the way experiments were conducted or the lack of in certain parts of thepaper , it needs serious alterations and repetitions of experiments in order to be ready for publication and prove the claims the authors are making .

So i recomend Major Revision and only if the below major comments are properly address to go ahead for publication in this journal.

Response: We thank the reviewer for appreciating the importance of our work, one aspect of which is macroporous structure of CL. We also appreciate the comment about the novelty of our work. We regret that we did not use a pertinent baseline catalyst when comparing cell performance, even if higher performance was not the primary claim of our work. We have performed new measurements with a low Pt content catalyst as reference catalyst. It took much longer time to complete the additional experiments, which also required significant physical modification to our lab (to accommodate a CO gas line) and delays associated with paperwork and implementation of this modification.

Major comments that needs to be addressed before considering for publication:

- Unfair comparison with commercial catalyst (TKK) and MEA comprising of the TKK catalyst:

The authors are using a commercial catalyst from Tanaka K.K. (TKK) and they are several problem with the catalyst they use as reference:

1. The authors do not specify the type of catalyst besides its Pt loading. The carbon support – as also shown in the present under revision work -plays a critical role in mass activity and mass transport. So the authors should mention exact type of catalyst and its carbon support. Based on the CVs in the supporting information one can deduct –with a certain level of error of course- that the TKK catalyst is supported on a lower surface area carbon compared to the cPDA one- either a Vulcan but most probably a graphitized Ketjen based on the DLC value

of the CV. My latter assumption is also supported based on the provided mass activity of the reference catalyst at 80 A/gPt – where a high surface area carbon would have been in the range of 250-300 A/gPt in a fuel cell. Nevertheless in both cases (Vulcan or graphitized Vulcan or graphitized Ketjen) - the Pt particles are located on the external carbon support and hence the authors would only see the effect of meso/macro porous structure of the catalyst in the catalyst layer, as no effect could originate from mass transport resistances from the Pt particles located in the micropores of the carbon as is usually seen in commercial high surface area carbons (Nagappan Ramaswamy et al 2020 J. Electrochem. Soc. 167 064515). So the reviewer understands and agrees with the use of such type of carbon supported catalyst in general for fair comparison as the aim of this work is to investigate the effect of the carbon size and hence the effect of the catalyst layer structure on the mass transport resistance.

Response: We understand and fully agree with the reviewer’s concerns. Insofar the comparison with other catalyst is concerned, we understand the concerns. Our thinking was that our claims for high ORR activity and low local oxygen transport resistance did not hinge on the comparison with the 46.6 wt% Pt/VC catalyst rather with other literature data. However, we fully realize the value of having a better comparator catalyst as emphasized by the reviewer.

We also agree with the reviewer that an important information about the nature of carbon support should not have been missed due to oversight. In the original submission, we had used TTK 46.6 wt% Pt catalyst on Vulcan carbon (TEC10V50E, lot #1013-0241). However, as per the reviewer’s suggestion, we prepared a new reference catalyst layer using TTK 10.2 wt% Pt on Vulcan carbon (TEC10V10E, lot # 1019-1681). Pictures of both catalyst bottles are attached below. We have added the details of the catalyst in the experimental section.

Indeed, as pointed out by the reviewer, the focus of our work is to examine the effect of catalyst layer macroporous structure on oxygen transport without any complication arising from Pt present in the micropores of the carbon support. Thus, in addition to the Pt content (8.5 wt% for Pt/cPDA vs 10.2 wt% for Pt/ VC TKK), the VC support is also a better baseline because most of Pt for Pt/cPDA catalyst is located on the external surface of the support and not inside the micropores as discussed in the section “**Pt/cPDA catalyst characterization**”. The location of Pt catalyst for the Pt/cPDA can be visualized from the 3D TEM video submitted and the TEM images of the CL with sectional cuts of the PDA support particles (see image from Figure 3d reproduced below).

Fig. 3d STEM images of microtomed Pt/cPDA CL in - (i) and (ii) dark field and (iii) bright field mode. The marked regions are the cross-section of Pt/cPDA catalysts highlighting that the Pt particles are located at the exterior of the cPDA support.

2. However the difference of the Pt loading over the carbon support is a major concern that the authors overlooked: the Pt loading of ~46 wt% on the carbon support of TKK is extremely high as opposed to the claimed 8,5 wt% on the cPDA. The difference in the Pt loading affect dramatically the inter-particle distance and hence the local mass transport resistance as has been thoroughly shown in numerous publication over the past decade. So comparing the cPDA catalyst superiority over inter-Particle distances in the manuscript is completely unfair comparison as this alone would affect local mass transport resistance and hence performance (especially at high current densities) , in addition to any changes in activity. All recent studies –including the ones heavily sighted in the paper- when comparing different carbon supports for fair comparison all have similar Pt loading over the carbon support and same Pt loading in the MEA. This is largely done to ensure that no additional kinetic or ohmic or transport phenomena could –to the best of their ability- compromise the outcome of the study. So the authors should have done the same in order to prove their claims.

Response: It was our thinking that we were highlighting two key properties of the catalyst layer made with the new carbon support: (1) high ORR mass activity, and (2) low local mass transport resistance. Our comparison for these properties were not with respect to the reference

catalyst (46.6 wt% Pt) rather with data from other groups available in recent literature - both for ORR mass activity and for local mass transport resistance. In the original submission, we had provided the reference catalyst only as a baseline case for overall FC characteristics in our hardware/system. However, we fully agree with the reviewer that a better reference catalyst would be a commercial low Pt-content (e.g. 10%wt low Pt/C) catalyst.

Accordingly, we have carried out new experiments with catalyst layer made with a new reference catalyst TKK (10 wt% Pt/C; Vulcan) catalyst of similar Pt content as the Pt/cPDA catalyst (8.5 wt% Pt). We have removed the results for the 46.6 wt% Pt catalyst.

3. In addition, the choice of such a high loaded Pt catalyst -46wt%Pt/C from TKK – created an additional problem during the MEA manufacturing. Due to the high Pt loading of the catalyst, an electrode containing $\sim 0,03$ mgPt/cm² for direct comparison with the cPDA would have been impossible due to make due to the extremely thin electrode thickness. Hence the authors –if my assumption is correct- had to manufacture electrodes with almost a factor of 10 higher Pt loading in order to achieve a minimum of $\sim 3\mu\text{m}$ of electrode thickness (assuming a stacking density of $28\mu\text{m}/\text{mgC}/\text{cm}^2$) to ensure proper electrode transfer and homogeneous electrode thickness. However that by itself creates an additional issue- that there is no direct performance comparison –especially in air – for demonstrate being reasonable doubt the advantage and superiority of cPDA catalyst. The reviewer strongly recommended a repetition of the experiments using a proper reference catalyst in the same range of Pt loading over the carbon support (with a maximum Pt loading of 20wt%Pt/C), where the Pt particles are all located on the external surface only (like a Vulcan or a graphitized Ketjen supported catalyst) and with an MEA that has a similar Pt loading with the cPDA.

Response: We have performed new set of experiments with low-Pt content reference catalyst (10 wt% Pt/C) with similar roughness factor ($\sim 40 \text{ cm}^2_{\text{Pt}}/\text{cm}^2_{\text{MEA}}$). The Pt loading of the cathode catalyst layer with reference catalyst is $0.058 \text{ mg}/\text{cm}^2$ (thickness $10 \pm 1 \mu\text{m}$) and that of catalyst layer Pt/cPDA is $0.034 \pm 2 \text{ mg}/\text{cm}^2$ (thickness $9 \pm 1 \mu\text{m}$, measured using micrometer). We have included a performance comparison in Air between commercial (TKK) and cPDA catalyst layer showed in Figure 5(b) (mass loading normalized) of the main manuscript and Figure S13 (b), HFR corrected and Figure S14 (b), uncorrected of the SI (MEA performance). For the reviewer's convenience, the above-mentioned performances are also included below:

Table. Summary of the key parameters of Pt/cPDA and TKK 10% Pt/C catalyst layers

	New reference catalyst	Pt/cPDA catalyst
Pt content	10 wt%	8.5 wt%
Catalyst Support	Vulcan carbon	cPDA
RF by CO _{ads} (cm ² _{Pt} /cm ² _{MEA})	40	41
MEA Pt loading (mg/cm ²)	0.058	0.034 ± 0.002
Thickness (microns)	10 ± 1	9 ± 1
i _m (mA/mg _{Pt}) at 0.9V (iR free); 100%RH, O ₂	103	638 ± 68

Figure. HFR-free performance comparison between Pt/cPDA and TKK10% Pt/C catalyst : (left) electrode area normalized current, (right) mass loading normalized current. Condition: 70 °C and 140 kPa_{abs} pressure in (0.3/0.5 NLPM flow). Voltage was corrected for iR (HFR) loss and current density was corrected by adding crossover current to the measured current.

Figure. Uncorrected performance comparison between Pt/cPDA and TKK10% Pt/C catalyst : (left) electrode area normalized current, (right) mass loading normalized current. Condition: 70 °C and 140 kPa_{abs} pressure in (0.3/0.5 NLPM flow).

4. This work does not take into consideration the Pt location or the pore opening of the carbon supports as it has been proven to affect significantly mass transport resistance and mass activity. A discussion and comparison on this effect with their work is needed in order not to mislead readers regarding the effect of the carbon support on fuel cell performance. In addition the authors failed to mention or demonstrate the Pt location with respect to their carbon support. This is very critical as in the manuscript the authors are comparing Pt accessibilities.

Response: We agree with the reviewer about the influence of Pt location on the support. In the original submission, we had included a video (as a SI) obtained from 3D TEM of our Pt/cPDA catalyst showing that most of the catalyst particles are located on the surface of the cPDA support. Perhaps we should have highlighted in the main body text that the 3D TEM indicates that a majority of Pt catalyst are present on the surface of the support.

We have reorganized the manuscript to discuss the Pt location issue specifically. The Pt location is now discussed in the section “**Pt/cPDA catalyst characterization**”. The location of Pt catalyst for the Pt/cPDA can be visualized from the **3D TEM video** submitted as well as the TEM images of the CL with sectional cuts of the PDA support particles (see image from Figure 3 reproduced below).

Fig. 3d STEM images of microtomed Pt/cPDA CL in - (i) and (ii) dark field and (iii) bright field mode. The marked regions are the cross-section of Pt/cPDA catalysts highlighting that the Pt particles are located at the exterior of the cPDA support.

5. The Pt accessibility or dry proton accessibility was not measured properly. The authors used CVs in H₂/N₂ vs RH and used the Hup_d region for the determination, where it has been proven since 2011 that it has a high level of uncertainty as Hup_d features are highly affected by the RH. Hence only CO stripping is used to accurately draw conclusions regarding Pt accessibility vs RH[examples of publications that proves the validity and superiority of CO stripping:

Journal of The Electrochemical Society, 165 (3) F173-F180 (2018), Journal of The Electrochemical Society, 158 (5) B467-B475 (2011)].

The above is highly probable to be the origin of the discrepancy of the Pt utilization (or dry proton accessibility) of the commercial 46wt%Pt/C. Even though it is not mentioned-based on the provided evidence CV double layer- if my hypothesis is correct and they used a Vulcan or a graphitized Ketjen then the Pt accessibility should be much higher than the one they are claiming. [Journal of The Electrochemical Society, 2020 167 06451 and Journal of The Electrochemical Society, 158 (5) B467-B475 (2011)]. Another possibility for the low (please refer to the next comment) is the improper catalyst layer manufacturing recipe that could result in poor ionomer distribution over the reference catalyst. Which is also evident form the polarization curves but will be discussed further down. Nevertheless, repetition of the experiments with CO stripping is needed to draw a proper conclusion regarding ionomer distribution and Pt accessibility.

Response: We have now performed the CO stripping experiments for ECSA determination. We had to make modification to the lab setup to accommodate for CO gas. This took a longer than expected time for approval and modification due to COVID restrictions imposed measures.

Figure S10 (reproduced below) shows the comparison of RH dependency of ECSA for reference catalyst (10 wt% Pt; TKK/Vulcan) and our Pt/cPDA catalyst determined by CO stripping method.

Fig. S10. (a) Roughness factor (RF, determined by CO stripping) comparison between Pt/cPDA and TKK 10% Pt/C catalyst as a function of RH, (b) dry proton (H^+) accessibility (estimated by ECSA CO stripping – open symbol, dashed line; CDL – filled symbol, solid line) comparison between Pt/cPDA and in-house TKK 10% Pt/C catalyst and literature Pt utilization

data of 10% Pt/V published by Padgett et. al. Condition: 70 °C, Nafion EW1100, I/C – 0.8 (this work), 80 °C, Nafion EW950, I/C – 0.95 (Padgett et. al.)

Further discussion and comparison have been included in Supplementary Information. Figure S10 (reproduced above) also shows the comparison with a literature data (Padgett et al, JES, 165 (3) F173-F180 (2018)) for the similar reference catalyst (10% Pt on Vulcan). Our catalyst layer with new reference catalyst shows similar utilization at 80% RH but 10% (relative) lower utilization at lower RH, which might be due to lower I/C ratio (0.8) compared to the above-mentioned work published by Padgett et al. (I/C-0.9). The published work by Padgett et al. indicated that almost 40% of the Pt particles are located in the carbon interior, thus the RH dependency.

The ratio of ECSA determined by CO stripping and by hydrogen underpotential deposition is also reported in Figure S11 (reproduced below). A comparison of this ratio with other reported literature data is also reported in Figure S11 and shown below (right side Figure below).

Fig. S11. (a) ECSA (CO stripping)/ECSA (H_{ad}) (ECSA_{CO}/ECSA_{HAD}) ratio comparison (a) between Pt/cPDA and in-house TTK 10% Pt/C catalyst as a function of RH, (b) between in house Pt/cPDA and TTK 10% Pt/C catalyst, and literature 10% Pt/V (Padgett et al.) and Pt/HSC catalysts (Garrick et. al.) at 100% RH. Condition: 70 °C, Nafion EW1100, I/C – 0.8 (this work), 80 °C, Nafion EW950, I/C – 0.95 (Padgett et al. and Garrick et. al.)

Padgett et al. , *Journal of The Electrochemical Society* **165**, F173-F180 (2018).
Garrick et. al., *Journal of The Electrochemical Society* **164**, F55 (2016).

6. The authors used the same catalyst layer recipe for both catalysts. It is widely known that Pt loading over the carbon, the type of carbon and the carbon surface groups on the carbon supports as well as the type of ionomer used plays a critical role in the ionomer distribution and require unique catalyst ink recipes. The authors used for the two very different catalyst types but use the same ink recipe. Even though the authors demonstrated beyond reasonable doubt that the cPDA catalysts exhibit excellent Ionomer distribution (TEM) using this particular ink recipe, one has to wonder regarding the reference catalyst. And if one takes into consideration the discrepancy regarding the Pt accessibility (see above comment) it is rightfully so that one would assume that the catalyst layer was not properly manufactured and hence a direct a fair comparison cannot be made. As it is known that the ink recipe can affect the ECSA, performance, mass activity and mass transport (Journal of The Electrochemical Society, 165 (14) F1254-F1263 (2018)). Further more there are evidences that support that the ionomer distribution in catalyst layer using the TTK catalyst was not very good if one takes a close look at the 100%RH polarization curve. The polarization curve exhibits already mass transport resistance even in Pure O₂. The latter is a clear indication that the catalyst layer was not properly made-especially considering the Pt loading that it is not that small. The polarization curve in air exhibits such massive mass transport resistance that it is simply impossible due to the Pt loading of this MEA (0,26mgPt/cm²), unless the ionomer distribution was really inhomogeneous causing blockage of pores and flooding of the catalyst layer. Under the authors experimental conditions and Pt loading they authors should have a minimum current density at 0,6V of 1,8 A/cm² or 6,9 A/mgPt after HFR correction if the cathodic catalyst layer was properly manufactured. If the authors had taken the polarization curves at different RH for the reference catalyst as well this would have been evident. So any comparison is unfortunately in my opinion not valid. Unless the authors used a high surface area Ketjen black where most Pt particles are located inside the carbon support-hence this could explain the higher mass transport losses at high current densities. But the latter is contradicting the DLC data and the mass activity that they have provided.

Response: The reviewer brings up a lot of valid points, some of which have already been answered in responses above. For example, as alluded earlier, it was our thinking that the comparison of the two key properties of Pt/cPDA based CLs are – (i) ORR activity in MEA (A/mgPt) and local-O₂-transport resistance (RO₂, pressure-independent) – are **against the literature reported data** and not with our reference catalyst. However, we understand the reviewer's skepticism regarding the quality of ink and resulting CL.

We have prepared the new catalyst layer for 10 wt% Pt/VC-TKK catalyst adopting an ink recipe similar to the one described in the following reference: JES, 164 (4) F418-F426 (2017). The key results for the new reference catalyst are summarized below:

- RH-dependent ECSA and C_{DL} (see Figure in the preceding response)
- i at 0.6V in Air, at 100% RH after HFR correction: 7.5 A/mg_{Pt} (Figure 5b, main manuscript)
- No sharp mass transport related drop is observed in O₂ (polarization curve below, Figure S13a)

Figure S13. (a) HFR-free performance comparison between Pt/cPDA and TTK10% Pt/C catalyst in H₂/O₂, Condition: 70 °C and 140 kPa_{abs} pressure in (0.3/0.5 NLPM flow). Voltage was corrected for iR (HFR) loss and current density was corrected for crossover loss.

7. The authors have to provide evidences that the ionomer distribution in the catalyst layer of the reference was the best possible –such that the Pt accessibility of the reference (based on the type of catalyst used) is in agreement with published work based on their Pt location to ensure fair comparison of the presented data and also polarization curves with no significant mass transport resistance at a Pt loading of 0,26mgPt/cm² especially under such high flows for a 1cm² MEA that the experiment were taking place (by taking consideration the flows mentioned in figure S4).

Response: We have addressed this concern in the preceding responses. The RH-dependent ECSA and C_{DL} as a measure of ionomer distribution for catalyst layer made with the new reference catalyst as well as repeat measurement for fresh CL made with Pt/cPDA catalyst have been discussed in the responses above and also in the manuscript. The polarization curve showing no dramatic mass transport loss (as was observed for CL with 46.6 wt% Pt/VC catalyst) is also shared above in response and included in the revised manuscript.

8. The authors mentioned that the experiments were repeated twice for each type of experiments. However error bars only appear in tables and not on the performance curves. Error bars should be added in the x-axis (as it is normalized by Pt loading of the electrode) and y axis direction of the polarization curves.

Response: The error bars have been added in the X axes of the polarization curves of Pt/cPDA CL since the polarization curves were recorded in a potentiostatic mode (Figure 4b-e, Figure 5a-b, SI Figure S12a (Pt/cPDA results), S13 - S17), activity plots (Figure 4c and 4d), Table S5, RH dependency C_{DL} and ECSA (Figure 4b), and Pressure independent resistance plot (Figure 4e).

The above-mentioned Figures are reproduced below from the manuscript and SI.

Fig. 4 (b) Effect of RH on ECSA (H_{ad}) and C_{DL} of Pt/cPDA CL, (c) Mass activity Tafel plot comparison between Pt/cPDA and TKK Pt/C catalyst in MEA condition*; the error bars for Pt/cPDA-MEA represent deviation from average measurements of three catalyst layers, (d) $i_{m,0.9V}$ comparison in both RDE and MEA condition between Pt/cPDA* and different Pt catalysts reported in literature, (e) comparison of pressure independent O_2 transport resistance between Pt/cPDA and other Pt based catalysts reported in the literature (values inside the bracket indicate Pt loadings in $mg_{Pt} cm^{-2}$), *Pt/cPDA MEA condition: 70°C, 100% RH (80% RH for O_2 transport resistance), 140 kPa_{abs} , H_2/O_2 for activity measurements. E was corrected for iR loss and current density was corrected by adding crossover current to the measured current.

Literature data recorded at 80 °C, 100% RH, 150 kPa_{abs}, H₂/O₂, ^ai_{m,0.9V} was assumed at 150 kPa_{abs}, operating pressure was not mentioned; ^bi_{m,0.9V} measured at 300 kPa_{abs}, recalculated at 150 kPa_{abs} following the method explained in ref¹

Fig. 5I (a) and (b) effect of RH and polarization curve comparison between Pt/cPDA and commercial Pt/C (TKK 10 wt% Pt) catalyst in H₂/O₂ (a) and H₂/Air (b), respectively[#], in MEA at 70 °C, 100% RH and 140 kPa pressure in. *ECSA* was measured using *CO* stripping method. [#]MEA condition: 70 °C, 100% RH; 140 kPa_{abs}, *E* was corrected for *iR* losses, current density is corrected by adding crossover current to the measured current.

Fig. S12. (a) Tafel plots of Pt/cPDA catalyst and Pt-free cPDA catalyst at varying RH, Condition: 70 °C and 140 kPa_{abs} pressure, in H₂/O₂ (0.3/0.5 NLPM); voltage was corrected for *iR* (HFR) loss and current density was corrected for crossover loss.

Table S5. Electrochemically determined geometric and kinetic parameters of Pt/cPDA and commercial Pt/C catalyst both in RDE and MEA condition.

Catalyst	Geometric Parameters			Kinetic Parameters		Reference
	ECSA ($\text{m}^2 \text{g}_{\text{Pt}}^{-1}$)	RF ($\text{cm}_{\text{Pt}}^2 \text{cm}_{\text{geo}}^{-2}$)	L_{Pt} ($\mu\text{g}_{\text{Pt}} \text{cm}_{\text{geo}}^{-2}$)	$i_{\text{s},0.9\text{V}}$ ($\text{mA cm}_{\text{Pt}}^{-2}$)	$i_{\text{m},0.9\text{V}}$ ($\text{mA mg}_{\text{Pt}}^{-1}$)	
Pt/cPDA (RDE)	106 (H_{ad})	–	~8	1.017	1073	This work
Pt/C TKK (10 wt% Pt) (MEA)	70 (CO)	40 (CO)	58	0.147	103	This work
Pt/cPDA (MEA)*	101 ± 5 (H_{ad})	35 ± 2 (H_{ad})	34 ± 2	0.632 ± 0.06	638 ± 68	This work
	113 (CO)	41 (CO)				
Pt/C TKK 46% (RDE- H_2SO_4)	99 ± 15	–	–	0.093 ± 0.008	92 ± 22	Meier et al.
Pt/C TKK 46% (RDE- HClO_4)		–	–	0.38 ± 0.06	370 ± 11	

ECSA = Pt electrochemical surface area, RF = roughness factor of MEA working electrode (WE), L_{Pt} = WE Pt loading, $i_{\text{s},0.9\text{V}}$ and $i_{\text{m},0.9\text{V}}$ Pt specific and mass activity calculated at 0.9 V versus RHE, respectively (RDE - measured at a scan rate of 20 mV s^{-1} in $0.5 \text{ M H}_2\text{SO}_4$ at $23 \text{ }^\circ\text{C}$ and ambient pressure ($\sim 90 \text{ kPa}$), mass activities estimated via calculation of kinetic current - i_k and normalization to L_{Pt} , Pt area-specific activity calculated using the ECSA)

MEA condition: 70°C , 100% RH, $140 \text{ kPa}_{\text{abs}}$, H_2/O_2 for activity and ECSA values were determined at $70 \text{ }^\circ\text{C}$, 100% RH, $140 \text{ kPa}_{\text{abs}}$, H_2/N_2 (for H_{ad}).

*the error represents the deviation from average value measured for three different MEAs.

Fig. S13. HFR-free performance comparison between Pt/cPDA and TKK10% Pt/C catalyst :

(a) in H_2/O_2 , (b) H_2/Air . Condition: 70°C and $140\text{ kPa}_{\text{abs}}$ pressure ($0.3/0.5\text{ NLPM}$ flow).

Voltage was corrected for iR (HFR) loss and current density was corrected for crossover loss.

Fig. S14. Uncorrected performance comparison between Pt/cPDA and TKK10% Pt/C catalyst

: (a) in H_2/O_2 , (b) H_2/Air . Condition: 70°C and $140\text{ kPa}_{\text{abs}}$ pressure in ($0.3/0.5\text{ NLPM}$ flow).

Fig. S15. Uncorrected mass loading normalized performance comparison between Pt/cPDA and TKK10% Pt/C catalyst : (a) in H_2/O_2 , (b) H_2/Air . Condition: 70°C and $140\text{ kPa}_{\text{abs}}$ pressure in (0.3/0.5 NLPM flow).

Fig. S16. (a) HFR values used to perform the iR corrections recorded during the polarization curves at different RH for Pt/cPDA and TKK Pt/C catalyst layer (100% RH): (a) in H_2/O_2 , (b) H_2/Air . Condition: 70°C and $140\text{ kPa}_{\text{abs}}$ pressure in (0.3/0.5 NLPM flow).

Fig. S17. Mass loading normalized polarization curve and specific power in (a) H_2/O_2 and (b) H_2/Air (0.3/0.5 NLPM) at 70°C , 100% RH and $140\text{ kPa}_{\text{abs}}$; Voltage was corrected for iR (HFR) loss and current density was corrected for crossover loss.

8. The experimental section does not meet the necessary transparency requirements. The experimental section lacks sufficient information for testing conditions and protocols in order to ensure that the results can be reproduced by others and well understood. In addition it is widely known that the experimental conditions affects the outcome of the tests therefore are needed for complete evaluation of the validity of the presented data. To name a few:

o Materials section: no mention the supplier and exact precursor used –

Response: We have added all the available details about the chemicals and suppliers in the SI.

o Fuel Cell testing:

♣ The conditions of CV for the evaluation of ECSA are not mentioned at all (Cell temperature, RH, backpressure etc.)

Response: We have added the conditions of CV for the ECSA evaluation in the SI: The H_{ad} CV was performed at 70 °C, 30-100% RH and 140 kPa in H_2/N_2 (0.1/0.2 NLPM).

♣ What are the conditions of the polarizations curves in terms of flows? There is no record in the experimental section. Also where the polarization curve obtained from OCV to low voltages or vis versa? As it pays a critical role in the performance as specified in the DOE and EU harmonized fuel cell testing protocol.

Response: The flows were mentioned in the polarization curves in SI. However, we have added the suggested information in all the polarization curves in main manuscript and in the experimental section ((0.3/0.5 NLPM)).

♣ Proper activation protocol was not used as per DOE or EU Harmonized protocols and also details on how the cell was activated is very vague. It is widely known that Activation protocol can affect the performance of the MEA and hence lead to misleading results. A proper justification is needed why the authors did not use standard activation protocols for their study. Especially since they use Low Pt loaded electrodes the proper activation and recovery of the Pt surface is of paramount importance.

Response: The conditioning protocol used in this study is a combination of USFCC, DOE and recent work from NREL (USA) group. The H_2 pumping, constant voltage hold at 0.6 V from USFCC and potential cycling conditioning protocol (OCV to 0.6 V) similar to DOE (OCV to 0.55 V) and Kabir et al. (OCV to 0.6V) [*ACS applied materials & interfaces* 11, 45016-45030 (2019)] were employed. As per the reviewer's recommendation a separate section on the details of the activation protocol has been added in the SI.

o The fuel cell testing hardware is not mentioned and it is known to affect the performance, assembly of the cell ,compression of the GDLs and in particular mass transport resistance evaluation during the limiting current (Baker et al, Journal of Electrochemical Society)etc.

Response: We agree with the reviewer’s inference, thus added the recommended details in the MEA fabrication section with a photo of the hardware (Figure S6).

Fig. S6. Optical image of the flow-field design and MEA geometry with respect to the flow field (left), and front and side view of the used hardware for cell assembly (middle and right).

o What kind of fuel cell test bench was used (commercial or inhouse made) Please specify.

Response: Commercial Greenlight Innovation test station (100 W, G20, Greenlight Innovation) was used. We have also added the information in the SI in Fuel cell testing subsection in the experimental section.

o Synthesis of PDA section: the authors do not mention the device they used as this can influence reproducibility for the readers.

Response: The PDA synthesis was carried out in a 500 mL glass beaker. We have added this information in the corresponding subsection of SI.

Recommendation: As the authors have sighted extensively in their work another Nature publication by Ott et al, we would recommend to use the same transparency in experimental section.

Response: We thank the reviewer for the recommendation. We have added as much experimental details as we could with the consideration that they should allow other researchers to follow the same protocol and obtain repeatable results.

- Pt determination of the catalyst is not properly determined. Even though the authors claimed that the acquired values from ICP-MS were significantly lower –especially when comparing the commercial catalyst and therefore did not trust the outcome, the manner of which they evaluated the Pt loading has extreme error and assumptions. Electrochemical active area -even in RDE- is never the same as the ECSA derived by TEM particle size. It is always smaller.

Response: We agree with the reviewer. We acknowledged the challenges and error from ICP-MS. Accordingly, we had previously assumed the conservative value of Pt loading or maximum theoretically possible (10 wt% Pt/C assuming all Pt were deposited). We have performed TGA analysis (Figure 3c, Figure S4 and Table S2 in SI) and obtained 8.5 wt% in line with TEM image analyses data. All pertinent data – mass activity, ECSA, polarization curve – have been updated to reflect the TGA obtained Pt loading.

It is widely known that the ECSA in RDE depends on the film quality, the type of ionomer used, how much Ionomer has been used in the ink etc. It is also known that when using Nafion or any other type of ionomer in RDE the ECSA and Activity of the catalyst significantly decrease as a result of –SO₃ poisoning originating from the ionomer. So since the authors used Nafion in their RDE (RDE section of experimental) how can they claim that this is the highest possible obtained ECSA?

Response: We agree with the reviewer about Nafion induced poisoning. It is also known that activity in H₂SO₄ is lower than that in HClO₄. Accordingly, if the mass activity of Nafion-free Pt/cPDA catalyst were to be obtained from RDE (in HClO₄ electrolyte) would be even higher than what we have reported for Pt/cPDA with Nafion in H₂SO₄ electrolyte.

Perhaps there is a misunderstanding about the claims of highest activity. We are claiming that the activity is highest reported for **pure Pt in a realistic situation of MEA** and not in liquid electrolyte RDE.

Furthermore, the ECSA obtained from TEM data analysis was considered as the maximum, and the Pt content estimation was based on that ECSA. Nevertheless, as mentioned in the previous response, we have now determined the Pt content by TGA and corrected for the mass specific activity, which makes it even higher

The authors should have then used alternative way to determine Pt loading, like using a TGA- which has been proven to be very reliable in Pt determination by other groups. The authors need to confirm the Pt loading with an alternative ex-situ characterization technique, as the one currently used has a lot of assumptions. The wrong Pt loading on the carbon can alter the reported mass activity, roughness factor, ECSA in the MEA etc. In addition since all the MEA data and performance enhancement using this new catalyst is shown via polarization curves

that are being normalized for the electrode Pt loading it is essential to accurately determine the Pt loading.

Response: We thank the reviewer for the recommendation. As mentioned above, we have performed TGA analysis and obtained 8.5 wt% in line with TEM image analyses data. All pertinent data – mass activity, ECSA, polarization curve – have been updated to reflect the TGA obtained Pt loading.

- Corrections of HFR of all the performance curves is wrong at the low RH. As mentioned in the experimental section (Supporting information) for the HFR determination only 1 value was measured at 0,4V. Hence this would mean that the entire polarization curve was corrected using this one value. Even though the MEA is 1cm² and thus to obtain the polarization curves in stoichiometric mode is very challenging to say the least – it is obvious that the flows used were much higher than the one would use under standard stoichiometric mode (H₂=1,5 and Air=2,0 and O₂=9,5). Therefore the HFR would change significantly with respect to the current density of the cell due to the water production –however this depends on the total flow rate and the partial pressure of the produced water. So to conclude at 100% RH the HFR is not expected to change at all due to the excess of water over the different current densities so the error of correcting set polarization curve is minor. But that is not the case for low RH obtained polarization curves. The HFR should have been taken at each voltage during the polarization curve acquisition and not separately. Therefore the HFR correction of all RH < 80%RH are not reliable- unless the authors can prove that the HFR did not change during the polarization curve acquisition under differential flows. Where the latter would not be possible especially for the 30%RH and 45%RH: It is clear from Figure S4 at 30 -45%RH at high current density that there has been an over-correction of the polarization curve –as the polarization curve is parallel to the x-axis. This is clearly due to the fact that at higher current densities due to higher water production the HFR goes down at those low RH conditions and hence when corrected for the value at 0,4V is over-corrects the curve. Please comment.

Response: We agree with the reviewer's comment that at low RH, HFR values were changed significantly, and the low RH performance may have been over corrected. However, we had measured HFR values at different current densities during measuring the polarization curves and used those for iR correction. We should have included the HFR data. Nevertheless, the HFR values used for iR correction are provided below and in SI Figure S16.

Fig. S16. (a) HFR values used to perform the iR corrections recorded during the polarization curves at different RH for Pt/cPDA and TKK Pt/C catalyst layer (100% RH): (a) in H_2/O_2 , (b) H_2/Air . Condition: 70°C and $140\text{ kPa}_{\text{abs}}$ pressure in (0.3/0.5 NLPM flow).

Minor comment in Main manuscript:

- It is generally known that when referring for a publication as a reference in a figure or in a text the first author name is used and not the last author's name in order to avoid confusion as to per which reference is used for which data. Please correct throughout the manuscript.

Example \diamond Figure 4 please correct.

Example \diamond Line 276 : change author from Orfanidi to Ott et al. and from et al.

Response: We have reviewed and revised the manuscript as per the suggestion of the reviewer and changed accordingly.

- The axis of the polarization curves thought out the manuscript are wrong: when one corrects a polarization curve of the H_2 crossover current, the measured or applied current density is then corrected by adding the H_2 -crossover current to the respective current. So the x-axis should be (measured current + H_2 crossover current)/mgPt. And the H_2 crossover wording should be removed from the y-axis.

Response: We have modified the axes in all the performance and activity plots numbers: (Figure 4c, Figure 5a, 5b, 5d, SI Figure S12, S13 - S15, S17)

- There are a lot of typos throughout the manuscript that needs to be corrected.

Response: We have reviewed the manuscript and made as many typographical corrections as we could.

- Reference list format does not use a consistent nomenclature. Please make sure all references have the same format when listed.

Response: We have made the formatting as consistent as we can. We also expect that IF the manuscript makes it to the production stage, further corrections may be needed.

- Line 212-215 : the fact that the lowering of RH results in increase of mass activity is a direct indication of the local flooding within the catalyst layer. This is highly probable to be due to the hydrophilic nature of the carbon support. So the authors should also take that into consideration during their conclusion.

Response: We agree with reviewer's hypothesis. However, we have removed the bar chart where mass activity (at 0.9V in a MEA) was reported at different RH. The results have been moved to supporting information as Tafel Plots (Figure S12a).

- Section of Degradation of Pt

- o Figure 5c contains data from Reference 16 for comparison. The data acquired from reference 16 were obtained at different cell temperature (80°C with 100%RH under 100kPa abs, out pressure of both anode and cathode)- which is known to influence degradation kinetics. So one has to wonder if this comparison is valid to show their catalyst superiority.

Response: Our primary claims are for high mass activity and low O₂ transport resistance but not superior Pt degradation. We apologize if our presentation led to an implication of superior degradation characteristics of the catalyst. We have removed the comparison with reference 16 (Ott et al.) and compare our results with the in-house TKK10% Pt/C catalyst at the same operating condition subjected to DOE recommended square wave AST protocol.

Please comment.

- o Line 276-277 : The slight increase of the activity reported by Reference 16 and 46 only occurs during the first 100 cycles, thereafter activity is decreasing. So the authors should be careful when using this statement as it might be misleading as they only present BOL, 3000 and 5000 cycles. Please rephrase to avoid misconception or add in your graph the 100 cycles activity for comparison purposes.

Response: We understand and agree with the reviewer's concerns. We have included new durability data for full 30k cycles. Consequently, we have modified the section in the main manuscript.

- Mass transport resistance comparison: One has to be careful when comparing total mass transport resistances between different groups- as there are significant differences in the MEAs and testing hardware that affect the recorded resistances (GDL, flow field geometry, compression of GDL, ionomer type, I/C and effective ionomer film thickness, Pt location, type of carbon, flows of the reacting gasses etc). The author does not compare or even discuss these differences between each publication with his own experimental set up. In particular the EW of the ionomer is known to significantly affect the local mass transport resistance –as also recently published by the same group (Poojary et al *Molecules* 2020, 25, 3387). The different works that the authors have sighted all use different ionomer from different suppliers with different EW and different I/C and hence have different effective ionomer film thickness over the Pt particles which are known to hinder mass transport resistance. Since the authors did not specify in the experimental what is the EW used in their study it is hard to draw a concrete conclusion. A thorough discussion over this matter is needed in order not to draw the wrong conclusions. Due to the limitations of the journal this discussion could also be added in the supporting information. However it should be at least mentioned and referenced in the main text.

Response: We completely agree with reviewer's comments. As recognized by the reviewer, we do not have much room in the main body of the paper to discuss this. Nevertheless, since the one of the main focuses of this work is the transport resistance originated from the CL, we removed the total transport resistance data and focused on the $R_{O_2}^{P,ind}$ transport resistance and local O₂-transport resistance, which are mainly dictated by the CL and its subcomponents properties. Also, we have included a discussion on that matter in the SI (page 31).

- Line 88 -91: The statement of this sentence is without evidences. There are no evidences that the N groups of the aforementioned publications do not distribute homogeneously N groups over the carbon support. On the contrary-especially for Ref 16, due to its high Pt accessibility almost reaching 100% actually shows the opposite that Ionomer is homogeneously distribute and hence one could assume that the N groups are also. Please clarify.

Response: This is a fair point about no evidence that N groups are not distributed homogeneously. We agree that there is no direct proof to refute the homogeneous distribution of N-group.

That inference was drawn based on the ECSA data (in RDE; left Figure below), TEM and those obtained in MEA (right Figure below) reported in Ott et al. We had concluded that not all catalysts are accessible. A comparison between the ECSA based on TEM, RDE and MEA is also shown in the bottom Figure. If one considers the $ECSA_{TEM}$ to be the maximum available surface area, The Pt/KB-600 catalyst layer showed a 50% Pt utilization at 100% RH in MEA. However, this could also arise from other factors including HUPD for RDE and CO-stripping for MEA or the use of perchloric acid for RDE and PFSA ionomer for MEA. Furthermore, no DCL data or direct imaging of ionomer in CL was given to gain further insight.

Regardless, we do not know if the ionomer coverage was 100% in ref 16 (Ott et al.) and this point does not add to our paper. Hence, we have removed that statement.

ECSA from RDE

ECSA determined from TEM, RDE and MEA

- The additional film uploaded avi from the Authors did not work.

Response: We will change the file format and re-upload the video file again.

- Figure 4f the graphs is problematic:

- o The figure caption is wrong. Based on the absolute values of the kW/gPt the obtained values were in air and not in pure O₂ as stated. Please correct

Response: We thank the reviewer for pointing out the typographical mistake. We have corrected it.

Reviewer #2 (Remarks to the Author):

Review comment on “Pt nanoparticles on novel N-containing carbon 1 support ...” by Islam, et. al

This paper describes a comprehensive investigation on a low loading platinum catalyst for ORR and fuel cell application. The authors provided extensive study including catalyst structural characterization, RDE/fuel cell tests and mass transfer modeling. The effort and information gathered are highly commendable and information could be useful for the researchers in the field of fuel cell research.

Response: We appreciate the reviewer’s positive comment.

This reviewer, however, is not convinced that the manuscript is publishable at Nature Comm in its current form due to the following major deficiencies:

1. Although the Authors reported the PEMFC data, which is very important to this reviewer’s opinion in any fuel cell manuscript, the data were not properly represented. First of all, only iR corrected fuel cell performances were reported for both H_2/O_2 and H_2/air cells. This is not acceptable. Particularly, it does not make much sense to report iR corrected data for H_2/air cell. Uncorrected data must be reported since they represent how the cell will perform in real application.

Response: We understand the reviewer’s concern. All the DOE targets were based on the iR corrected values; thus, we had not provided uncorrected data in the original submission. However, as per the reviewer’s recommendation, we have included the uncorrected performances, both cell and mass loading normalized, in both O_2 and Air. We have also provided the data below for reviewer’s convenience (Figure S14 and S15 in SI).

Fig. S14. Uncorrected performance comparison between Pt/cPDA and TKK10% Pt/C catalyst : (a) in H₂/O₂, (b) H₂/Air. Condition: 70 °C and 140 kPa_{abs} pressure in (0.3/0.5 NLPM flow).

Fig. S15. Uncorrected mass loading normalized performance comparison between Pt/cPDA and TKK10% Pt/C catalyst : (a) in H₂/O₂, (b) H₂/Air. Condition: 70 °C and 140 kPa_{abs} pressure in (0.3/0.5 NLPM flow).

2. Using Tanaka’s 46.6 wt% Pt on vulcan carbon as a benchmark is a poor choice. It contains about 5x more Pt than that prepared by Authors, rendering it an inappropriate comparative sample for structural characterization. Case in point, Authors argued that their average inter-particle distance (AID) is significantly longer (22 nm) compared to that of Tanaka’s (7nm) so that theirs would have the benefit to mitigate the ‘territory effect’. Wouldn’t high AID is expected when the loading is much lower. In fact, we do not even know what is the specific surface area that Authors used in their comparative sample. It was only loosely stated between 17 to 365 m²/g, a huge variation. Tanaka market Pt/C catalyst from 5% to 50%. Authors should selected a more propriety one, such as 10% for structure and activity comparisons.

Response: We understand and fully agree with the reviewer’s concerns pertinent to AID. It was our thinking that our claims for high ORR activity and low local oxygen transport resistance did not hinge on the comparison with the 46.6 wt% Pt/VC catalyst rather with other literature data. However, we fully realize the value of having a better reference catalyst.

As per the reviewer's suggestion, we prepared a new catalyst layer with using a new reference catalyst TKK 10.2 wt% Pt on Vulcan carbon (TEC10V10E, lot # 1019-1681). Regarding the surface area of the cPDA carbon support, we meant to write in the main manuscript at line #85 that the surface area was increased from 17 m²/g to 365 m²/g upon carbonization (from PDA to carbonized PDA). Also, the microstructural properties of cPDA carbon support determined from N₂ adsorption isotherm is presented in Table S1 and compared with Vulcan carbon (provided below) from a reference [Soboleva *et al.*, *ACS applied materials & interfaces* **2**, 375-384 (2010)]

Table S1. Microstructural properties of cPDA carbon support and Vulcan carbon determined from N₂ adsorption isotherm.

Sample	SA _{BET, total} (m ² g _{carbon} ⁻¹)	SA _{> 2 nm} (m ² g _{carbon} ⁻¹)	SA _{< 2 nm} (m ² g _{carbon} ⁻¹)	V _{pore, total} (cm ³ g _{carbon} ⁻¹)	V _{pore, < 2nm} (cm ³ g _{carbon} ⁻¹)	Ref
cPDA	365.4	62.9	302.5	0.51	0.16	This work
Vulcan carbon	227.7	114.7	113.0	0.4	0.06	Soboleva et al.

3. A similar problem arose when Authors used Tanaka's 46.6 wt% Pt/C as comparative sample for fuel cell and RDE tests. It is more favorable for Authors' sample in term of mass activity when the benchmark catalyst has so much higher Pt loading. Again, Authors are suggested to use Tanaka catalyst with similar loading to retest RDE and fuel cell.

Response: As explained in the previous comment, we have prepared a new reference catalyst layer with using TKK 10.2 wt% Pt on Vulcan carbon (TEC10V10E, lot # 1019-1681), and replaced all the fuel cell data. However, we were unable to perform the RDE tests as we do not have access to a RDE facility at this time.

4. There are some issues regarding to fuel cell tests. First, the mass activity was derived at H₂/O₂ cell at 70 C and 140 kPa. The data were used to compare of mass activity to US DOE target, which should be tested at 80 C and 150 kPa. Why not just to test at 80 C then? The accelerated stress test was also stopped at 5k cycles while DOE test protocol calls for 30k cycles. These missing data reduced the significance of the manuscript.

Response: We understand the reviewer's concern regarding the activity comparison at the similar condition. We have recorded the activity data at both 70 °C and 80 °C; however, the

difference is insignificant. The comparison between 70 °C and 80 °C Tafel plots are included in the SI in Figure S12b (given below).

Fig. S12. (b) Tafel plots of Pt/cPDA catalyst at varying temperature (70 °C and 80 °C), Condition: 140 kPa_{abs} pressure, in H₂/O₂ (0.3/0.5 NLPM); voltage was corrected for iR (HFR) loss and current density was corrected for crossover loss.

As per the reviewer’s recommendation, we have also performed 30k catalyst durability AST cycles following DOE recommended square wave (0.6 – 0.95 V) protocol for both Pt/cPDA and TKK 10% Pt/C catalyst. We have updated the data in Figure 5c of main manuscript and also included here:

Fig. 5l (c) ECSA loss during square wave AST degradation cycles (AST protocol: DOE square wave, 0.6-0.95 V, potential changed at $\sim 700 \text{ mV s}^{-1}$), at 70 °C, 100% RH and atmospheric pressure in H₂/N₂ (0.2/0.2 NLPM).

5. Ref 16 is reported by Strasser's group instead of Gasteiger group.

Response: We thank the reviewer for pointing it out. We have corrected it.

I recommend that these issues are addressed by Authors before resubmitting the manuscript to Nature Comm.

Reviewer #3 (Remarks to the Author):

The authors report very high ORR activities on a "novel" N-containing carbon support. I cannot recommend publication as the activity data seem not reliable for a number of reasons.

- The measurements are not done according to state of the art. To compare the data to published ones - also the Tkk catalyst - the RDE measurements should be performed in perchloric not sulfuric acid. The latter is known to interfere with the ORR.

Response: We would like to emphasize and clarify that the claim of high ORR activity in the original as well as in the revised manuscript is for ORR activity in a MEA and not in electrolyte. We have intentionally stayed away from staking claims about high activity in liquid electrolyte (either perchloric acid or sulphuric acid). This is because as stated in the Introduction of the original as well as the revised manuscript, huge difference between high activity in liquid electrolyte and activity in MEA (e.g. nanoframe PtNi catalysts) have been observed. This has led to ongoing debate regarding: (a) the origin of such large differences between activity in liquid electrolyte and in MEA (b) the *practical value of reporting such high activity in liquid electrolyte* when the metric that matters for fuel cell application is the ORR activity in MEA. This debate was also highlighted few years ago in a Science article (see Figure below) [Stephens *et al.*, *Science*, 2016, 354 (6318), 1378-1379].

From model studies to real devices

Novel Pt-based catalysts, including those reported by Bu *et al.* and Li *et al.* in this issue, perform better than commercial pure Pt nanoparticles in model liquid half cells (15). The next challenge is to translate the full extent of this superior performance to fuel cells.

Mass activity (A/mg Pt)

● Liquid half cell ● Fuel cell

Accordingly, we have focused on ORR activity in MEA and our claim of high activity as well as comparison with literature data from other groups is kept to ORR activity in MEA. The activity data in liquid electrolyte was/is provided as a **supplementary result**.

Despite our best efforts, we were unable to get access to RDE facility, which belongs to a colleague (who retired last year) – so we have been unable to perform the experiments in HClO₄. We completely agree with the reviewer’s comment that sulfuric acid may interfere with the ORR activity. On the other hand, the ORR activity in perchloric acid is also not representative of activity in a MEA. Poisoning by sulphonic groups of perfluoro-sulphonic acid ionomer has been a topic of longstanding debate and there has also been proposition of side-chain ether group interacting with the Pt surface.

Many groups have reported ORR activity in MEA as an in-operando characteristic of the catalyst. US DOE also has set activity target for new catalysts in MEA (discussed in manuscript). Accordingly, consistent with the focus of our work, we have shared the comparison of ORR activity in MEA (Figure 4d) The activity data in liquid electrolyte (H₂SO₄) is provided as supplementary results and also compared with the literature data recorded in H₂SO₄ (Table S5, provided below).

Table S5. Electrochemically determined geometric and kinetic parameters of Pt/cPDA and commercial Pt/C catalyst both in RDE and MEA condition.

Catalyst	Geometric Parameters			Kinetic Parameters		Reference
	ECSA (m ² g _{Pt} ⁻¹)	RF (cm _{Pt} ² cm _{geo} ⁻²)	L_{Pt} (μ g _{Pt} cm _{geo} ⁻²)	$i_{s,0.9V}$ (mA cm _{Pt} ⁻²)	$i_{m,0.9V}$ (mA mg _{Pt} ⁻¹)	
Pt/cPDA (RDE)	106 (H _{ad})	–	~8	1.017	1073	This work
Pt/C TTK (10 wt% Pt) (MEA)	70 (CO)	40 (CO)	58	0.147	103	This work
Pt/cPDA (MEA)*	101 ± 5 (H _{ad})	35 ± 2 (H _{ad})	34 ± 2	0.632 ± 0.06	638 ± 68	This work
	113 (CO)	41 (CO)				
Pt/C TTK 46% (RDE-H ₂ SO ₄)	99 ± 15	–	–	0.093 ± 0.008	92 ± 22	Meier et al.
Pt/C TTK 46% (RDE-HClO ₄)		–	–	0.38 ± 0.06	370 ± 11	

ECSA = Pt electrochemical surface area, RF = roughness factor of MEA working electrode (WE), L_{Pt} = WE Pt loading, $i_{s,0.9V}$ and $i_{m,0.9V}$ Pt specific and mass activity calculated at 0.9 V versus RHE, respectively (RDE -

measured at a scan rate of 20 mV s^{-1} in $0.5 \text{ M H}_2\text{SO}_4$ at $23 \text{ }^\circ\text{C}$, ambient pressure (90 kPa), mass activities estimated via calculation of kinetic current - i_k and normalization to L_{Pt} , Pt area-specific activity calculated using the ECSA MEA condition: 70°C , $100\% \text{ RH}$, $140 \text{ kPa}_{\text{abs}}$, H_2/O_2 for activity and ECSA values were determined at $70 \text{ }^\circ\text{C}$, $100\% \text{ RH}$, $140 \text{ kPa}_{\text{abs}}$, H_2/N_2 (for H_{ad}).

*the error represents the deviation from average value measured for three different MEAs.

Meier JC, et al., *Beilstein journal of nanotechnology* 5, 44-67 (2014).

Comment: - the shown RDE curves look very strange, both for the tkk and the home-made catalyst. The diffusion limited currents do not overlap and no rotation dependent plots are shown also it is written that they were recorded.

Response: We share below the rotation-dependent plots and corresponding Koutecky-Levich plot for Pt/cPDA catalyst (provided in SI; Figure S5b and S5c). The Koutecky-Levich plot for Pt/cPDA catalyst showed the expected linearity, which were considered as an auxiliary indicator that the data were reasonable.

As stated above, since we had deemed the ORR activity in MEA to be the main information and activity in liquid electrolyte to be supplementary result, we had not included rotation-dependent plot

Fig. S5. (b) RDE voltammogram under O_2 atmosphere at varying rotation speed, and (c) corresponding Koutecky-Levich plot of Pt/cPDA. Sweep rate - 10 mV/s . Electrolyte - $0.5 \text{ M H}_2\text{SO}_4$ (RE - RHE; CE - Platinum).

We had also noticed the small but significant difference in the limiting current for the two catalysts. We also noted other data in literature (from prominent groups; e.g. Adzic, Stamenkovic) who reported data showing small deviations from the theoretically predicted value (e.g. 6 mA/cm^2 at 1600 rpm) and attributed the differences to difference in the morphology of the catalyst layer deposited on the RDE. Some of the data from notable groups

are provided below, where deviation from the theoretical value can be noticed. Accordingly, we deemed our results to be reasonable.

Figure. Comparison of the activities for the ORR of Pt/C, core-shell Pt_{ML}/Pd/C, and Pt/PtPb and Pt/PdFe in 0.1 M HClO₄; 1600 rpm; 10 mV/s. Ref: Ghosh et al., *J. Am. Chem. Soc.* 2010, 132, 3, 906–907 (*Radoslav Adzic group*)

Figure. cyclic voltammograms in O₂-saturated 0.1 M HClO₄ at 50 mVs⁻¹ and 1,600 rpm. Ref: Hernandez et al. Nature Chem 6, 732–738 (2014).

Figure. Characteristic CV profiles for the bare and modified Pt(111) surfaces and their respective polarization curves for ORR. (b) their corresponding ORR polarization curves (positive-going potential sweep). The curves were measured in (b) O₂-saturated 0.05 M sulfuric acid solutions (~293 K), at a sweep rate of 50 mV s⁻¹. The ORR polarization curves were recorded at 1600 rpm using the RDE configuration. The bare Pt(111) ORR polarization curve recorded in a 0.1 M perchloric acid solution (gray dashed curve) is shown for comparison in (b). A dotted vertical line present in (b) serves as an eye guide to spot the difference in the ORR activities ($E = 0.9$ V) between the bare and modified Pt(111) surfaces.

Zorko et al. ACS Appl. Mater. Interfaces 2021, 13, 2, 3369–3376. (Stamenkovic group)

Comment- Also one should show a Tafel plot as the Tafel slope is very different from the tkk catalyst. Also the CV seems to have an issue with the iR drop.

Response: In the present form of the manuscript, we have removed the previous 46.6% Pt data from the manuscript since we have changed the reference catalyst to TKK 10% Pt/C. Nonetheless, we report the Tafel plot for review purpose only in this document.

Figure. Tafel plot comparison between Pt/cPDA (TS ~ 68 mv/dec) and TKK 46% (TS ~ 69 mv/dec) catalyst in RDE condition.

Regarding the CV, we agree with the reviewer that there was a small iR drop in the cyclic voltammetry data of both TKK Pt/C and Pt/cPDA catalyst, which mostly affected the anodic sweep. The implication of this iR drop would reflect in ECSA only. As mentioned in the supplementary information, we have calculated the ECSA by integrating the H_{ad} area under the cathodic sweep. While calculating the ECSA, integrated area affected by the iR drop was also taken into consideration. In addition, in the revised version, we have also performed the ECSA calculation using CO stripping method. The calculated ECSA in liquid electrolyte (via H_{ad}) and MEA (both H_{ad} and CO stripping) are in good agreement (please see Table S5 included above). Also, the estimated ECSA based on TEM data is within reasonable agreement with the $\text{ECSA}_{\text{liquid}}$ and ECSA_{MEA} .

Comment: The Pt loading via digestion seems not to work according to the authors and they resort to indirect measures. Nevertheless very high accuracy is suggested, e.g. 8.42 wt.%

Response: We have now measured the Pt content on Pt/cPDA using TGA and the Pt content was found to be 8.5 wt% in line with our previous TEM image analysis estimation. Details in the SI. (Figure 3c, SI Figure S4, Table S2 and S3). Also provided below.

Figure 3. (c) TGA thermogram of Pt/cPDA and TKK Pt/C (10.2% and 19.8%) catalysts for the determination of Pt content (details in supplementary information)

Fig. S4. TGA thermogram of Pt/cPDA, TKK Pt/C 10%, TKK Pt/C 20%, Pt/cPDA CL and TKK Pt/C 10% CL for the determination of Pt and ionomer content (Pt/cPDA CL).

Table S2. Summary of the TGA determined and theoretical Pt content in Pt/cPDA CL, TKK 10% Pt/C (Vulcan carbon) CL.

Sample	TGA-Pt/C (%)	Pt/C _{Th} (%)*
Pt/cPDA CL	7.8	-
Pt/cPDA	8.5	
Pt/C TKK 10%	9.5	10.2
Pt/C TKK 10% CL	10.4	
Pt/C TKK 20%	19.5	19.8

*The Pt/C_{Th} for the commercial catalysts are the manufacturer provided values.

Table S3. Estimated Pt content on cPDA determined from TEM image analysis.

ECSA Calculation		Pt content (wt%); Pt/(Pt+C)		
Method	ECSA _{TEM} (m ² /g _{Pt})	Pt/cPDA _{liquid} (%)	Pt/cPDA _{MEA} (%)	TGA Pt/cPDA (%)
$\Sigma SA_i / \Sigma m_i$	108	8.4	8.5	8.5%
$\Sigma(SA_i / m_i)$	123	7.3	7.4	
(SA_{avg} / m_{avg})	117	7.7	7.8	

*Assuming all Pt precursor was loaded on cPDA support

Comment: the data in Figure 3 suggest ORR activity above the OCP for Pt, which is very unlikely.

Response: The activity at OCV is 0 for both catalyst that is why they are not included in the data in Figure 3. The obtained OCV for TKK 10% Pt/C and Pt/cPDA at 70C, 100% RH and 140 kPa pressure in pure O₂ was 0.96 and 1.01 V, respectively.

Comment: Last but not least the N-modification of supports is not new and the term **novel** should be avoided in the title

Response: We would like to state that we are neutral about keeping or dropping the term novel.

We understand the reviewer's concern regarding the Nitrogen functionalization. However, we have to disagree with the reviewer here. Unlike other reported work, where a carbon black support is taken and its surface is functionalized with nitrogen containing groups, we synthesize the catalyst support particle using an N-containing precursor.

Our work is not about **modification** of the support rather a self-assembly and polymerization of N-containing precursor into a supramolecular structure, which is then carbonized to create a support. The nitrogen is an intrinsic part of the molecules that make up the whole support and it is not just grafting of N-containing moiety on the surface of a support. This allows more uniformly distributed Nitrogen functional groups as also shown by the TEM mapping in Figure 2e of main manuscript.

- 1 Neyerlin, K., Gu, W., Jorne, J. & Gasteiger, H. A. Determination of catalyst unique parameters for the oxygen reduction reaction in a PEMFC. *Journal of The Electrochemical Society* **153**, A1955-A1963 (2006).

Reviewers' comments:

Reviewer #1 (Remarks to the Author):

I greatly appreciate all the efforts the authors have done in addressing the most of the reviewers concerns. And the manuscript is by far in a better state than before. However they are still some major issues that have to be addressed properly before publishing in Nature.

1. In Supporting information you mention that :“Once dried, CCM with a geometrical area of 1 cm² was prepared by hot-pressing the catalyst against 25 μm thick Nafion 211 (NRE-211, Fuel cell store, USA) membrane at 150 °C and 2 MPa pressure for 3 minutes with an applied force of 0.12 kN cm⁻². The geometrical area of 1 cm² was controlled by kapton window. ”

The CCM cannot be 1 cm² in area and at the same time be controlled by an active window of 1cm² via Kapton. Typically in subframed CCMs , the CCM is bigger than the active window- as also clearly illustrated in the picture you provided- the CCM (black) is bigger than 1cm² in the image (as the flow field is clearly ~7.1 x ~7.1 cm in order to be able to give a 50cm² geometric cell area). Please either correct the above statement or rephrase as it is misleading.

2. Please make sure for future reference how you conduct your experiments- you also have convection in your experimental set up based on the way your MEA is placed on the flow field. I would recommend for future experiments not to place the CCM in the location where there is crossflow along the length of the channels. For example if you look at Ott et al or Baker et al or generally General Motors experiments/papers, all had the MEA placed in the same flow direction to avoid under the rib convection. As this significantly influences your measurements for the mass transport determination and fair comparison with literature. For example Orfanidi et al and Harzer et al used a 5cm² flow field where also in their case they must have had contribution from under the rib convection. In your case the way the CCM was placed you clearly have contribution of convection during the limiting current measurement which could result in lower mass transport values. Be careful when comparing to other studies as your Mass transport values will be lower compared to other studies due to under the rib convection- if they did not have in their experiments.

3. As the authors clearly have acknowledged in the Limiting current discussion section in SI, that the mass transport resistance can be affected by the ionomer film thickness over the Pt particles. The authors also provide the BET data with regards to the micro and mesoporous structure of the two different carbons. . It is widely known in literature that the effective ionomer film thickness will be lower for a carbon support with a lot of micropores (<2nm). Please refer to Liu et al (Liu et al 2011 J. Electrochem. Soc. 158 B614) for the details on how to calculate the effective I/C or ionomer film

thickness over the external carbon surface (> 2nm) for a better comparison and more useful discussion. However, they fail to discuss the dramatic difference in the effective ionomer film thickness between the two samples (Pt/V and Pt/cPDA) and how this difference is actually in favour of their findings. The authors need to discuss the above in the manuscript.

For example, the authors write Line 232: "As hypothesized, the large cPDA particles result in a dramatic decrease in the Knudsen resistance, which is estimated to be around 0.04 s cm⁻¹ compared to the values of 0.14 s cm⁻¹ and 0.08 s cm⁻¹ reported for Pt/KB and Pt/KB-N CLs16

It would be beneficial for the readers if the authors could also write after the above sentence that the cPDA had a thicker ionomer film (and give a value) covering the external cPDA particles compared to the KB and N-KB of another published study or even, and hence it is evidently clear that the difference originates from the mesoporous structure. I think a statement like this would greatly enhance the authors' claims and make it easier for other readers to make a fair comparison with literature presented data.

4. Line 176 to Line 180: "A dry proton accessibility of 0.8 indicates high ionomer coverage. The cPDA particles possess hydrophilic characteristics as noted by spreading of water on a layer of particles during attempts to measure contact angle. The high ionomer coverage coupled with the hydrophilic nature of support, attributed to the N-functional groups^{34,179}, should result in a well-hydrated Pt/ionomer interface even in dry conditions, facilitating the proton accessibility to most of the Pt particles."

The authors make bold statements "0.8 indicates high ionomer coverage" when they have no strong evidence to support that, in my opinion. First of all, even though they have repeated the experiments 3 times—again, there are no error bars in either the Pt/V or Pt/cPDA to actually demonstrate whether the 10% difference in the dry proton accessibility is actually a meaningful difference or simply an experimental discrepancy. In addition, they compare in Figure S10 their data with that of Padgett et al. Padgett et al. also used 10wt% Pt/V from TKK and got almost the same dry proton accessibility with Pt/cPDA. Hence, how can the authors claim that the N groups have contributed to this 10% difference, since Padgett had an N-free carbon and yet showed almost the same values of dry proton accessibility? If the authors had shown ~90% dry proton accessibility for the cPDA, I would then agree with their statement—as it would be above the value reported by Padgett. It might simply be that the authors' 10wt%Pt/V catalyst layer was not as well manufactured as that of Padgett and hence this is why they observed the 10% difference in dry proton accessibility.

I am not questioning the fact that the N is homogeneously distributed or the N and ionomer interaction; I am simply questioning the boldness of the statements without having hard evidence to support it. I completely disagree with the way the above-mentioned lines in the manuscript are written as they are

misleading and not scientifically accurate. Please also when re-writing this section state that the 80% of cPDA shows the same value as reported by Padget.

In addition, the presence of N groups does not automatically mean that the ionomer will be more homogeneously distributed. Recent publications from the group of Athanasov recently presented XPS study that showed that this is not the case and that the different N types could also change the ionomer orientation over the carbon support and hence affect the proton conductivity and hydrophilicity of the CL. In order to be able to claim this you need to provide proton conductivity measurements of the catalyst layer and compare it with the 10wt%Pt/V, which was not done in this study. If I were you I would write the above like this-as it is scientifically more accurate based on the presented data :

Line 176 to Line 180: "The dry proton accessibility value of the Pt/cPDA might indicate higher ionomer coverage. Nevertheless it has to be stated that a 80% dry proton accessibility has also been reported by Padget et al for the commercial 10wt%Pt/C ,same catalyst used in this study (70%). The cPDA particles possess hydrophilic characteristic as noted by spreading of water on a layer of particle during attempts to measure contact angle. The slightly higher ionomer coverage of cPDA coupled with hydrophilic nature of support, could be attributed to the N-functional groups³⁴¹⁷⁹ ,and is expected to result in well hydrated Pt/ionomer interface even in dry conditions facilitating the proton accessibility to most of the Pt particles."

5. The discussion section of the main manuscript does not provide any link between the presented data and their hypothesis in the text nor do they provide a plausible explanation for the origin of the higher mass activity of the cPDA (clarification are given below)- as the neglect a critical part of ionomer poisoning effect. To be more precise : What is the hypothesis of why the mass activity is higher despite the fact that all the Pt particles are located on the external surface area of the carbon as the authors clearly demonstrated? It is widely known that the mass activity declines with respect to the type of carbon used and their fraction of location on the carbon surface. This is why typically high surface area carbon based catalyst (Ketjen blacks) –where the 70% of the Pt particles are located inside the carbon support inside the porous structure- where they are not in direct contact with the sulfonic acid groups of the ionomer exhibit higher mass activity compared to a Vulcan based catalyst. This is widely known that the ionomer poisons the Pt particles and hence reduces its mass activity. So how do the authors explain the origin of their high mass activity since all Pt particles are located on the external surface area of the carbon and the Pt particles are covered by the ionomer as evidently shown by the dry proton accessibility data provided in this manuscript? The authors have to provide a scientific explanation for this –as their data and findings contradict basic principles of already established mechanisms. The authors did provide a hypothesis based on the presence of N groups but they do not explain how come their catalyst exhibits no ionomer poisoning effect at all.

6. Polarization curves section in the manuscript Line 250.258: The authors are misleading with their text by only referring the HFR corrected pol curves. The authors need to address in the manuscript that the HFR of the Pt/ cPDA MEA vs 10wt%Pt/C TTK MEA is clearly a factor of 2 higher at 100%RH. The authors

fail to comment on the origin of this. For someone with a lot of fuel cell experience it is clear that based on the nature of the two carbon size and secondary carbon structure/agreegate, this could only be rationalized as a contact resistance between the GDL and the Catalyst layer -as both MEAs had the same compression during cell assembly it could have not originated for GDL compression or flow field/GDL contact resistance . This is very critical information for real fuel cell application as it would affect the overall fuel cell performance and needs to be clearly what is origin of high HFR in the main manuscript.

The authors should first comment on the performance of the uncorrected polarization curves Pt/V vs Pt/cPDA –especially under air and the mass transport region and then address the HFR issue and its origin and the HFR correct pol curve. Then they authors could have a closing statement that if the CL design was further optimized to reduce the contact resistance between CL and GDL then the full benefit of using a ~100nm carbon particles as supports would be better exhibited –as future work.

As HFR corrected pol curves are of no use in real fuel cell application --the only thing that matters is the as measured fuel cell performance. This does not reduce the significance of the authors findings, it simply highlights that here is an issue that would need to be addressed In future studies and that catalyst layer design and optimization would be required to solve this high contact resistance between the GDL and CL- as this is a novel material after all and no one expects that there won't be issues In implantation for realistic fuel cell applications. Perhaps this could be added in the discussion section.

7. Figure 5: Replace all HFR corrected polarization curves with the uncorrected ones- and place the HFR corrected ones in SI. Alternatively , have the as measured and HFR corrected graph as 5a and 5b for comparison reasons in Air and leave the pure Oxygen plot in the SI –since the authors od not comment on the effect of pure oxygen vs air on the pol curves anyways. Also change the x-axis to A/cm² as this what is relevant for fuel cell application. The A/mgPt type of graphs can be put in SI and commented there accordingly- as these data are not relevant for real fuel cell application . This comments was already mentioned from Reviewer 2 : “ First of all, only iR corrected fuel cell performances were reported for both H₂/O₂ and H₂/air cells. This is not acceptable. Particularly, it does not make much sense to report iR corrected data for H₂/air cell. Uncorrected data must be reported since they represent how the cell will perform in real application” yet the authors keep insisting on presenting HFR corrected graphs in the main manuscript.

Minor Issues to be addressed:

8. Figure Caption 4: where is the b. referring too? “b im,0.9V measured at 300 kPaabs, recalculated at 150 kPaabs following the method explained in ref41”

9. Line 81, 85: I assume that you are referring to the particle diameter ? As "size" is not appropriate characterization of a particle dimension. Please change in the manuscript.

10. Line 187 –confinsingly – please delete as it makes no sense in that sentence.

11. Line 189 – please rephrase (suggestion) - grammatical error in sentence structure – suggestion in red to change too

“ A comparison with other Pt-based catalysts reported in literature by Harzer and coworkers³⁵ further confirms the impressive attribute of Pt/cPDA catalysts (Fig. 4d) of nearly 1.7 times higher mass activity compared to the 372 mA mgPt⁻¹ for Pt/KB TTK catalyst in an MEA.”

12. Line 227 ☐ replace the word "small" it is too general and can lead to misunderstandings as one would need to define also what a big and what a small particle would be referred to- better specify ~30nm average diameter particles (typical value for Vulcan and Ketjen type carbons) .

13. The DOE AST protocol clearly states that the temperature is 80°C and not 70°C as it is in the present work. So please rephrase your sentence here as this is misleading. I would recommend to state: that a DOE AST protocol was used with the only difference that the temperature was 70°C instead of 80°C in the present manuscript. And not simply say the DOE protocol for AST was followed. Also for more accurate comparison of your AST data you should compare with Harzer et al (Journal of The Electrochemical Society, 165 (6) F3118-F3131 (2018)) and not Stariha et al . As Stariha et all had a Pt loading of 0,2 mgPt/cm² for the PtV while Harzer had 0,1mgPt/cm² and it is closer to your Pt loading-as you will see from Harzer et all the Pt loading does have an impact on the ECSA loss percentage.

Reviewer #3 (Remarks to the Author):

the reviewer acknowledges that the authors put significant work in improving the manuscript. I can also agree to the point that the MEA measurements are the most important results of their work. This however does not change the fact that the RDE measurements are not performed at state of the art quality and it seems that the authors are not very familiar with the details of the analysis of RDE measurements. the authors show a KL-plot in figs5, also in their response. a linear dependence of the limiting current can be seen (at what potential are the currents "taken"???) but extending the plot to infinite rotation (low inverse sqrt(omega) values), should lead to much higher inverse currents. the point is to show that the extrapolations goes to zero on the y-scale (or very close to zero).

the referee agrees that also RDE measurements in perchoric acid do not display the situation in a mea and indeed nafion blocking etc is widely discussed. but the point of the rde is to establish kinetic parameters of catalysts without or with minimized effects of these parameters.

the authors explain difficulties in accessing a rde setup, therefore my recommendation would be to take these measurements out. if their quality is not sufficient and the mea data speak for themselves this might be the best option. including rde data of moderate quality is not recommended by the reviewer.

Reviewer #4 (Remarks to the Author):

The authors have addressed properly the questions raised by the reviewer 2. The only problem is that the authors have not been able to provide RDE data for TKK 10.2 wt% on Vulcan carbon as benchmark. However, I believe that the fuel cell data results are relevant enough to recommend the publication of the present work without that comparison.

We thank the reviewers for their additional comments. We have provided a detailed response to each of the points. The changes made to the text of main manuscript and supporting information has been also included in the response as text highlighted in green. These changes are also highlighted in green in the revised manuscript and SI.

Reviewer #1 (Remarks to the Author):

I greatly appreciate all the efforts the authors have done in addressing the most of the reviewers concerns. And the manuscript is by far in a better state than before. However they are still some major issues that have to be addressed properly before publishing in Nature.

Comment-1. In Supporting information you mention that :“Once dried, CCM with a geometrical area of 1 cm² was prepared by hot-pressing the catalyst against 25 μm thick Nafion 211 (NRE-211, Fuel cell store, USA) membrane at 150 °C and 2 MPa pressure for 3 minutes with an applied force of 0.12 kN cm⁻². The geometrical area of 1 cm² was controlled by kapton window. “

The CCM cannot be 1 cm² in area and at the same time be controlled by an active window of 1cm² via Kapton. Typically in subframed CCMs , the CCM is bigger than the active window- as also clearly illustrated in the picture you provided- the CCM (black) is bigger than 1cm² in the image (as the flow field is clearly ~7.1 x ~7.1 cm in order to be able to give a 50cm² geometric cell area). Please either correct the above statement or rephrase as it is misleading.

Response: We agree with the reviewer that the sentence regarding the CCM preparation method needs further clarification. We thank the reviewer for pointing it out. We have modified the sentence in the SI (page 15) accordingly and copied the modified part below.

“CCM with an active area of 1 cm² (sub-framed and controlled by kapton window) was prepared by hot-pressing catalyst coated decal against 25 μm thick Nafion 211 (NRE-211, Fuel cell store, USA) membrane at 150 °C and 2 MPa pressure for 3 minutes with an applied force of 0.12 kN cm⁻².”

Comment-2. Please make sure for future reference how you conduct your experiments- you also have convection in your experimental set up based on the way your MEA is placed on the flow field. I would recommend for future experiments not to place the CCM in the location where there is crossflow along the length of the channels. For example if you look at Ott et all or Baker et all or generally General Motors experiments/papers, all had the MEA placed in the same flow direction to avoid under the rib convection. As this significantly influences your measurements for the mass transport determination and fair comparison with literature. For example Orfanidi et al and Harzer et all used a 5cm² flow field where also In their case they must have had contribution from under the rib convection. In your case the way the CCM was placed you clearly have contribution of convection during the limiting current measurement which could result in lower mass transport values. Be carefull when comparing to other studies as your Mass transport values will be lower compared to other studies due to under the rib convection- if they did not had in their experiments.

Response: We thank the reviewer for recommending that in the future we consider the placement of CCM to minimize the effect of under-the-rib-convection. We also appreciate the reviewer's comments about exercising care when comparing mass transport values with other studies. We agree with the reviewer that the mass transport values will be affected by convection BUT only the *pressure-dependent* and *overall mass transport resistance* values and NOT *the pressure-independent value* which is dominated by local oxygen transport resistance.

To this end,

- First, we present *experimental* results for oxygen transport resistance from studies referenced by the reviewer. We include the literature data for cells suspected of rib convection (Orfanidi et al.; Harzer et al.) by the reviewer with cells supposedly having no rib convection (Ott et al.; Baker et al.) as per the reviewer suggested reference
- Next, we share results from a numerical study where the rib convection effect was examined for single serpentine and 3-parallel channel serpentine channels.
- Then, we discuss the original work by Baker et al. (GM group) who introduced the theory and methodology for determination of oxygen transport resistance wherein total oxygen transport resistance was broken into contributions from different transport mechanisms that were either pressure-dependent and pressure-independent.
- Finally, and more importantly, we illustrate that fundamentally the rib convection **does not impact** the **local** oxygen transport resistance. Thus, the configuration employed in our experiments does not affect our results for local oxygen transport and our claim that the new catalyst Pt/cDPA yields low $R_{O_2, local}$.

(i) Comparison of Transport Resistance in cells suspected of rib convection (Orfanidi et al.; Harzer et al.) with cells where no rib convection is expected (Ott et al.; Baker et al.): The reviewer stated that “*Orfanidi et al. and Harzer et al. used a 5cm² flow field where also In their case they **must** have had contribution from under the rib convection*”.

The reviewer also states that “*For example if you look at Ott et all or Baker et all or generally General Motors experiments/papers, all had the MEA placed in the same flow direction to avoid under the rib convection*”.

Notwithstanding the differences in diffusion media and CL properties, if the rib convection contributes **significantly** to reduce the O₂ transport resistance values, then we can expect that the oxygen transport resistance from cells in studies by Orfanidi et al. and Harzer et al. (as pointed by the reviewer) would be lower than those from cells in studies by Ott et al. and Baker et al. To examine this, the oxygen transport resistance data from the references mentioned by the reviewer is compared in the Figure 1 below. Three bars for each study represent the following three values - the total transport resistance, the pressure-dependent resistance, and the pressure-independent. As can be noted, the oxygen transport resistance data in Orfanidi et al. and Harzer et al. papers (suspected of enhanced mass transport as per the reviewer) is actually higher (when as per the reviewer it should have been lower) than those by Ott et al. and Baker et al. To reiterate, if the rib convection was significant and led to enhanced mass transport, then the total oxygen resistance would have been lower for experiments by Orfanidi et al. and Harzer et al.

Figure 1. Comparison of oxygen transport resistance reported in studies with cells with and without expected under-the-rib-convection.

Our conclusion: We do not see any obvious evidence of cells with configuration suspected of inducing rib convection to result in a lower total and/or pressure-independent oxygen transport resistance.

(ii) Numerical studies on rib-convection effect in cells with serpentine flow-field channels:

There are several numerical studies that have examined the effect of flow-field channel configuration on the fuel cell performance. A recent review article summarizes the flow-field patterns and their pros and cons (<https://doi.org/10.3389/fenrg.2020.00013>). Pertinent to the question of under-the-rib-convection, on study specifically examined the convection under the ribs of the flow-field (Wang et al., Journal of Power Sources, 2009, 193, 684–690) wherein the extent of under-the-rib-convection for single channel and three-channel configurations was examined. The geometry considered in their study is shown below. Note the flow-field used in our study has five parallel channels.

Figure 2. Schematic representation of single (left) and triple (right) serpentine flow fields

The gas diffusion layer parameters (specifically, the permeability) used in the model as well as total flow rates and pressure applied will affect the results of the model. However, broader conclusions can be drawn about the role of multiple channels on under-the-rib convection. The results presented by the authors (Wang et al., 2009) regarding the convection is reproduced below. In multiple channels, the ribs far away from the central rib experience very low convection effect. In this case, less than $1/10^{\text{th}}$ the velocity compared to single channel configuration. Roughly $1/5^{\text{th}}$ of the area under the 3 parallel channels experiences convection effect and 80% of the area experiences very less convection effect, if present. For 5-parallel channel configuration, it can be estimated that $1/10^{\text{th}}$ of the area under the flow-field would experience convection effect and 90% would experience smaller convection effect. Regardless, even if convection effect is present, it will be illustrated that rib convection does not impact the pressure-independent oxygen transport resistance determination.

4. Discussion

4.1. Sub-rib convection in cell

The gas velocities in diffusion layer under ribs 2 and 3, the said sub-rib convection velocities, were much higher for single serpentine flow than the triple serpentine flow cells (Fig. 6). Restated, the single serpentine flow field has much stronger sub-rib convection than the triple serpentine flow field. For example, at aspect ratio of 1.25, the velocity ranges $0.2\text{--}0.8\text{ m s}^{-1}$ under ribs 2 and 3 for the single serpentine flow field, while it ranges $0.04\text{--}0.07\text{ m s}^{-1}$ under rib 2 and $0.05\text{--}0.3\text{ m s}^{-1}$ under rib 3 for the triple serpentine flow field.

(iii) Original work by Baker et al. (General Motors/GM Group) for the determination of oxygen transport resistance: In this section of the response, we discuss why rib convection does not affect the *pressure-independent* oxygen transport resistances (and thereby, the local oxygen transport resistance), which are essentially the dominant contributors to the MPL and CL transport resistances. We start with the original work by GM researchers that proposed the theory and methodology for using limiting current analysis for quantifying the total oxygen transport resistance and subsequently break this total resistance into pressure-dependent resistance and pressure-independent resistance terms. This foundational approach has since been adopted in nearly all work reporting oxygen transport resistance. We also consider the well-established theory as well as experimental evidence of pressure-independence of Knudsen diffusion. We illustrate that even if rib convection were present, it will not affect the **local oxygen transport resistance ($R_{O_2, \text{local}}$)**.

Below is the text reproduced from Baker et al. (Journal of the Electrochemical Society, 2009) from GM group where the methodology for oxygen transport resistance obtained from limiting current measurements was first introduced.

The left panel discusses the phenomena that contribute to oxygen transport resistance in a fuel cell. Then, the concept of pressure-dependent (R_p) and pressure-independent (R_{NP})

transport resistance is introduced via equation (4). On the left panel, the physical processes affected by or contributing to R_p and R_{NP} is discussed.

Types and sources of transport resistance.— Several different types of oxygen-transport resistance occur in the cell. In the relatively large pores of the DM, oxygen-transport resistance is dominated by intermolecular diffusion. In the smaller pores of the MPL or electrode, Knudsen diffusion becomes important. A third type of transport occurs when oxygen diffuses through thin layers of liquid water or electrolyte. Diffusion coefficients for this type of diffusion can also be based on gas concentration differences, like intermolecular and Knudsen diffusion, by making use of Henry's law to relate the oxygen concentration in the water/ionomer to its equilibrium value in the adjacent gas. For moderate total pressures p , the binary diffusion coefficients for intermolecular gas diffusion are inversely proportional to p , whereas Knudsen diffusion coefficients and diffusion coefficients for liquid water/ionomer (combined with Henry's law) are independent of pressure. In what follows, we exploit this fact to separate the total transport resistance into pressure-dependent and pressure-independent components.

Suppose we assume that the total transport resistance R_T may be decomposed into pressure-dependent R_p and pressure-independent R_{NP} components according to

$$R_T = R_p + R_{NP} \quad [4]$$

Based on the above discussion, we expect that R_p is due to intermolecular gas diffusion, while R_{NP} is due to either Knudsen diffusion or diffusion through liquid water or ionomer. Despite its simplicity, the additive decomposition in Eq. 4 is not entirely self-evident because the different sources of diffusion may not always be arranged in series. For example, the MPL usually has a range of different pore sizes. Intermolecular diffusion may dominate in the larger pores, and Knudsen diffusion may dominate in the smaller ones. Because the gas diffusing through the MPL can access both pore sizes in parallel, an additive decomposition may not be strictly valid in this case. In our experiments, however, it turns out that the MPL contributes so little to R_{NP} that any error associated with assumption 4 is probably insignificant (see Fig. 21).

At least three component parts of the cell contribute to the total oxygen-transport resistance: the flow channels, the DM, and the MPL. Because these three components are arranged in series, the total transport resistance is just the sum of their separate resistances according to

$$R_T = R^{ch} + R^{DM} + R^{MPL} + R^{other} \quad [5]$$

where R^{other} represents all other sources of oxygen-transport resistance in the cell. While such an additive decomposition may be quite

As introduced by Baker et al., the total oxygen transport resistance obtained from limiting current measurement is broken into two parts: pressure-dependent (R_p) and pressure-independent (R_{NP}) resistances. The pressure-dependent resistance is attributed primarily to molecular diffusion of oxygen through the gas diffusion layer and *additional contribution from convective transport* in the channel and channel/gas diffusion layer interface. The pressure-independent resistance is attributed to Knudsen diffusion in catalyst layer and/or MPL plus the diffusion of oxygen through the ionomer thin film coating the catalyst.

Thus, if convection transport does exist, it will manifest in lowering the “pressure-dependent” resistance term and also the total oxygen transport resistance but not the pressure-independent resistance.

Breakdown of pressure-independent resistance (R_{NP}) into $R_{O_2,local}$ and $R_{Knudsen}$: R_{NP} is composed of CL gas phase O_2 transport resistance, which is mainly dominated by the restrictive Knudsen diffusion resistance ($R_{O_2,Knudsen}$), and the resistance of O_2 transport through the Pt–ionomer–water interface, also known as local O_2 -transport resistance ($R_{O_2,local}$). The local ionomer resistance is assigned to the interfacial effects at or near the Pt surface, which is assumed to scale inversely with the normalized Pt area (roughness factor, RF).

Comparison and claim made about oxygen transport resistance In our manuscript (figure 4e, mainbody of the manuscript), we had compared the **pressure-independent** O_2 transport resistance – the Figure is also reproduced below. It must be noted that the comparison is not selective rather it is a set of data for different catalyst layers with different platinum loading or roughness factor. It is a well-respected dataset from General Motors group. It is noteworthy that we had included three out of the four references mentioned by the reviewer (Harzer et al., Orfanidi et al. – which were suspected of under-the-rib-convection, and Ott et al. – which was not suspected of under-the-rib-convection; furthermore, we had included the results from Owejan et al. – from GM group)

Oxygen transport in catalyst layer occurs through nanopores and then by diffusion of oxygen through ionomer films covering the Pt catalyst. It is considered that Knudsen diffusion dominates through the pores. Both Knudsen diffusion resistance and ionomer film diffusion resistance is known to be pressure independent. Therefore, their combined effect is captured in the pressure-independent transport resistance terms.

The local or ionomer film diffusion resistance is inversely proportional to the total Pt surface area in the CL. This total Pt surface area is lumped into the Pt loading. Higher Pt loading ($\text{mg}_{\text{Pt}}/\text{cm}_2_{\text{electrode}}$) typically implies higher Pt surface area. Thus, with lower Pt loading, the pressure-independent transport resistance increases, generally. This can be noticed by comparing Ono et al and Ott et al data (Pt loading $\sim 0.1 \text{ mg}/\text{cm}^2$) with Orfanidi et al, Harzer et al and Owejan et al. ($0.025\text{-}0.078 \text{ mg}/\text{cm}^2$)

The pressure-independent oxygen transport resistance for Pt/cPDA catalyst is lowest despite having low Pt loading. This is one of the two of our primary claims.

Figure 3 (Figure 4e, main manuscript). Comparison of pressure independent O_2 transport resistance between Pt/cPDA and other Pt based catalysts reported in the literature (values inside the bracket indicate Pt loadings in $\text{mg}_{\text{Pt}} \text{ cm}^{-2}$)

Summary of our point/discussion:

- Harzer et al. and Orfanidi et al. cells suspected of rib convection effect actually have higher resistance (see Figure 1 above) and not lower resistance than Baker et al. and Ott et al. cells.
- Rib convection, if present can influence pressure-dependent resistance terms and total resistance but not the pressure-independent O_2 transport resistance (denoted as R_{NP} by Baker et al. and R_{PI} in other studies and $R_{\text{O}_2}^{P,ind}$ in current manuscript).
- The current work has compared only the pressure-independent resistance – which is not affected by rib convection. Hence, the concern raised by reviewer does not apply to the pressure-independent resistance comparison we are making.
- The local R_{O_2} is pressure-independent resistance – which is also unaffected by rib convection effect, if at all present.

- The comparison of pressure-independent R_{O_2} was not selective.
- In conclusion, our claim of low $R_{O_2, local} = 5.2$ s/cm should not be affected by the configuration used in the study, which could potentially lead to rib convection (although there is no direct evidence) but will **not affect pressure-independent** transport resistance term.

Nonetheless, we have revised the manuscript to add the following statement acknowledging the under-the-rib convection effect. (See page 31-32, SI)

“The $R_{O_2}^{P,dep}$ (molecular diffusion) part depends on the fuel cell hardware, i.e., the flow-field channel geometry, location of MEA on flow-field, and the GDL types. In serpentine flow-field, under the rib convection can occur between some channels where pressure difference between two channels is large. This effect has shown to be significant in single-serpentine channel compared to three-parallel channel²⁰. In five parallel channel configuration, such as that employed in our study, the contribution of under-the-rib convection to enhancing the overall oxygen transport would be even smaller. If present, the under-the-rib convection will manifest in lowering of pressure-dependent oxygen transport resistance component of the overall transport resistance. However, the Knudsen diffusion and $R_{O_2}^{P,ind}$ will not be affected by the convective transport. Caution must be exercised in comparing total oxygen transport resistance from different studies by examining if the configuration employed may lead to under-the rib convection.”

Comment-3. As the authors clearly have acknowledged in the Limiting current discussion section in SI, that the mass transport resistance can be affected by the ionomer film thickness over the Pt particles. The authors also provide the BET data with regards to the micro and mesoporous structure of the two different carbons. . It is widely known in literature that the effective ionomer film thickness will be lower for a carbon support with a lot of micropores (<2nm). Please refer to Liu et al (Liu et al 2011 J. Electrochem. Soc. 158 B614) for the details on how to calculate the effective I/C or ionomer film thickness over the external carbon surface (> 2nm) for a better comparison and more usefull discussion However they fail to discuss the dramatic difference in the effective ionomer film thickness between the two samples (Pt/V and Pt/cPDA) and how this difference is actually in favour of their findings. The authors needs to discuss the above in the manuscript.

For example the authors write Line 232: ‘‘As hypothesized, the large cPDA particles result in dramatic decrease in the Knudsen resistance, which is estimated to be around 0.04 s cm-1 compared to the values of 0.14 s cm-1 and 0.08 s cm-1 reported for Pt/KB and Pt/KB-N CLs16

It would be beneficial for the readers if the authors could also write after the above sentence that the cPDA had thicker Ionomer film (and give a value) covering the external cPDA particles compared to the KB and N-KB of another published study or even and hence it is evidently clear that the difference originates from the mesoporous structure. i think a statement like this would greatly enhance the authors claims and make it easier for other readers to make a fair comparison with literature presented data.

Response: We have added a few sentences to point out that ‘‘despite’’ higher ionomer thickness, the local O2 transport for our CL is lower (quoted below). On a related but different note, we do not agree with the assumption made by Liu et al. and other that only 2

nm pores are inaccessible to ionomers, which was an assumption without any direct proof. We have unpublished data wherein we systematically studied ionomer self-penetration in ordered pores of controlled pore sizes. Our results, indicate that even in 12 nm diameter pore, the ionomer does not diffuse/impregnate.

Regardless, the point made by the reviewer is an important one to explicitly highlight in our manuscript, which has been done on page 10 (line 241) of the revised manuscript.

“Despite having a higher estimated effective average ionomer thickness (~ 6 nm vs ~3.4 nm for Pt/V) due to the higher contribution from micropores in the Pt/cPDA catalyst, lower ionomer-related resistance of 5.0 s cm⁻¹ for Pt/cPDA is observed compared to the closest value of 8.63 s cm⁻¹ reported recently for Pt/KB-N”

Comment 4. Line 176 to Line 180: “A dry proton accessibility of 0.8 indicates high ionomer coverage. The cPDA particles possess hydrophilic characteristic as noted by spreading of water on a layer of particle during attempts to measure contact angle. The high ionomer coverage coupled with hydrophilic nature of support, attributed to the N-functional groups, should result in well hydrated Pt/ionomer interface even in dry conditions facilitating the proton accessibility to most of the Pt particles.”

The authors make bold statements “0.8 indicates high ionomer coverage” when they have no strong evidence to support that in my opinion. First of all, even though they have repeated the experiments 3 times –again there are no error bars in either the Pt/V or Pt/cPDA to actually demonstrate whether the 10% difference in the dry proton accessibility is actually a meaningful difference or simply experimental discrepancy. In addition, they compare in Figure S10 their data with that of Padgett et al. Padgett et al also used 10wt% Pt/V from TKK and got almost the same dry proton accessibility with Pt/cPDA. Hence how can the authors claim that the N groups have contributed to this 10% difference, since Padgett had a N-free carbon and yet showed almost the same values of dry proton accessibility? If the authors had shown ~90% dry proton accessibility for the cPDA, I would then agree with their statement as it would be above the value reported by Padgett. It might simply be that the authors 10wt%Pt/V catalyst layer was not as well manufactured as that of Padgett and hence this is why they observed the 10% difference in dry proton accessibility.

I am not questioning the fact that the N is homogeneously distributed or the N and ionomer interaction, I am simply questioning the boldness of the statements without having hard evidence to support it. I completely disagree with the way the above mentioned lines in the manuscript are written as they are misleading and not scientifically accurate. Please also when re-writing this section state that the 80% of cPDA shows the same value as reported by Padgett.

In addition, the presence of N groups does not automatically mean that the ionomer will be more homogeneously distributed. Recent publications from the group of Athanasov recently presented XPS study that showed that this is not the case and that the different N types could also change the ionomer orientation over the carbon support and hence affect the proton conductivity and hydrophilicity of the CL. In order to be able to claim this you need to provide proton conductivity measurements of the catalyst layer and compare it with the 10wt%Pt/V, which was not done in this study. If I were you I would write the above like this- as it is scientifically more accurate based on the presented data :

Line 176 to Line 180: ‘The dry proton accessibility value of the Pt/cPDA might indicate higher ionomer coverage. Nevertheless it has to be stated that a 80% dry proton accessibility has also been reported by Padgett et al for the commercial 10wt%Pt/C ,same catalyst used in this study (70%). The cPDA particles possess hydrophilic characteristic as noted by spreading of water on a layer of particle during attempts to measure contact angle. The slightly higher ionomer coverage of cPDA coupled with hydrophilic nature of support, could be attributed to the N-functional groups^{34,179} ,and is expected to result in well hydrated Pt/ionomer interface even in dry conditions facilitating the proton accessibility to most of the Pt particles.’

Response: We thank the reviewer for pointing this out. We were not trying to assert a bold claim but we do recognize that the statement was not balanced. As the reviewer indicated, it might be the case that our 10 wt% Pt/V catalyst layer was not as well manufactured as that of Padgett et al. and, therefore, this is why they observed the 10% difference in dry proton accessibility. Also, it should be note that Padgett et al. have used different ionomer for CL and different MEA than ours. Padgett et al. have used a shorter side chain ionomer (EW-950 vs 1100 in this study) with a higher ionomer to carbon ratio (0.95 vs 0.8 in this study, almost 18% higher), which would facilitate the proton transport even at lower RH. Also, they have used an 18 μm thick reinforced PFSA membrane vs 25 μm thick Nafion membrane in this study. It is a well-known fact that membrane plays a critical role in water management, which is very crucial for proton accessibility.

We have modified the text in the main manuscript (page 8, line 176) as suggested by the reviewer:

“A dry proton accessibility value of 0.8 compared to that of 0.7 for TKK Pt/C CL might indicate higher ionomer coverage or more homogenous ionomer distribution in the Pt/cPDA CL. Similar dry proton accessibility (80%) has also been reported by Padgett et al.³⁴ for 10 wt% Pt/V sample. However, the CL composition was different – Padgett et al.³⁴ employed a higher ionomer to carbon ratio (0.95 vs 0.8 in this study) and the ionomer had shorter side chain ionomer (EW-950 vs 1100). The hydrophilic nature of cPDA, likely due to N-functional groups³⁵, as noted by spreading of water on a layer of particle, may help to keep the ionomer/Pt and ionomer/cPDA interfaces well hydrated. Thus, the slightly high ionomer coverage coupled with hydrophilic nature of support, is expected to result in well hydrated Pt/ionomer interface even in dry conditions facilitating the proton accessibility to most of the Pt particles.

”

Comment 5. The discussion section of the main manuscript does not provide any link between the presented data and their hypothesis in the text nor do they provide a plausible explanation for the origin of the higher mass activity of the cPDA (clarification are given below)- **as the neglect a critical part of ionomer poisoning effect.**

To be more precise : What is the hypothesis of why the mass activity is higher despite the fact that all the Pt particles are located on the external surface area of the carbon as the authors clearly demonstrated? It is widely known that the mass activity declines with respect

to the type of carbon used and their fraction of location on the carbon surface. This is why typically high surface area carbon based catalyst (Ketjen blacks) –where the 70% of the Pt particles are located inside the carbon support inside the porous structure- where they are not in direct contact with the sulfonic acid groups of the ionomer exhibit higher mass activity compared to a Vulcan based catalyst. **This is widely known that the ionomer poisons the Pt particles and hence reduces its mass activity.**

So how do the authors explain the origin of their high mass activity since all Pt particles are located on the external surface of the carbon and the Pt particles are covered by the ionomer as evidently shown by the dry proton accessibility data provided in this manuscript? The authors have to provide a scientific explanation for this –as their data and findings contradict basic principles of already established mechanisms. **The authors did provide a hypothesis based on the presence of N groups but they do not explain how come their catalyst exhibits no ionomer poisoning effect at all.**

Response: It is our interpretation that the reviewer has two (2) concerns: (i) the primary concern of the reviewer is why no ionomer poisoning effect has been observed although they do acknowledge that we have hypothesized the presence of N groups plays a role, and (ii) a secondary concern is that in the Discussion section of the manuscript “no links between the presented data and our hypothesis in the text” has been provided.

(i) Ionomer poisoning effect for Pt/cPDA catalyst: As far as we can tell, **we did not claim that no ionomer poisoning exists** in our Pt/cPDA catalyst layer. Accordingly, the notion that we neglected ionomer poisoning effect is incorrect.

Ionomer poisoning of Pt catalyst is a topic of hot debate, especially because of the large suppression of activity of shape-controlled Pt and Pt-alloy catalysts in MEA compared to activity in liquid electrolyte with weakly adsorbing anionic species, e.g. electrolyte such as perchloric acid. In the previous response to reviewers document, we had shared some of the discussions in the literature pertaining the problem of low MEA activity for catalysts exhibiting high activity in liquid electrolyte. The suppression of activity in MEA compared to activity in sulphuric acid is expected to be much lower. For Pt/cPDA catalysts, the ORR activity in sulphuric acid (RDE) is 1.8 times the activity in MEA, so right away we can see that there is a suppression of activity of catalyst in the MEA.

We had mentioned the suppression of the ORR activity in MEA on page 8, line 199-201. It has now been modified to add the text in green: “*The much smaller suppression of ORR activity of Pt/cPDA catalyst in MEA due to ionomer poisoning compared to that determined in RDE can be attributed to the use of H2SO4 electrolyte.*”

In literature, the ionomer poisoning of catalyst in MEA has been quantified by comparing the activity of a catalyst in MEA and the activity of *ionomer-free* catalyst in HClO₄, which is an electrolyte with weakly adsorbing anionic species. The activity of ionomer-covered catalyst in HClO₄ is known to be suppressed by up to a factor of two and depends on the ionomer-content of the catalyst film deposited on the RDE. Since, we did not have any measurements for ionomer-free catalyst activity in HClO₄, and we have activity in ionomer-covered catalyst in H₂SO₄, we can only estimate the activity of ionomer-free Pt/cPDA catalyst in HClO₄. We do so by comparing the typical ratio of *ionomer-free* Pt catalyst specific activity (SA) in HClO₄ to that in H₂SO₄, i.e. SA_{HClO₄}/SA_{H₂SO₄}. Since, the specific activity of ionomer-free Pt catalyst and ionomer-coated Pt catalyst in H₂SO₄ is nearly identical, we can take the typical

value for $SA_{HClO_4}/SA_{H_2SO_4}$ and multiply by $SA_{H_2SO_4}$ of our Pt/cPDA catalyst to estimate the ionomer-free Pt catalyst activity in $HClO_4$.

In reference (Meier et al., Beilstein J. Nanotechnol. 2014, 5, 44–67), Mayrhofer group reported the SA and MA of several different Pt catalysts varying in Pt content and particle size (ECSA) in two different electrolytes - $HClO_4$ and H_2SO_4 . In Figure 4 below, the $SA_{HClO_4}/SA_{H_2SO_4}$ and $MA_{HClO_4}/MA_{H_2SO_4}$ for catalysts of different ECSA (m_2/g), i.e. different particle size, reported by Meier et al. is plotted. It can be gleaned that the activity of ionomer-free catalyst in perchloric acid is about 4-5 times the activity of ionomer-free catalyst in sulphuric acid as shown in Figure below.

Figure 4. Catalyst activity ratio ($HClO_4/H_2SO_4$) of different Pt based catalysts collected from Meier et al., Beilstein J. Nanotechnol. 2014, 5, 44–67

The activity of ionomer-free and ionomer-coated Pt catalyst in H_2SO_4 for both single-crystal Pt catalyst and nanoparticle Pt catalyst (Sarapuu et al, J Electroanal Chem, 2019, 848, 1, 113292 ; Subbaraman et al, ChemPhysChem 2010, 11, 2825 – 2833). As an example, the figure below (Figure 5) shows the i-V curves for Pt(111) catalyst uncoated and coated with Nafion in sulphuric acid are identical (Subbaraman et al, ChemPhysChem 2010, 11, 2825 – 2833). On the other hand, a suppression of activity is seen for Nafion-coated catalyst in perchloric acid compared to uncoated Pt catalyst in perchloric acid.

Figure 5. CV plots for the Pt(111) surface (both Nafion-free and Nafion-covered) in 0.1 M HClO₄ and 0.05 M H₂SO₄ (Subbaraman et al, ChemPhysChem 2010, 11, 2825 – 2833)

We can estimate the extent of sulphonic group poisoning effect for Pt/cPDA catalyst in MEA compared to its estimated activity in perchloric acid.

1. Activity of *ionomer-free* Pt catalyst in perchloric acid is known to be 4-5 times the activity in sulphuric acid from literature.
2. Activity of ionomer-free and ionomer-containing catalyst in sulphuric acid is nearly the same as briefly discussed above.
3. We can consider 4-5 as a range for the ratio for $(MA_{HClO_4})_{ionomer-free} / (MA_{H_2SO_4})_{ionomer-containing}$
4. Then, we can estimate MA_{HClO_4} / MA_{MEA} from the following:

$$MA_{HClO_4} / MA_{MEA} = [(MA_{HClO_4})_{ionomer-free} / (MA_{H_2SO_4})_{ionomer-containing}] \times [(MA_{H_2SO_4})_{ionomer-containing} / MA_{MEA}]$$

$$= [4 \text{ to } 5] \times [1.07 \text{ A/mg} / 0.64 \text{ A/mg}] = 6.6 \text{ to } 8.3$$

5. These estimate would imply that significant ionomer poisoning occurs in the MEA. The estimated suppression of activity is closer to that reported for very high activity catalysts such as Pt/Ni NW and jagged Pt catalyst.

Figure 6. Catalyst activity ratio (HClO_4/MEA) of different Pt based catalysts collected from literature.

Conclusion: Significant ionomer poisoning effect defined as catalyst activity in MEA to activity of ionomer-free catalyst activity in HClO_4 is estimated. It would appear that the Pt catalyst have intrinsically high mass

(ii) Links between the presented data and our hypothesis in the text: In the Discussion section, we had stated that: “attainment of uniform and well-dispersed deposition of Pt catalyst NPs with controlled size and inter-particle distance, both of which result in effective high ORR activity”

We had stated in the results section that: “The TEM image analyses of the Pt/cPDA catalyst (Fig. 3b), revealed that a majority of the particles are in 1.8-2.5 nm range, which is expected to exhibit high mass activity of $0.9\text{-}1.2 \text{ A mg}_{\text{Pt}}^{-1}$ as per the GCN correlation^{15,33}.”

In other words, we had linked the high activity to the size control – which was our design target. Insofar as the role of N-containing carbon support is concerned, we had not provided any specific hypothesis of how that may contribute to enhanced activity in an MEA. Internally, we have theorized that the hydrophilic nature of the cPDA support may be affecting many aspects of the ORR process as discussed below.

Potential role of hydrophilic carbonized PDA support on enhanced activity in MEA:

Although speculative at this stage, we suspect that the hydrophilic nature of carbonized PDA support creates a higher interfacial (ionomer/carbon and ionomer/Pt) water content. The higher interfacial water could potentially enhance the ORR activity in MEA considering the following results reported in the recent literature.

(i) High activity with higher acid dilution. Recent analyses of an older data set (Takeshita et al., Journal of Electroanalytical Chemistry 871 (2020) 114250) for ionomer-free Pt/C catalyst activity in perchloric acid with different molarity shows that the dilution of acid results in

increase in ORR activity – see Figure 7 below. This is counter-intuitive since dilution results in lower protonic activity. On the other hand, it does imply higher water content and also lower anion adsorption. If the higher water content is responsible for higher activity in these systems, then by extension, higher interfacial water in our Pt/cPDA catalyst layer because of hydrophilic nature of cPDA would potentially result in higher activity for our catalyst compared to classical, hydrophobic carbon (Vulcan carbon or Ketjen black carbon) supported Pt catalyst.

Figure 7. Specific activity of Pt catalyst increases with HClO₄ dilution (Figure from Takeshita et al, 2020)

(ii) Nature of interfacial water affects oxygen transfer energetics and ORR kinetics. Recent combined DFT/MD simulation study has found that less strongly bound water improves the gaseous oxygen transfer (to liquid layer) in the overall oxygen reduction reaction mechanism, which can lead to enhancement in activity by as much as a factor of two. If the hydrophilic cPDA promotes higher interfacial water content, then as per this study, the oxygen transfer step would be enhanced and a higher activity in MEA for Pt/cPDA would be expected compared to Pt/Vulcan catalysts.

PHYSICAL SCIENCES

Direct correlation of oxygen adsorption on platinum-electrolyte interfaces with the activity in the oxygen reduction reaction

Shiyi Wang¹, Enbo Zhu², Yu Huang^{2,3}, Hendrik Heinz^{1*}

The oxygen reduction reaction (ORR) on platinum catalysts is essential in fuel cells. Quantitative predictions of the relative ORR activity in experiments, in the range of 1 to 50 times, have remained challenging because of incomplete mechanistic understanding and lack of computational tools to account for the associated small differences in activation energies (<2.3 kilocalories per mole). Using highly accurate molecular dynamics (MD) simulation with the Interface force field (0.1 kilocalories per mole), we elucidated the mechanism of adsorption of molecular oxygen on regular and irregular platinum surfaces and nanostructures, followed by local density functional theory (DFT) calculations. The relative ORR activity is determined by oxygen access to platinum surfaces, which greatly depends on specific water adlayers, while electron transfer occurs at a similar slow rate. The MD methods facilitate quantitative predictions of relative ORR activities of any platinum nanostructures, are applicable to other catalysts, and enable effective MD/DFT approaches.

Copyright © 2021
The Authors, some
rights reserved;
exclusive licensee
American Association
for the Advancement
of Science. No claim to
original U.S. Government
Works. Distributed
under a Creative
Commons Attribution
NonCommercial
License 4.0 (CC BY-NC).

(iii) High water content reduces sulphonic group poisoning. The hydrophilic cPDA support resulting in higher interfacial water would also minimize the sulphonic group poisoning effect. Early work by Toyota Central R&D group (Kodama et al. *Electrochemistry Communications* 36 (2013) 26–28) has shown that the binding of sulphonic group with Pt is enhanced at low relative humidity or low water content. In other words, the poisoning is lesser when there is higher interfacial water. Hydrophilic cPDA can retain higher interfacial water at the ionomer-Pt/cPDA interface than that at conventional ionomer-Pt/C interface; Vulcan carbon and graphitized carbon surface are hydrophobic.

Fig. 3. Changes in the CV for the Nafion/Pt(111) interface in the solid-state cell after (a) the switching to the dry-gas flow and (b) the re-switching to the wet-gas flow. The potential is referred to the terminal potential of the RE.

Figure 8.

Summarized hypotheses of role of N-containing carbon in enhanced activity: The N-containing cPDA support is hydrophilic, which would result in *high interfacial water*.

Higher interfacial water would result in (a) higher interfacial acid dilution, which would result in enhanced ORR activity consistent with recent result from Toyota group (reference); (b) improved gaseous oxygen to water transfer, thereby in higher overall ORR rate as per the recent DFT/MD work (reference); (c) lower strength of absorptivity of sulphonic acid (lower sulphonic group poisoning) or lesser sulphonic group poisoning effect.

We have added the following text in the Discussion section of the manuscript (page 14, lines 307-314) to provide a possible explanation of enhanced activity, recognizing its speculative nature.

“Additionally, the hydrophilic nature of cPDA may be contributing to an enhancement in overall ORR activity. It is speculated that hydrophilic cPDA may result in higher interfacial water, which can then result in higher ORR activity consistent with higher activity for Pt/C catalyst upon dilution of perchloric acid⁴⁶, lower energy for oxygen transfer into liquid phase⁴⁷, and reduced poisoning by sulphonic acid upon hydration⁴⁸.”

Comment-6. Polarization curves section in the manuscript Line 250.258: The authors are misleading with their text by only referring the HFR corrected pol curves. The authors need to address in the manuscript that the HFR of the Pt/ cPDA MEA vs 10wt%Pt/C TKK MEA is clearly a factor of 2 higher at 100%RH. The authors fail to comment on the origin of this. For someone with a lot of fuel cell experience it is clear that based on the nature of the two carbon size and secondary carbon structure/agreegate, this could only be rationalized as a contact resistance between the GDL and the Catalyst layer -as both MEAs had the same compression during cell assembly it could have not originated for GDL compression or flow field/GDL contact resistance . This is very critical information for real fuel cell application as it would affect the overall fuel cell performance and needs to be clearly what is origin of high HFR in the main manuscript.

The authors should **first comment** on the performance of the uncorrected polarization curves Pt/V vs Pt/cPDA –especially under air and the mass transport region and then address the HFR issue and its origin and the HFR correct pol curve. Then they authors could have a closing statement that if the CL design was further optimized to reduce the contact resistance between CL and GDL then the full benefit of using a ~100nm carbon particles as supports would be better exhibited –as future work.

As HFR corrected pol curves are of no use in real fuel cell application --the only thing that matters is the as measured fuel cell performance. This does not reduce the significance of the authors findings, it simply highlights that here is an issue that would need to be addressed In future studies and that catalyst layer design and optimization would be required to solve this high contact resistance between the GDL and CL- as this is a novel material after all and no one expects that there won't be issues In implantation for realistic fuel cell applications. Perhaps this could be added in the discussion section.

Response: We thank the reviewer for pointing out the HFR issue. Our intention behind presenting the HFR-corrected performance was to negate the differences coming from iR drop since the focus of the work is on catalyst layer activity and transport resistance. It would be appreciated by the reviewer that while OEMs have access to state-of-the-art membranes (e.g. Gore membranes), most of the researchers do not have access to such materials. The performance of a cell with Nafion1 100 membrane and Gore membrane is decidedly different due to many factors including difference in HFR. In addition, most of the DOE target values are also to be measured at iR free matrix.

Nevertheless, we agree with the reviewer's point of view that uncorrected performance is more significant in real life application and thus we modified the text and figures in the main manuscript as suggested. Also provided below (main manuscript, page 12, lines 259-267):

“The cells with Pt/cPDA CLs exhibited high ohmic resistance (high frequency resistance or HFR) compared to cells with Pt/C CLs. Although the origin of high HFR for Pt/cPDA is not known, it is speculated to arise from contact resistance between CL and MPL. The HFR-free performance data is included in SI (Fig. S13). Overall, the Pt/cPDA catalyst layer illustrated better uncorrected performance ($560 \text{ mA/cm}^2_{\text{geo}}$ compared to that of $330 \text{ mA/cm}^2_{\text{geo}}$ of TKK Pt/C CL) as well as better HFR-free performance ($700 \text{ mA/cm}^2_{\text{geo}}$ compared to that of $420 \text{ mA/cm}^2_{\text{geo}}$ of TKK Pt/C CL) on an electrode geometric area basis at 0.6 V in air (Fig. 5a, Supplementary Fig. S13). TKK Pt/C CL showed a sharp drop in performance in the transport dominated region ($> 1000 \text{ mA/cm}^2_{\text{geo}}$) due to higher transport resistance as opposed to the Pt/cPDA CL.”

Figure 8. (Figure 5, main manuscript) (a) Uncorrected cell performance and (b) Uncorrected Pt mass loading normalized performance comparison between Pt/cPDA and commercial Pt/C (TKK 10 wt% Pt) catalyst in H₂/Air#, (c) ECSA loss during square wave AST degradation cycles (AST protocol: DOE square wave, 0.6-0.95 V, potential changed at $\sim 700 \text{ mV s}^{-1}$), at $70 \text{ }^\circ\text{C}$, 100% RH and atmospheric pressure in H₂/N₂ (0.2/0.2 NLPM), (d) Uncorrected cell performance at BOL and EOL (after 30000 AST cycles) at $70 \text{ }^\circ\text{C}$, 100% RH and 140 kPa pressure in H₂/Air, ECSA was measured using CO stripping method. #MEA condition: $70 \text{ }^\circ\text{C}$, 100% RH; 140 kPaabs, H₂/Air (0.3/0.5 NLPM).

Comment on HFR: We have included the following comment on the origin of HFR (SI page 25)

“The higher HFR values ($\sim 100 \text{ m}\Omega\text{-cm}^2$) of Pt/cPDA CL compared to that of Pt/V CL ($\sim 60 \text{ m}\Omega\text{-cm}^2$) could be attributed to the higher interfacial contact resistance between the Pt/cPDA CL and MPL.”

Inclusion of HFR issue in mainbody of manuscript (page 12):

“The cells with Pt/cPDA CLs exhibited high ohmic resistance (high frequency resistance or HFR) compared to cells with Pt/C CLs. Although the origin of high HFR for Pt/cPDA is not known, it is speculated to arise from contact resistance between CL and MPL.”

Inclusion of HFR issue in Discussion section (page 14, main manuscript, lines 318-319): We have added the following text in the Discussion section of the revised manuscript.

On the other hand, the approaches to minimize the high HFR for Pt/cPDA CLs must also be addressed if such materials are to be incorporated into MEA. Our results indicate that a more rational approach would be to start with a larger carbon or other electron-conducting support particle, coat it with thin layer of dopamine, carbonize the coating, and then deposit the Pt nanoparticles to achieve Pt size control and dispersion. Enhancement in oxygen transport resistance would then be achieved through dual benefits of large carbon support particles and favorable interaction of ionomer with N-containing functional groups of the dopamine coating. Further, investigation would be needed to determine optimal support size and to delineate the effect of N-functional group in controlling ionomer-support interaction.

Comment 7. Figure 5: Replace all HFR corrected polarization curves with the uncorrected ones- and place the HFR corrected ones in SI. Alternatively, have the as measured and HFR corrected graph as 5a and 5b for comparison reasons in Air and leave the pure Oxygen plot in the SI –since the authors do not comment on the effect of pure oxygen vs air on the pol curves anyways. Also change the x-axis to A/cm² as this what is relevant for fuel cell application. The A/mgPt type of graphs can be put in SI and commented there accordingly- as these data are not relevant for real fuel cell application . This comments was already mentioned from Reviewer 2: ‘ ‘ First of all, only iR corrected fuel cell performances were reported for both H₂/O₂ and H₂/air cells. This is not acceptable. Particularly, it does not make much sense to report iR corrected data for H₂/air cell. Uncorrected data must be reported since they represent how the cell will perform in real application’ ’ yet the authors keep insisting on presenting HFR corrected graphs in the main manuscript.

Response:

We thank the reviewer for the feedback. Previously, we provided the uncorrected performance in the SI as suggested by the reviewer-2. The reviewer did not specify that the uncorrected performance needed to be included in the main body, not in the SI. Nevertheless, as indicated by the reviewer, we replaced the HFR free performance with the uncorrected ones in Figure 5 of main manuscript.

Minor Issues to be addressed:

Comment 8. Figure Caption 4: where is the b. referring too? ‘ ‘b im,0.9V measured at 300 kPaabs, recalculated at 150 kPaabs following the method explained in ref41’ ’

Response: The superscript “a” and “b” is related to the mass activity measuring conditions for Pt/HSC and GBP-Pt-600 in Figure 4d.

Comment-9. Line 81, 85: I assume that you are referring to the particle diameter ? As “size” is not appropriate characterization of a particle dimension. Please change in the manuscript.

Response: we thank the reviewer for pointing it out. We have replaced size with diameter in the mainbody.

Comment-10. Line 187 –confinsingly – please delete as it makes no sense in that sentence.

Response: We have deleted it.

Comment-11. Line 189 – please rephrase (suggestion) - grammatical error in sentence structure – suggestion in red to change too

“ A comparison with other Pt-based catalysts reported in literature by Harzer and coworkers³⁵ further confirms the impressive attribute of Pt/cPDA catalysts (Fig. 4d) of nearly 1.7 times higher mass activity compared to the 372 mA mgPt⁻¹ for Pt/KB TTK catalyst in an MEA.”

Response: we thank the reviewer for pointing it out. We have corrected the grammatical error.

Comment-12. Line 227 ◊ replace the word “small” it is too general and can lead to misunderstandings as one would need to define also what a big and what a small particle would be referred to- better specify ~30nm average diameter particles (typical value for Vulcan and Ketjen type carbons).

Response: we thank the reviewer for pointing it out. We have corrected the sentence as follows: (in main manuscript page 9, line 231)

In fact, it appears that the $R_{O_2}^{P,ind}$ trendlines observed for Pt-catalyst supported on conventional used ~30 nm average diameter carbon black particles are remarkably shifted to left and downwards, i.e., toward a lower R_{O_2} (Figure S18f).

Comment-13. The DOE AST protocol clearly states that the temperature is 80°C and not 70°C as it is in the present work. So please rephrase your sentence here as this is misleading. I would recommend to state: that a DOE AST protocol was used with the only difference that the temperature was 70°C instead of 80°C in the present manuscript. And not simply say the DOE protocol for AST was followed. Also for more accurate comparison of your AST data you should compare with Harzer et al (Journal of The Electrochemical Society, 165 (6) F3118-F3131 (2018)) and not Stariha et al . As Stariha et all had a Pt loading of 0,2 mgPt/cm² for the PtV while Harzer had 0,1mgPt/cm² and it is closer to your Pt loading-as you will see from Harzer et all the Pt loading does have an impact on the ECSA loss percentage.

Response: we thank the reviewer for pointing it out. We have corrected the sentence as follows: (main manuscript page 18, line 275-279)

The durability of the Pt/cPDA catalyst and the reference TTK Pt/C catalyst was also assessed over 30,000 potential cycles following the DOE suggested square wave (SW)

accelerated stress test (AST) protocol (0.6 – 0.95 V, at 70 °C, 100% RH and atmospheric pressure in H₂/N₂ (0.2/0.2 NLPM), details in Method, Supplementary information) with the only exception that the temperature was 70°C instead of 80°C.

We appreciate the reviewer's suggestion of comparing the durability with the similarly loaded CL. However, for more meaningful comparison, we compared it with similar roughness factor in-house CL as the hardware and operating conditions are similar.

Reviewer #3 (Remarks to the Author):

Comment: the reviewer acknowledges that the authors put significant work in improving the manuscript. I can also agree to the point that the MEA measurements are the most important results of their work. This however does not change the fact that the RDE measurements are not performed at state of the art quality and it seems that the authors are not very familiar with the details of the analysis of RDE measurements. the authors show a KL-plot in figs5, also in their response. a linear dependence of the limiting current can be seen (at what potential are the currents "taken"???) but extending the plot to infinite rotation (low inverse $\sqrt{\omega}$ values), should lead to much higher inverse currents. the point is to show that the extrapolations goes to zero on the y-scale (or very close to zero).

the referee agrees that also RDE measurements in perchoric acid do not display the situation in a mea and indeed nafion blocking etc is widely discussed. but the point of the rde is to establish kinetic parameters of catalysts without or with minimized effects of these parameters. the authors explain difficulties in accessing a rde setup, therefore my recommendation would be to take these measurements out. if their quality is not sufficient and the mea data speak for themselves this might be the best option. including rde data of moderate quality is not recommended by the reviewer.

Response: We agree with the reviewer's comments regarding the RDE measurements. The limiting currents were taken at 0.4 V. The extension to infinite rotation should indeed lead to close to zero. The non-zero intercept could arise either due to thick films on the glassy carbon electrode or other experimental artefacts.

We can take out the RDE data from the supporting information, if the Editor does deem it necessary. Some minor edits to the main manuscript would have to be made.

We have performed triplicates of measurements for MEA activity measurements and we agree that these data do stand on their own.

Reviewer #4 (Remarks to the Author):

Comment: The authors have addressed properly the questions raised by the reviewer 2. The only problem is that the authors have not been able to provide RDE data for TTK 10.2 wt% on Vulcan carbon as benchmark. However, I believe that the fuel cell data results are relevant enough to recommend the publication of the present work without that comparison.

Response: We agree with the reviewer's comments.

REVIEWER COMMENTS

Reviewer #5 (Remarks to the Author):

The authors have thoroughly answered the questions raised by each reviewer. For the specific point related to RDE testing, I would recommend:

- 1) Add the electrolyte type/purity/final concentration to the experimental section of the RDE work.
- 2) Perform testing in HClO₄ (w/wo ionomer) and add some discussion of the results to the main body of the manuscript. Comparing the activity of the baseline catalyst to the authors' new catalyst in these two scales (RDE/MEA) would be helpful in understanding the origin of the high mass activity of the authors' purely Pt-based catalyst. If the N-species are helping to suppress ionomer-binding to the Pt sites, this should also be observed from the RDE tests. Furthermore, RDE testing in HClO₄ w/wo ionomer could be used to properly quantify this effect by comparing the 'suppression' observed for the baseline catalyst vs. the Pt/cPDA. Since this seems to be a key point in the current manuscript, and considering that the relevant RDE tests are not onerous, it is suggested to complete this work prior to publishing.

We thank the reviewer for the comments. The response to reviewer's comments are provided below. The changes made to the text of main manuscript and supporting information has been highlighted in yellow.

Reviewer #5 (Remarks to the Author):

Comment: The authors have thoroughly answered the questions raised by each reviewer.

Response: We appreciate the reviewer's acknowledgment of our thorough response.

Comment: For the specific point related to RDE testing, I would recommend:

1) Add the electrolyte type/purity/final concentration to the experimental section of the RDE work.

Response: The electrolyte information has been added to a new/updated section for the RDE work on Page 12 of Supplementary information. Perchloric acid of the highest quality (same as that reported in the highly respectable and carefully carried out works of Kocha et al, Shinozaki et al and Garsany et al – see SI for reference) was used. Specifically, perchloric acid 70% Veritas Doubly Distilled from GFS Chemicals, USA was procured and diluted to 0.1M concentration by adding deionized water. All glasswares were carefully cleaned (as described on Page 12 of SI) following the cleaning protocols described in the literature.

Comment: 2) Perform testing in HClO₄ (w/wo ionomer) and add some discussion of the results to the main body of the manuscript. Comparing the activity of the baseline catalyst to the authors' new catalyst in these two scales (RDE/MEA) would be helpful in understanding the origin of the high mass activity of the authors' purely Pt-based catalyst. If the N-species are helping to suppress ionomer-binding to the Pt sites, this should also be observed from the RDE tests. Furthermore, RDE testing in HClO₄ w/wo ionomer could be used to properly quantify this effect by comparing the 'suppression' observed for the baseline catalyst vs. the Pt/cPDA. Since this seems to be a key point in the current manuscript, and considering that the relevant RDE tests are not onerous, it is suggested to complete this work prior to publishing.

Response: We have completed the work suggested by the reviewer. We undertook the RDE work by procuring new RDE setup (Pine Instrument) and developing in-house expertise in preparing good films. Preparation of good films proved to be a significant undertaking despite the knowledge available in the literature. In addition to film preparation, careful work was also required for cleaning the glasswares and all pertinent components of the RDE setup. The quality of data was also verified by studying the electrochemical characteristics of Poly Pt (Pt disk, Pine Instrument, USA). Several weeks of trial and error work were required to obtain "good quality" films and optimization of the ink (e.g. sonification time).

Comparison of Pt/cPDA and baseline catalyst in RDE (ionomer-free): Ionomer-free activity of Pt/cPDA (8.5 wt% Pt) and commercial Pt/C (10 wt% Pt, TKK) in 0.1M HClO₄ were obtained from linear sweep voltammetry measurement. The Figure below shows higher activity of Pt/cPDA catalyst compared to commercial activity

Since the Pt loading can differ slightly from sample (catalyst film) to sample, it is meaningful to compare specific activities. The kinetic current obtained from LSV (Figure above) were normalized to the Pt surface area (obtained from CV – Hupd and CO stripping) for **the same sample** to obtain specific activity in mA/cm²_{Pt}. The specific activity for ionomer-free Pt/cPDA catalyst was determined to be **0.95 mA/cm²_{Pt}** which is more than double that for the commercial Pt/C catalyst (**0.43 mA/cm²_{Pt}**). This confirms that the intrinsic activity of the Pt/cPDA catalyst is itself higher than the comparator Pt/C catalyst that was used also for performance comparison in MEA.

Comparison of suppression due to ionomer: The Pt/cPDA activity was also found to have less suppression (24%) in specific activity (0.95 mA/cm²_{Pt} for ionomer free vs 0.72 mA/cm²_{Pt} for ionomer-containing catalyst films) compared to Pt/C commercial or baseline catalyst, which exhibited 42% suppression (0.43 mA/cm²_{Pt} for ionomer free vs 0.23 mA/cm²_{Pt} for ionomer containing catalyst films). The high suppression of Pt/C catalyst at I:C ratio of 0.2 is also consistent with the data reported in literature as shown in Figure below.

As suggested by the reviewer, “RDE testing in HClO₄ w/wo ionomer” confirms the lower extent of activity suppression (due to ionomer) observed in RDE experiments which supports the higher activity of Pt/cPDA catalyst in MEA.

Discussion added in the manuscript: As suggested by the reviewer, we have added the following text in the manuscript mainbody.

“The much smaller suppression of ORR activity of Pt/cPDA catalyst in MEA due to ionomer poisoning was also observed in liquid electrolyte (0.1 M HClO₄) as evident from comparison of activity between ionomer-free and ionomer-containing films (Fig. S7; Table S4). Whereas 42% suppression of activity was observed for commercial Pt/C catalyst (10 wt% TKK) at I:C ratio of 0.2 consistent with literature (Fig S8), Pt/cPDA catalyst at similar I:C ratio showed 24% suppression. The exact role of cPDA support in minimizing the activity suppression is not fully understood and may be due either to the ionomer/cPDA interactions affecting the Pt/ionomer interfacial structure or to the nucleation and growth of Pt crystallites with facets less prone to ionomer poisoning.”